# Identifying a key spot for electron mediator-interaction to tailor CO dehydrogenase's affinity

Suk Min Kim [1,4,5] ✉, Sung Heuck Kang[1,4], Jinhee Lee[1,4], Yoonyoung Heo[2,4], Eleni G. Poloniataki [1], Jingu Kang[1], Hye-Jin Yoon[2], So Yeon Kong[2], Yaejin Yun[2], Hyunwoo Kim[1], Jungki Ryu [1], Hyung Ho Lee [2,5] ✉ & Yong Hwan Kim [1,3,5] ✉

Fe–S cluster-harboring enzymes, such as carbon monoxide dehydrogenases (CODH), employ sophisticated artificial electron mediators like viologens to serve as potent biocatalysts capable of cleaning-up industrial off-gases at stunning reaction rates. Unraveling the interplay between these enzymes and their associated mediators is essential for improving the efficiency of CODHs. Here we show the electron mediator-interaction site on *Ch*CODHs (*Ch, Carboxydothermus hydrogenoformans*) using a systematic approach that leverages the viologen-reactive characteristics of superficial aromatic residues. By enhancing mediator-interaction (R57G/N59L) near the D-cluster, the strategically tailored variants exhibit a ten-fold increase in ethyl viologen affinity relative to the wild-type without sacrificing the turn-over rate ($k_{cat}$). Viologen-complexed structures reveal the pivotal positions of surface phenylalanine residues, serving as external conduits for the D-cluster to/from viologen. One variant (R57G/N59L/A559W) can treat a broad spectrum of waste gases (from steel-process and plastic-gasification) containing $O_2$. Decoding mediator interactions will facilitate the development of industrially high-efficient bio-catalysts encompassing gas-utilizing enzymes.

Industrial emissions of gases such as carbon monoxide (CO), carbon dioxide ($CO_2$), nitrogen ($N_2$), and hydrogen ($H_2$) are progressively considered abundant sources for sustainable chemical and energy conversion[1–4]. The use of biocatalysts to clean and utilize gaseous substances from industrial flue gases is garnering increasing attention in sustainable chemical industries owing to excellent performances under benign reaction conditions. In the steel-making industry, an overwhelming amount of waste gases containing CO and $CO_2$ are generated from POSCO (the largest South Korean steelmaking company) over 21.9 billion $Nm^3$ every year[5,6]. Enzymatic waste gas conversion, requiring no capturing or pre-treatment process, can provide

an alternative route unconstrained by inhibitors (CN, $SO_x$) in industrial gas mixtures[5]. For instance, CO and $CO_2$ in steel-making flue gases can be readily utilized as substrates for CO dehydrogenase (CODH) and formate dehydrogenase (FDH), respectively.

For the enzymatic conversion of such gases (CO, $CO_2$, $H_2$, and $N_2$), an electron mediator is essential for their redox reactions[1,7]. However, most gas-utilizing enzymes rely on unique electron carrier proteins specific to the individual strain and protein, hindering in-vitro applications of gas-converting enzymes. Viologens offer a promising alternative as universal and easily applicable electron mediators for various gas-converting metalloenzymes, including CODH and FDH[8]. The

[1]School of Energy and Chemical Engineering, Ulsan National Institute of Science and Technology (UNIST), 50 UNIST-gil, Ulsan 44919, Republic of Korea. [2]Department of Chemistry, College of Natural Sciences, Seoul National University, 1 Gwanak-ro, Gwanak-gu, Seoul 08826, Republic of Korea. [3]Graduate School of Carbon Neutrality, Ulsan National Institute of Science and Technology (UNIST), 50 UNIST-gil, Ulsan 44919, Republic of Korea. [4]These authors contributed equally: Suk Min Kim, Sung Heuck Kang, Jinhee Lee, Yoonyoung Heo. [5]These authors jointly supervised this work: Suk Min Kim, Hyung Ho Lee, Yong Hwan Kim. ✉e-mail: smkimlife@unist.ac.kr; hyungholee@snu.ac.kr; metalkim@unist.ac.kr

viologen-based enzymatic conversion of CO or $CO_2$ into valuable chemical compounds has recently gained traction due to their higher catalytic efficiency under mild conditions[4–6]. Nonetheless, the interactions between artificial electron mediators (viologens) and metalloenzymes for electron transfer remain elusive.

Redox mediators, such as viologens (V, 4,4′-bipyridinium salts), are renowned for their three reversible states: dication $V^{2+}$ (pale yellow/colorless), radical cation $V^{+\cdot}$ (violet/blue/green), and neutral $V^0$ (colorless)[9]. These versatile and cost-effective mediators have been widely employed in in-vitro electrochemical and photochemical redox systems as alternatives to native cofactors (NAD(P)H, FMN)[10,11]. Intriguingly, most enzymes associated with Climate Change that convert CO and $CO_2$ gases commonly utilize viologens (Supplementary Table 1).

Among viologen-utilizing biocatalysts, Ni–Fe carbon monoxide dehydrogenases (CODHs) with a unique $[NiFe_4S_4OH_x]$ cluster (E.C.1.2.7.4)[12,13] serve as suitable model enzymes for studying viologen mediator interaction sites. CODHs have high-resolution 3D structural information available and share highly conserved structures as simple homodimers[5,13–16]. Our recent study demonstrated that customizing oxygen-tolerant CODHs with a high-efficiency rate holds significant implications for industrial bioconversion from various syngas mixtures (CO, $CO_2$, and $H_2$)[5,17,18]. For the enzymatic utilization of industrial waste gases, oxygen tolerance, and efficient mediator interaction—closely linked to electron transfer—are crucial issues for most gas-utilizing enzymes[8]. However, the latter remains unsolved, which is why there is a need to study the viologen interaction with Ni–Fe CODHs.

In this study, we identified the viologen interaction site in ChCODHs (Ch, Carboxydothermus hydrogenoformans) through systematic alanine substitutions of aromatic surface residues, which disrupted enzyme activity. By scrutinizing surface residues and conserved amino acids in Ni–Fe CODHs, we discovered the critical site for viologen interactions at the D-cluster of ChCODH2. We confirmed it by determining the crystal structures of ChCODH2 in a complex with viologens. Some mutations led to enhanced kinetic properties for viologens, increasing mediator affinity by approximately 10-fold compared to the ChCODH2 wild type (WT) while maintaining the turn-over rates, as well as the resistance to industrial off-gases.

## Results

### Candidate surface residues for viologen interaction

All characterized Ni–Fe CODHs exhibit activity towards viologens, such as methyl viologen (MV), ethyl viologen (EV), and benzyl viologen (BV) (Supplementary Table 1). In the homodimeric ChCODH2-structure, the active site C-cluster is deeply buried on the protein surface, distinct from the B and D clusters. Moreover, viologen molecules are improbable to access the catalytic site C-cluster and B-cluster directly via substrate tunnels due to their relatively larger sizes than substrate CO or $CO_2$ (Supplementary Fig. 1). Based on this observation, we deduce that residues implicated in viologen interaction may be surface-exposed, viologen-interactive, and electron transferable.

Aromatic residues phenylalanine (F), tyrosine (Y), and tryptophan (W) are recognized to facilitate electron transfer[19,20]. These residues are more likely candidates than sulfur-containing residues for potential interaction. To identify candidate viologen-interacting residues, we initially examined aromatic residues that could interact with viologens on the protein surface (Fig. 1a). Indeed, aromatic (F, Y, and W) and sulfur-containing (C, cysteine; M, methionine) residues were displayed on the surface of the protein structure (Supplementary Table 2 and Supplementary Fig. 2).

Therefore, we verified the reactivities of Escherichia coli-expressed CODH variants with alanine-substituted surface residues (F, Y, W) to determine the most probable surface-exposed residue interacting with the EV mediator (Fig. 1b, Supplementary Table 3).

### Viologen responses of alanine–replaced ChCODH variants

On the surface of the ChCODH2 protein (Protein Data Bank (PDB) 1SU7)[14], the eight residues of F, Y and W appear as exposed (Fig. 1a). To search mediator interacting sites in the ChCODH2, we monitored the change in the activity of alanine-replaced variants by using EV as electron acceptor (Fig. 2a). Relative to ChCODH2 WT activity (100%), F41A exhibited a dramatic reduction in activity (<0.01%). In contrast, F61A (58.8%), W636A (65.5%), W29A (75.7%), F234A (80.9%), Y32A (85.9%), and F386A (89.5%) showed a moderate decrease in activity respectively. Y224A showed a minor activity increase (117%), likely influenced by protein purity. Unlike aromatic F41 (WT) and F41Y, aliphatic F41 variants (F41A, F41V, F41L) exhibited less than 0.5% relative activities (Supplementary Table 4). These findings imply that position F41, with nearly abolished activity in the F41A variant, is crucial for viologen reduction.

Additionally, surface-exposed C and M variants, including M355A (61.1%), M116A (116.7%), and C496A (60.6%), exhibited increased relative activity (Supplementary Fig. 3). Except C344A, which was not expressed in E. coli, surface-exposed C and M variants displayed no significant alterations in reactivity toward EV. Figure 2b shows the pivotal role of F41 in mediator interaction, irrespective of viologen type, encompassing MV, EV, BV, and diquat (DQ). Consequently, we deduced that residue 41 near the D-cluster constitutes the pivotal site for viologen interactions in ChCODH2.

The observed loss of activity in F41 variants of ChCODH2 prompted us to consider whether the alanine mutation affected more than just electron transfer, potentially influencing the properties of the adjacent D-cluster, including its redox potential[21]. To explore this possibility, we carried out viologen-free electrochemical studies, attaching the enzyme variants to an electrode to observe any changes in redox potential (Supplementary Fig. 4). The results showed no significant difference in redox potentials between WT and F41 variants (0.28 V vs RHE), indicating that the F41 mutations do not alter the redox potential of the D-cluster. Furthermore, the structure of the F41C variant showed no changes in the FeS cluster environments, including the D-cluster (Supplementary Fig. 5). Thus, the diminished activities in the viologen reduction of F41 aliphatic variants are likely attributed to disruptions in electron transfer, affirming the key role of F41 in facilitating viologen interactions and electron transfer within ChCODH2.

### Common phenylalanine residues for EV interaction in other CODHs

The D-cluster, an exposed Fe-S cluster, is crucial for the catalytic activity of CODH[16], as it facilitates electron transfer during the conversion of CO and $H_2O$ to $CO_2$ and $2H^+$. Ni–Fe CODHs feature two distinct D-cluster metal contents: the 4Fe–4S cluster and the 2Fe–2S cluster (Fig. 2c). The cubane-like 4Fe–4S cluster is larger than the butterfly-like 2Fe–2S cluster, with corresponding differences in the amino acid sequences of the 4Fe–4S and 2Fe–2S groups (Fig. 2c and Supplementary Fig. 6 and 7). The 4Fe–4S clusters are generally coordinated by four conserved cysteine residues[16] at the D-cluster (Cys-X-Phe-X_5-Cys) within each monomer. The 2Fe–2S cluster is coordinated by four cysteine residues[16], but with a closely positioned pair of cysteine residues (Cys-X-Phe/Tyr-Cys) in each monomer. All sites corresponding to ChCODH2 F41 are highly conserved as F or Y residues.

Two representative structures, 4Fe–4S RrCODH (Rr, Rhodospirillum rubrum; PDB 1JQK)[13] and 2Fe–2S DvCODH (Dv, Desulfovibrio vulgaris; PDB 6OND)[22], individually exhibit each D-cluster coordination (Fig. 2c). Consequently, we investigated whether surface phenylalanine residues commonly interact with viologens in other CODH enzymes. We monitored the viologen-based activities of WT and variants from RrCODH and DvCODH using the same approach employed for ChCODH2. Figure 2d reveals that F43 and F44 positions in RrCODH and DvCODH, respectively, are essential for viologen reduction using the four mediators MV, EV, BV, and diquat (DQ). These findings clearly

**a**

$$CO + H_2O + 2Md_{ox} \rightarrow CO_2 + 2H^+ + 2Md_{red}$$

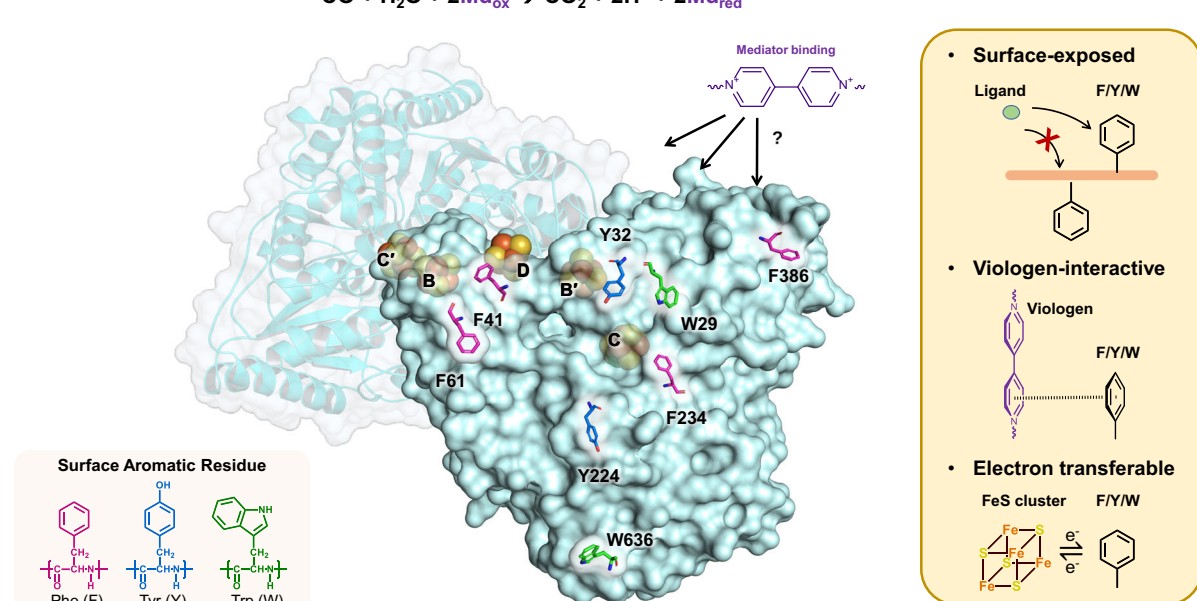

**b**

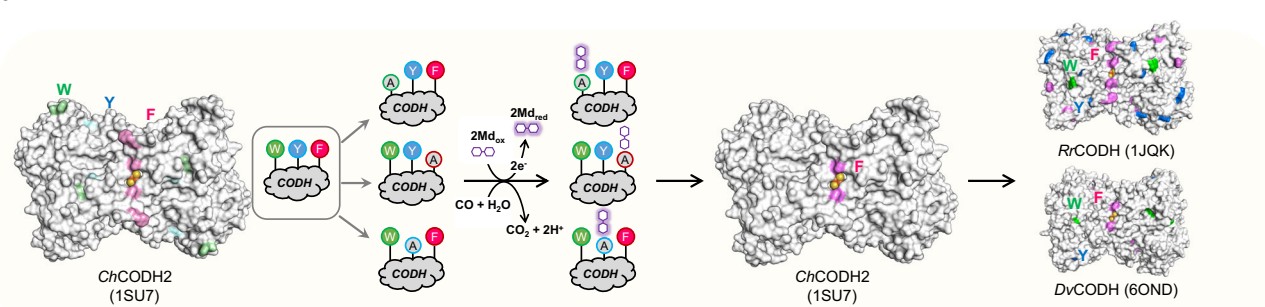

**Fig. 1 | A rational approach for identifying potential sites to interact with viologen in _Ch_CODH2. a** Illustration of surface-exposed aromatic residues. In the _Ch_CODH2 structure (PDB ID: 1SU7)[14], the surface-exposed aromatic residues (Phe, Tyr, and Trp) are putative sites for viologen interaction. **b** Procedure to identify key sites for interacting with CODH enzymes and viologen. Aromatic residues on the enzyme surface were replaced with alanine residues, likely alanine braille. The influences of mutations on enzymatic activity show relationships between target residues and viologen. This approach could be expanded to other CODHs including _Rr_CODH (PDB 1JQK)[13] and _Dv_CODH (PDB 6OND)[22].

show that the surface phenylalanine positions (regions 40–44) of the D-cluster commonly serve as critical sites for viologen interactions.

### EV responses and catalytic properties of CODH variants

Consistent with observations in Ni-Fe CODHs, we hypothesized that the sites of _Ch_CODH2 F41A, _Rr_CODH F43A, and _Dv_CODH F44A participate in EV interactions and are highly conserved. Due to that, we alternatively explored neighboring residues of F41 as potential candidates for intimate EV interactions. In the _Ch_CODH2 structure, 11 residues reside within 4 Å of F41 (Supplementary Table 5). Excluding residues with distinct roles or inward-facing side chains leaves only R57 and N59 (Fig. 3a). Generally, Ni-Fe CODHs exhibit poor EV affinities ($K_M > 2$ mM; Supplementary Table 6). Analyzing R57A and N59A activities at low (2 mM) and high (20 mM) EV concentrations suggests that these residues may influence viologen interactions, as evidenced by activity ratios compared to WT. We subsequently examined various R57 and N59 mutations and their activity changes in response to EV concentration, revealing that positions 57 and 59 affect EV interaction, with position 57 exerting a more substantial impact (Fig. 3b).

To evaluate alterations in viologen affinity, we determined the catalytic properties of the respective effects of size, charge, and hydrophobicity at positions 57 and 59 of _Ch_CODH2 toward EV (Table 1 and Supplementary Table 7, Supplementary Fig. 8 and 9). The Michaelis constant ($K_M$) for EV in _Ch_CODH2 WT was estimated to be 2.3 mM, a value similar to the recently reported 1.8 mM by our group[5] but lower than the 4 mM for MV described by the Ragsdale group[23]. Kinetic data reveal that mutations at positions 57 and 59 exhibit discernible effects on viologen affinity. Notably, the R57G/N59L mutant displayed a 10-fold enhanced mediator affinity ($K_M \approx 0.2$ mM at 30 °C) compared to WT without compromising the catalytic rate ($k_{cat} \approx 2,100$ at 30 °C). In contrast, mutating two residues, _Ch_CODH2 F41A and _Rr_CODH F43A, in the putative key position resulted in 3–5-fold reductions in EV affinities ($K_M \approx 7.1$ mM and 11.4 mM at 30 °C, respectively), indicating the importance of F41/F43 positions for CODH activities toward viologens. These suggest that EV interaction occurs near surface phenylalanine residues (region 41 for _Ch_CODH2) and auxiliary residues (regions 57–59 for _Ch_CODH2) at the D-cluster of CODHs, inferring electron transfer from the D-cluster can proceed

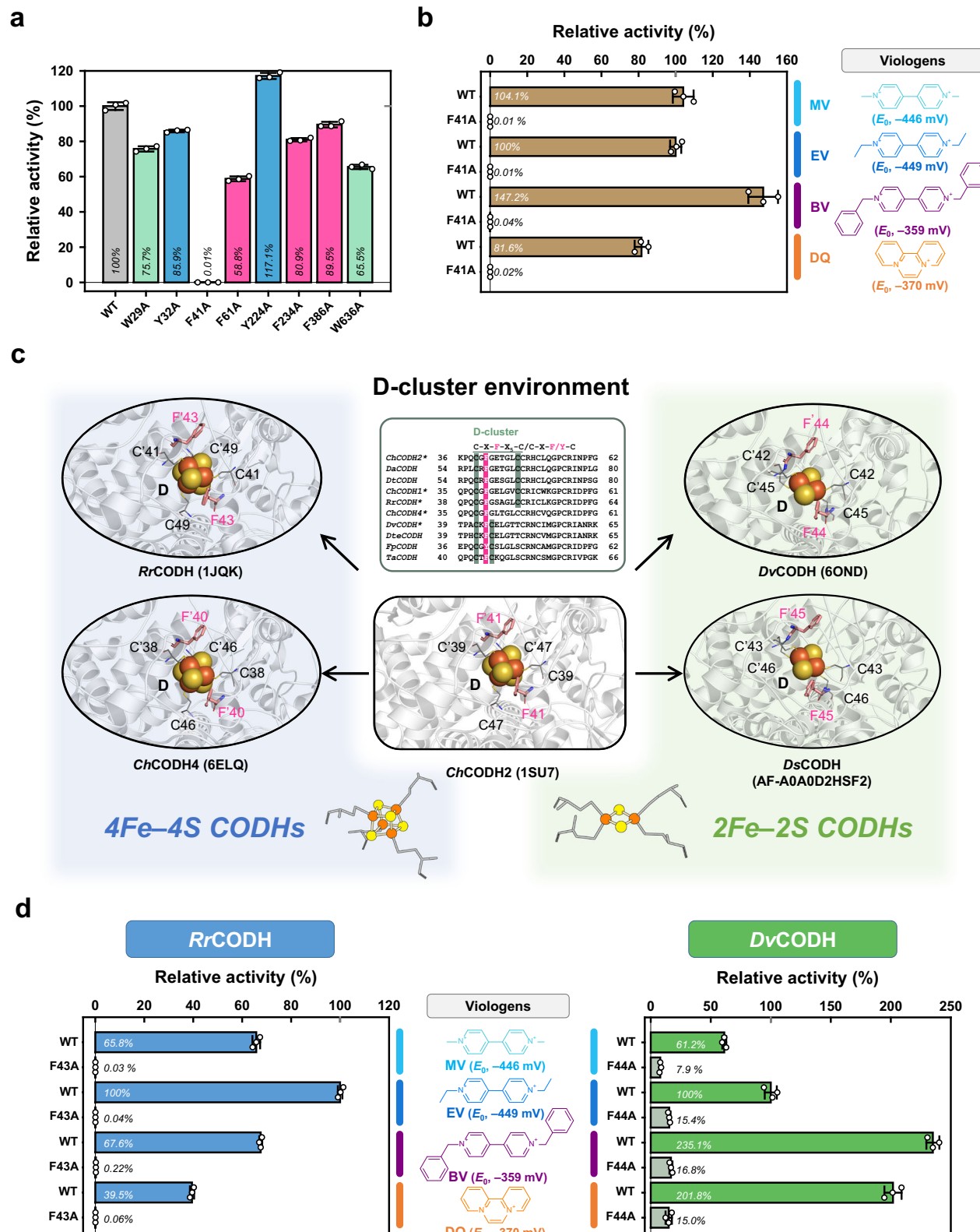

through F41/F43 spots to mediator viologens. Consequently, these findings imply that modifying the D-cluster coordination can affect mediator affinity in *Ch*CODH2.

To determine the binding affinity of WT and R57G/N59L mutants for viologen, we performed isothermal titration calorimetry (ITC) analysis under conditions with substrate CO (Supplementary Fig. 10).

The dissociation constant ($K_D$) values were 449 μM for WT and 144 μM for R57G/N59L, showing that WT has approximately four times lower affinity than R57G/N59L. This corroborates with the lower $K_M$ values observed for R57G/N59L. Despite differences in ITC affinity and $K_M$ values under varying conditions, the trends consistently indicated a notably increase in the affinity of R57G/N59L for EV compared to WT.

**Fig. 2 | EV response of surface aromatic residues in CODHs. a** Comparison of the relative activities of the *Ch*CODH2 variants. Enzyme activities were assayed using 20 mM EV (ethyl viologen) at 30 °C, pH 8. The dashed line indicates the relative activity of WT (wild type). White dots for each point overlay bar charts. The data represent the mean ± standard deviation (S.D.), as determined from *n* = 3 independent experiments. **b** Relative F41 activity for various viologens. The activity of the WT and the F41A mutant towards viologen homologs was observed. The dotted line indicates WT activity toward EV. The F41A mutant showed no activity towards any of the viologen homologs, whereas the WT enzyme showed activity. White dots for each point overlay bar charts. The data represent the mean ± S.D., as determined from *n* = 3 independent experiments. Abbreviation: MV, methyl viologen ($R_1 = R_2 = CH_3$); EV, ethyl viologen ($R_1 = R_2 = CH_2CH_3$); BV, benzyl viologen ($R_1 = R_2 = CH_2C_6H_5$); DQ, diquat. **c** Conserved coordination of D-cluster in CODHs.

The D-cluster environments of both 4Fe4S and 2Fe2S CODHs were analyzed by examining their sequences and structures. While the cysteine residues differ between 4Fe4S and 2Fe2S CODHs, the position of aromatic residues remains unchanged. The PDB IDs are *Ch*CODH2 (1SU7)[14], *Rr*CODH (1JQK)[13], *Ch*CODH4 (6ELQ)[15], *Dv*CODH (6OND)[22], and *Ds*CODH (AF-A0A0D2HSF2) is a predicted structure from the AlphaFold database[48]. **d** Relative activities of *Rr*CODH F43A and *Dv*CODH F44A for various viologens. The standard redox potential ($E_0$), under standard conditions (pH 7 and 25 °C), is derived from the references[55,56]. MV reduction ($MV^{2+} + e^- \rightarrow MV^{+\bullet}$) of –446 mV, EV reduction ($EV^{2+} + e^- \rightarrow EV^{+\bullet}$) of –449 mV, BV reduction ($BV^{2+} + e^- \rightarrow BV^{+\bullet}$) of –359 mV, DQ reduction ($DQ^{2+} + e^- \rightarrow DQ^{+\bullet}$) of –370 mV. White dots for each point overlay bar charts. The data represent the mean ± S.D., as determined from *n* = 3 independent experiments.

## High efficiency of *Ch*CODH2 variant using real waste CO gases

Diffusion-limited enzymes, such as *Ch*CODHs, exhibit high catalytic rates yet low mediator affinities[24]. Enhancing the mediator affinity of the enzyme is a critical factor directly linked to cost-effectiveness in large-scale applications. As depicted in Fig. 3c, the R57G/N59L mutant, with a 10.1-fold higher efficiency ratio than WT, is an appealing candidate for efficient waste CO gas conversion in practical applications. The oxygen sensitivity of CODHs is another critical issue for the biocatalytic conversion of waste gas CO in various fields. Therefore, we have considered introducing the recently engineered $O_2$-tolerant *Ch*CODH2 A559W mutant[5].

Searching tunnel-forming and viologen-interacting residues in the *Ch*CODH2 A559W structure (PDB 7XDM[5]; Supplementary Fig. 11), we found that residues 57 and 59, not directly involved in tunnel formation, are situated distantly from W559 (20.1 Å and 25.9 Å, respectively). Consequently, we concluded that A559W and R57G/N59L mutations would not strongly influence each other. Furthermore, since viologen molecules are larger than tunnel diameters (Supplementary Fig. 1), they cannot access the enzyme's active site C-cluster internally, indicating that enzyme affinity for viologens is primarily dictated by the external D-cluster rather than the catalytic site (C-cluster). Based on these observations, we introduced the combined R57G/N59L/A559W mutation, which maintained high catalytic efficiency toward EV (Table 1) and exhibited high oxygen tolerance (Fig. 3d). These findings suggest that alterations in EV affinity and tunnel modification in CODH are independent phenomena.

Moreover, we extensively evaluate whether the enzymatic reactivity of R57G/N59L/A559W was evident in industrial waste gas mixtures containing CO. We obtained different CO mixtures (Fig. 3e and Supplementary Table 8): three off-gases from the steel process (Hyundai Steel) including coke-oven gas (COG), blast-furnace gas (BFG), Linz–Donawitz converter gas (LDG)[5]; gas from gasification of mixed plastic waste (Korea Institute of Energy Research) including solid refuse fuel (SRF)-derived gas[25,26]. Figure 3f results demonstrated that compared to the low-content CO mixture (5% (*v/v*) CO and 95% (*v/v*) N$_2$), the relative activities of R57G/N59L/A559W towards other CO mixtures: low CO gas (100%), COG (98%), BFG (110%), LDG (112%), and SRF-derived gas (87%). These outcomes suggest that the R57G/N59L/A559W biocatalyst has the potential as a versatile solution for treating a broad spectrum of real waste gases, including those generated from the steel process and mixed plastic waste. These findings pave the way for efficiently tailored biocatalysts in industrial and gas cleaning processes.

## Structural analysis of the *Ch*CODH2 R57G/N59L variant in complex with EV and BV

To uncover the reasons for the enhanced mediator interaction of the R57G/N59L variant, we determined the viologen-free structure of the R57G/N59L variant with 2.1 Å resolution (PDB ID: 8X9D, Fig. 4a). The overall structure of the R57G/N59L variant, including B, C, and D clusters, was highly similar to that of WT (RMSD of 0.240 Å for 604

Cα, Supplementary Fig. 12). However, the local environment around Gly57 was notably altered due to the mutation, leading to the formation of a cavity (cavity volume: 38.45 Å$^3$ for R57G/N59L, no cavity for WT). This structural change, particularly the enlarged cavity, appears to indirectly influence viologen interaction by altering the spatial arrangement around the interaction site. Analysis of the catalytic properties of R57 and N59 variants (Table 1 and Supplementary Table 7) revealed significantly different reaction kinetics for viologen reduction compared to WT. Specifically, these mutations seem to mitigate the repulsive effect between the positively charged electron mediator oxidized EV and Arg57, thereby facilitating easier access for viologen molecules. This structural alteration led to a notable improvement in the $K_M$ for EV in the R57G/N59L mutant, indicating an indirectly positive influence on viologen complex formation, in contrast to the WT (1SU7)[14].

To demonstrate the specific sites where *Ch*CODH2 interacts with electron mediators, we solved two crystal structures of the R57G/N59L variant complexed with EV and BV, respectively (PDB codes 8X9F and 8X9G) (Fig. 4b and Supplementary Table 9, Supplementary Fig. 13). The formation of complex structures appeared to be influenced by high concentrations of polyethylene glycol (PEG, 25% (*v/v*)) in the crystallization solution, potentially interfering viologen access to D clusters. However, when reduced to 5% (*v/v*), PEG's distribution narrowed near D-cluster cavity (Supplementary Fig. 14). We observed additional electron density maps (*Fo-Fc*, 2.5 σ) of each electron mediator near the F41 residue, which were not visible in the viologen-free form. EV and BV were modeled into the electron-density maps with B factors, respectively (EV: 1, 50.75; BV: 1, 74.74 Å$^2$) (Fig. 4b and Supplementary Fig. 13). In the R57G/N59L-viologen complex, viologen binding sites, involving charge and hydrophobic interactions, were near F41, primarily around the E43, T44, L583, and L612, with viologens showing buried surface area of 160.5 Å$^2$ for EV and 230.3 Å$^2$ for BV in PISA analysis[27]. Notably, compared to the WT structure (1SU7)[14], the viologen-CODH complexes show pronounced surface charge alterations at E43, attributed to differences in the side-chain conformation of E43 (Supplementary Fig. 15). The cause of these conformational changes at Glu43 remains unclear. To ascertain the role of E43's negative charge in binding with EV, we measured the $K_M$ changes in the E43K and E43R variants, featuring positively charged residues (Table 1). These variants exhibited a 2.5- to 3.2-fold increase in $K_M$ compared to WT, indicating directly electrostatic interactions between the negatively charged E43 and the positively charged oxidized viologen molecule. This observation suggests that the binding of CODH with viologen is facilitated by electrostatic interactions, particularly between E43 and the positively charged oxidized EV planar ring structure (Fig. 4b). Further, considering the binding site numbers derived from ITC results and the homodimeric nature of CODH, we calculate approximately one binding site per monomer (Supplementary Fig. 10). Therefore, we conclude that the interaction of CODH with viologen molecules is critically dictated by the electrostatic interaction at position 43.

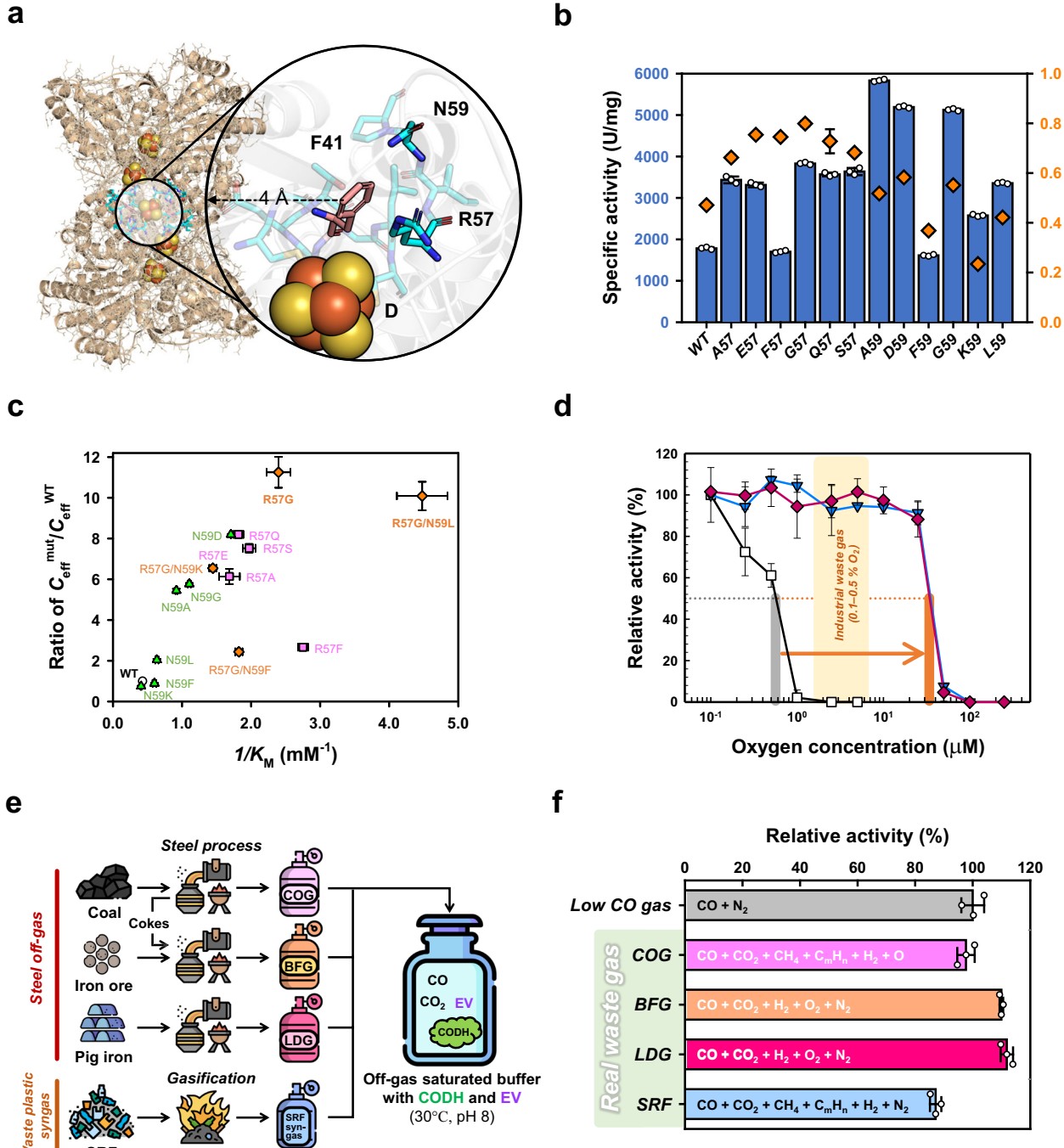

**Fig. 3 | Highly efficient CODH variants. a** Structure of the D-cluster region. Due to its importance for enzyme activity, the F41 residue cannot be targeted for engineering. Instead, neighboring residues R57 and N59 (cyan), located near the F41 residue (pink), have been chosen as potential targets for enhancing binding affinity. The D-cluster of the *Ch*CODH2 enzyme (PDB 1SU7)[14] is depicted as spheres. **b** *Ch*CODH2 variants activity. Specific activities of mutants were measured using 20 mM and 2 mM of EV. Mutations at the position 57 exhibited a higher activity ratio (orange diamonds) at 2 mM than 20 mM. Mutations at the position 59 did not significantly differ from the WT. White dots for each point overlay bar charts. The data represent the mean ± S.D., as determined from $n = 3$ independent experiments. **c** Efficiency ratios *Ch*CODH2 and variants. Michaelis' constant and catalytic efficiencies of mutants were plotted. The data represent the mean ± S.D., as determined from $n = 3$ independent experiments. **d** Increase in mutant $O_2$ sensitivity. The effects of $O_2$ concentration on WT (squares), A559W (reverse triangles), and R57G/N59L/A559W (diamonds) were observed. The grey and pink bars correspond to the half-maximal $O_2$ inhibition concentration of *Ch*CODH2 WT and

R57G/N59L/A559W, respectively. The horizontal arrow indicates the increase in the half-maximal concentration of $O_2$ inhibition from *Ch*CODH2 WT to R57G/N59L/A559W. Values are the means ± standard variation, $n = 3$ (Methods). The relative activities of WT and A559W are from the same published data, reproduced here with permission (reference[5], Copyright 2022, Springer Nature). **e** A schematic for CODH enzyme reactions using industrial off-gases and plastic waste syngas. The R57G/N59L/A559W mutant and EV were used in enzyme reactions with four different gases obtained from real waste gas. COG, coke-oven gas; BFG, blast-furnace gas; LDG, Linz–Donawitz converter gas; SRF, Solid refuse fuel. The icons were made by Freepik from Flaticon (www.flaticon.com). **f** Influence of gas mixtures on *Ch*CODH2 R57G/N59L/A559W. The activity of the mutant was measured using industrial flue gas. The relative activity was compared to the enzyme activity with pure CO as a standard. The constituents of each gas mixture are depicted within the bars. White dots for each point overlay bar charts. Values are the means ± standard variation, $n = 3$.

**Table 1 | Kinetic constants of CODH variants**

| Enzyme | Specific activity[a] (U·mg⁻¹) | $K_M^{EV}$ (mM) | $k_{cat}^{EV}$ (s⁻¹) | $k_{cat}/K_M^{EV}$ (s⁻¹·mM⁻¹) |
|---|---|---|---|---|
| *Wild type* | | | | |
| *Ch*CODH2 | 1,800 ± 29 | 2.4 ± 0.1 | 2,200 ± 30 | 910 ± 15 |
| *Ch*CODH4 | 85 ± 1.2 | 1.3 ± 0.1 | 85 ± 0.1 | 66 ± 2.0 |
| *Rr*CODH | 1,100 ± 15 | 2.4 ± 0.1 | 1,300 ± 12 | 560 ± 17 |
| *To*CODH | 170 ± 1.5 | 2.9 ± 0.3 | 210 ± 16 | 71 ± 2.7 |
| *F41 or F43 variants* | | | | |
| *Ch*CODH2 F41A | 0.3 ± 0.03 | 7.0 ± 0.5 | 0.4 ± 0.01 | 0.06 ± 0.004 |
| *Rr*CODH F43A | 0.5 ± 0.05 | 11.9 ± 0.3 | 0.8 ± 0.01 | 0.06 ± 0.001 |
| *Ch*CODH2 R57/N59 variants | | | | |
| R57G/N59F | 800 ± 31 | 0.2 ± 0.1 | 1,200 ± 40 | 5,300 ± 347 |
| R57G/N59K | 1,500 ± 38 | 0.5 ± 0.1 | 1,600 ± 65 | 3,400 ± 489 |
| R57G/N59L | 2,300 ± 21 | 0.2 ± 0.1 | 2,100 ± 24 | 11,800 ± 452 |
| *Ch*CODH2 A559W variants | | | | |
| A559W | 2,000 ± 13 | 2.0 ± 0.1 | 2,100 ± 39 | 1,000 ± 40 |
| R57G/ N59L/A559W | 2,700 ± 130 | 0.4 ± 0.1 | 2,000 ± 22 | 4,700 ± 47 |
| *Ch*CODH2 viologen-related variants | | | | |
| E43K | 1,000 ± 42 | 5.7 ± 0.1 | 1,600 ± 47 | 300 ± 10 |
| E43R | 300 ± 30 | 7.0 ± 0.4 | 500 ± 47 | 100 ± 2.8 |
| T44A | 2,200 ± 22 | 1.8 ± 0.1 | 2,500 ± 50 | 1,400 ± 23 |
| P60A | 1,500 ± 51 | 1.9 ± 0.2 | 1,500 ± 60 | 750 ± 32 |
| L583A | 1,700 ± 18 | 2.0 ± 0.4 | 1,700 ± 41 | 590 ± 21 |
| L612A | 1,200 ± 72 | 1.7 ± 0.2 | 1,000 ± 97 | 590 ± 33 |

[a] Specific activities were determined at 20 mM ethyl viologen (EV) in HEPES buffer saturated with CO (30 °C, pH 8). One unit (U) of CODH activity was defined as the amount of enzyme required to reduce 1 μmol of $EV_{ox}$ per min at 30 °C and pH 8. Values are the means ± standard variation, n = 3.
* Kinetic data were assayed at 30 °C, pH 8. The kinetic parameters were calculated by fitting the initial rates obtained at six different EV concentrations (0.0625–32 mM) to the nonlinear hyperbolic regression using SigmaPlot 10.0. All enzymatic activities were determined in triplicate (see details in the Methods section).
† The values of $k_{cat}$ were calculated from $V_{max}$ for EV.

Next, we examined the arrangement of viologens (EV and BV), F41, and Fe–S clusters in the R57G/N59L variant to elucidate electron transfer toward electron mediators, considering the distance constraints governing electron transfer efficiency. The distances from viologens to F41 or the Fe–S clusters were calculated (Fig. 4c): F41 9.9 Å, D-cluster 12.6 Å, B′-cluster 17.2 Å, C-cluster 24.8 Å from EV; F41 12.3 Å, D-cluster 15.1 Å, B′-cluster 18.7 Å, C-cluster 24.6 Å from BV, respectively. Furthermore, the distances between the internal Fe–S clusters (B′, C, D) of CODHs range from 12.6 to 14.2 Å, which can be considered within the range of electron transfer[12,28] (Supplementary Fig. 16), and the binding sites of EV and BV are located within the electron transfer-capable distance. Our structures reveal that viologens bind to the specific site of *Ch*CODH2 close to the F41 residue, which might be a key spot in mediator-based electron transfer.

## Discussion

According to Marcus' theory, electron transfer between the enzyme and electron mediator is determined by distance[29,30]. As such, F41, being closer to electron mediators than other clusters (B, C, D), is considered advantageous for electron transfer. Our findings from electrochemical and structural analyses suggest a dual pathway for electron transfer in CODH: both viologen-based F41 and/or the D-cluster, revealing a more intricate electron transfer mechanism. Additionally, our studies revealed the crucial role of E43 near F41 in viologen binding, related to specific electrostatic interactions rather

than random attachment. In conclusion, biochemical, electrochemical and structural data emphasize the notable role of F41 in EV interaction. Ultimately, electrons generated from CO oxidation in the C-cluster of CODH are known to flow through B′ and D. This study observed that F41 plays a significant role in this process (Fig. 4d), either through viologen-mediated electron transfer at F41 and/or D-cluster. It offers a comprehensive understanding of electron flow from the active site of CODH to external mediators and suggests methods for identifying essential amino acids, their positions and structural considerations.

Owing to the complexity and specificity of protein-ligand interactions[12,16,31], identifying an electron transfer doorway in metalloenzymes such as CODH, FDH, and hydrogenase is challenging. This study elucidated the electron mediator interacting site in Ni–Fe CODHs through a rational approach. It allowed for understanding the role of the D-cluster in the interaction between these enzymes and the mediator. Notably, it revealed that surface-exposed aromatic F/Y/W residues are vital in mediator interaction. This knowledge enables the identification of unknown mediator interaction sites (based on the criteria of surface F, Y, W; distance to the cluster; outward-facing side chains) in industrially important CODH and most viologen-interacting metalloenzymes such as FDH or hydrogenase (Supplementary Fig. 17). Our approach offers valuable insights into understanding mediator interactions and electron transfer processes in metalloenzymes, aiding in customizing biocatalysts for efficient chemical production in biochemical and electrochemical systems.

Viologen, not the natural but the ubiquitous electron mediator in many metalloenzymes[31–33], can provide valuable information on our understanding of the enzyme reactivity and electron transfer route. Based on mediator interactions of variants, we can more efficiently design and modify the reactivity to improve productivity. In addition, enzyme and mediator immobilization studies for direct electron transfer are essential in applying site-specific linkage on enzymes.

Industrial waste gas predominantly exists as mixed gases rather than a single gas. Therefore, reactions using gas-utilizing enzymes necessitate linkage with other enzyme reactions. Mediator regeneration is highly sensitive to productivity in multi-enzyme-linked reactions[8]. In the case of CODH, the $K_M$ for viologen is higher than that of FDHs, resulting in a mediator imbalance. An improved enzyme, such as the R57G/N59L mutant, can easily overcome this problem and enable more efficient biochemical production.

In summary, we unveiled vital discoveries: first, a rational approach and structure determination systematically reveals the key electron transfer pathways in Ni–Fe CODHs; and second, we demonstrated that enhanced interaction mutants exhibit highly efficient performance for real waste gases, irrespective of impurity content. Structures of viologen-complexed CODH variants highlighted the pivotal roles of surface phenylalanine residues as external conduits for the D-cluster to/from mediator, providing insight into electron transfer and the flow of electrons from the active site of CODH to external mediators. The engineered variants have potential application in coupled enzyme reactions (for example, $CO_2$ fixation into formate) involving electron transfer and various reaction studies (electrochemical/direct electron transfer systems). We anticipate that this study offers valuable insights into understanding protein-ligand interactions that frequently occur in diverse metalloenzymes.

## Methods
### Cloning, expression, and purification of CODHs
The DNAs of *Ch*CODH2 (Chy_0085), *Ch*CODH4 (Chy_0736), *Dv*CODH (Dvu_2098), and *Rr*CODH (Rru_A1427) were synthesized and confirmed by Macrogen (Korea). The pET28a (Novagen) expression vector containing a His tag was utilized for plasmid modification. Site-directed variants were generated using QuikChange site-directed mutagenesis (Stratagene) with *Pfu* DNA polymerase. The PCR products were treated with *Dpn*I (2 U·μl⁻¹) and incubated for 1 hour (37 °C) to

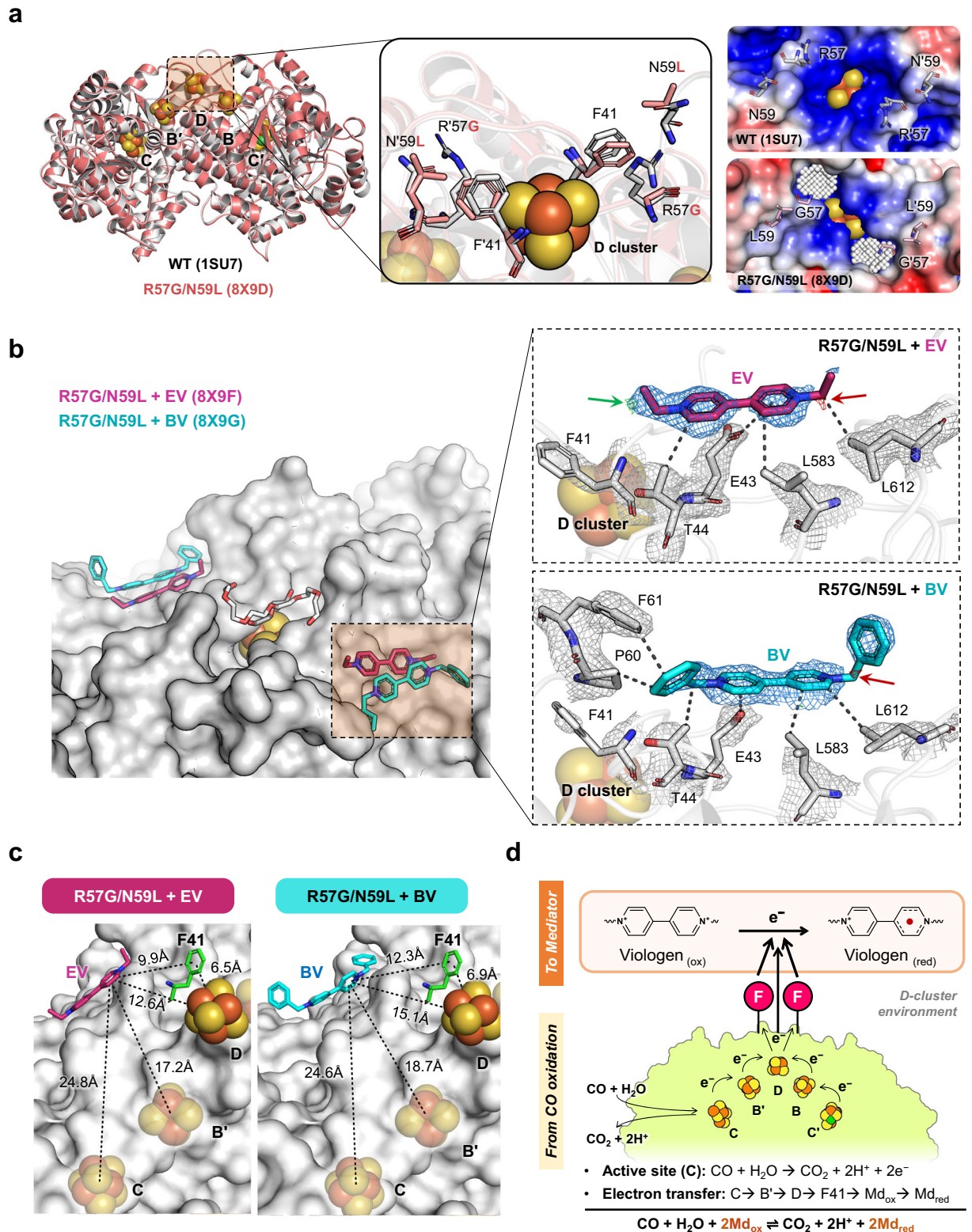

**Fig. 4 | Crystal structures of the *Ch*CODH2 R57G/N59L variants in both apo and viologen-bound forms. a** Structural comparison of the D-cluster region in the crystal structures of *Ch*CODH2 WT (grey, PDB ID: 1SU7)[14] and viologen-free R57G/N59L variant (pink, PDB ID: 8X9D). The surface charges and tunnel cavities near the D cluster of the R57G/N59L variant and the WT are shown on the right. Surface charge was calculated using the APBS and PDB2PQR web server[57] at pH 8, and cavity volumes were quantified through KMfinder's pseudo-atom filling method[58]. **b** Crystal structures of the R57G/N59L variant in complex with EV and BV (PDB ID:

8X9F and 8X9G, respectively). Interacting residues and electron density maps for EV and BV are illustrated, with 2*Fo-Fc* maps at 1 σ (blue) and *Fo-Fc* at 2.5 σ (green and red for +/− level), respectively. Colored arrows indicate EV and BV positions in the *Fo-Fc* maps. The black dashed lines indicate electrostatic and hydrophobic interactions between EV, BV, and surrounding residues. **c** Distances between EV and BV to the cluster C, B', D, and F41 residue. **d** Proposed model of EV interaction with F41 in the CO oxidation of CODH.

remove the residual plasmid. The mutated plasmid sequences were verified by Macrogen DNA sequencing after transforming the *Dpn*I-treated plasmids into *E. coli* DH5α. Confirmed variants were then transformed into *E. coli* BL21(DE3) with pRKISC[34]. For *Rr*CODH transformation, the maturation genes *CooCTJ*[35] were additionally synthesized and incorporated into the pCDFduet vector. We inserted the genes for tungsten-based *Me*FDH1, *fdh1A* (AC: C5ATT7) and *fdh1B* (AC: C5ATT6), into the pCM110 vector and expressed them in *Methylobacterium extorquens* AM1[36]. Similarly, we cloned the molybdenum-based *Rc*FDH genes (*fdhABC*, AC: D5AQH0–2; *fdhE*, AC: D5AQG8) from *Rhodobacter capsulatus*[37,38] into the pTrcHis A vector using *Nhe*I and *Sac*I sites, and expressed them in *E. coli* MC1061. Information regarding primers, plasmids, and strains can be found in Supplementary Table 3.

Cultures were grown aerobically using 1.5% (*v/v*) glycerol, 0.1 mM FeSO4, 2 mM L-cysteine, 0.02 mM NiCl2, kanamycin (50 μg·ml⁻¹), tetracycline (10 μg·ml⁻¹), and/or streptomycin (50 μg·ml⁻¹) in modified TB medium at 37 °C. Upon reaching an $OD_{600}$ of 0.7–0.8, the culture solution was purged with N2 for 30 minutes, and recombinant enzymes were expressed by adding 0.2 mM IPTG (isopropyl β-D–1-thiogalactopyranoside) supplemented with 1 mM FeSO4, 50 mM KNO3, and 0.5 mM NiCl2, followed by an additional 30-minute N2 purge. After incubating for 16-18 hours at 30 °C, cells were collected by centrifugation at 3,220 × *g* for 30 minutes at 4 °C and stored aerobically at –70 °C. For the expression of *Me*FDH1[39], *M. extorquens* AM1 was cultured in defined medium containing 16 g·L⁻¹ succinate and 9.9 mg·L⁻¹ Na2WO4 at 30 °C and 200 rpm. When the $OD_{600}$ reached 0.4–0.6, methanol was added at 0.5% (*v/v*) to induce expression. *Rc*FDH was produced in LB medium with 150 μg·mL⁻¹ ampicillin, 1 mM sodium molybdate, and 20 μM IPTG, at 30 °C and 130 rpm agitation, following established protocols[37,40].

Proteins were purified in an anaerobic chamber under 97.5% N2 / 2.5% H2 conditions. Cells were disrupted by sonication in 50 mM potassium phosphate (pH 8.0), 0.3 M NaCl, and 10 mM imidazole buffer with 2 mM dithioerythritol (DTE) and 2 μM resazurin. Cell suspensions were sonicated for 30 minutes with 2-second pulses at 40% output. Homogenates were centrifuged at 4 °C for 30 minutes at 14,000 × *g* and loaded onto a 0.5 ml Ni-NTA agarose column (Qiagen) in the anaerobic chamber. Columns were washed with 50 mM potassium phosphate (pH 8.0), 20 mM imidazole, and 0.3 M NaCl buffer containing 2 mM DTE and 2 μM resazurin solution. CODH proteins and their variants were then eluted using a 50 mM potassium phosphate (pH 8.0), 250 mM imidazole, and 0.3 M NaCl buffer with 2 mM DTE and 2 μM resazurin. The purified CODH proteins were analyzed via SDS-PAGE on 12% gels, appearing as a single soluble band with an observed size (69 kDa) similar to the calculated size (67 kDa of proteins with a 2 kDa His6-tag). For crystallization, additional purification was conducted through size-exclusion chromatography (HiLoad 16/600 Superdex 200 prep-grade, GE Healthcare Bio-Sciences, PA, USA), previously equilibrated with a buffer containing 20 mM Tris-HCl pH 7.5 and 3 mM dithioerythritol (DTE). Cells expressing FDH were buffered with 50 mM MOPS (3-(*N*-morpholino)propanesulfonic acid) buffer (pH 7.0) with 300 mM NaCl, 2 mM DTE, and 2 μM resazurin. After cell lysis, the clear supernatant from centrifugation (14,000 × *g*, 4 °C, 20 minutes) was incubated with Ni-NTA agarose beads for 15 minutes in an anaerobic chamber. The beads with attached enzyme were washed and eluted for further analysis. Elution buffers, similar to the lysis buffer, contained 20–250 mM imidazole for CODH and 20–300 mM imidazole for FDH, with an extra 10 mM KNO3 for *Rc*FDH purification. Protein concentrations were determined using the Bradford Assay with Bradford Reagent (Sigma).

## Enzyme activity of CODHs and FDHs

CO oxidation activity was evaluated at 30 °C following CO saturation using screw-cap cuvettes with a CO headspace. The activity was assayed by monitoring the CO-dependent reduction of oxidized viologens[41]:

oxidized benzyl viologen ($BV_{ox}$, $\varepsilon_{546} = 9{,}750$ M⁻¹·cm⁻¹), oxidized ethyl viologen ($EV_{ox}$, $\varepsilon_{578} = 10{,}000$ M⁻¹·cm⁻¹), oxidized methyl viologen ($MV_{ox}$, $\varepsilon_{578} = 9{,}700$ M⁻¹ ·cm⁻¹) and oxidized diquat ($DQ_{ox}$, $\varepsilon_{370} = 32{,}000$ M⁻¹·cm⁻¹)[23]. All viologens were available from Sigma-Aldrich: benzyl viologen dichloride (97%, Sigma-Aldrich), ethyl viologen dibromide (99%, Sigma-Aldrich), methyl viologen dichloride hydrate (98%, Sigma-Aldrich), and diquat dibromide monohydrate (98%, Supelco). The reactions were initiated by enzyme injection (0.1–0.2 μg) into a screw-cap cuvette containing 1 ml of reaction buffer (20 mM oxidized viologens, 0–250 μM O2, 2 mM DTE, 50 mM HEPES/NaOH buffer pH 8, saturated with CO from 50 ml of stock solution flushed with CO gas for 1 h)[42]. Measurements were taken up to 250 μM O2, which is close to the atmospheric air concentration of 20% (*v/v*) O2 (~260 μM).

To investigate the effect of O2 inactivation for R57G/N59L/A559W, we measured the residual activity of the enzyme at suitable O2 concentrations. Prior to assaying residual activity, the CODHs (2–4 μg) were incubated in 50 mM HEPES (final volume of 200 μl) without DTE for 2 min with O2 addition, followed by dilution of the reaction mixture 200–400 times to achieve a final O2 concentration below 0.6 μM. One unit of CODH activity was defined as the amount of enzyme required to reduce 1 μmol of $EV_{ox}$ per min at 30 °C and pH 8.

To measure FDH activity, we started the reaction by adding 1 μg of the enzyme to a solution of 200 mM sodium phosphate (pH 7) with 100 mM sodium bicarbonate, 0.1 mM $EV_{red}$, and 2 mM DTE at 30 °C. We monitored the activity by measuring the decrease in absorbance of $EV_{red}$ at the beginning of the reaction. $EV_{red}$ was prepared in an oxygen-free environment by reacting 200 mM $EV_{ox}$ with 1 g of zinc and then filtering the mixture[43]. We repeated each experiment three times.

## Catalytic property

For the assessment of catalytic properties for $EV_{ox}$, a CO-saturated buffer equivalent to 0.91 mM CO (99.998% according to Henry's law) was employed at 30 °C. The headspace of a silicon-septum-sealed cuvette containing 0.0625–32 mM EV and 2 mM DTE in 50 mM HEPES/NaOH buffer (pH 8) was purged with CO for 1 h. To ascertain the initial velocity of CODHs, the enzyme reaction was initiated by injecting 0.1-0.2 μg of Ni-NTA purified CODHs. The absorbance change at 578 nm during the reduction of $EV_{ox}$ was spectrophotometrically monitored at 30 °C within an anoxic glove box. The kinetic parameters ($k_{cat}$ and $K_M$) for $EV_{ox}$ were calculated using the nonlinear hyperbolic regression (Supplementary Fig. 8 and 9). Kinetic data for FDH-mediated $CO_2$ reduction was calculated by varying the concentration of electron mediators (0.006–0.1 mM) under the same conditions used in the activity assays. All enzymatic activities were determined in triplicate.

## Viologen-free electrochemical experiments

Cyclic voltammetry (CV) was used to evaluate the electrochemical performance of *Ch*CODH2 WT and its variants using a SP-150 Biologic potentiostat (BioLogic Science Instruments, France). Measurements were conducted in a custom anaerobic cell (O2 < 2 ppm) equipped with a working electrode coated with CODH enzyme films, a platinum wire counter electrode, and an Ag/AgCl reference electrode[44]. The protocol involved applying 2 μL of a 1:3 CODH to polymyxin mixture onto the electrode to prepare stable enzyme films for electrochemical analysis, similar to methods validated in prior research[45].

On the fluorine-tin oxide (FTO) electrode surface, we applied a mixture of 5 μL of 400 mM carbodiimide and 5 μL of 100 mM *N*-hydroxysulfosuccinimide. After 20 minutes, 10 μL of the *Ch*CODH2 solution (containing 10 μg of protein) was added. The electrode was then left to dry overnight anaerobically inside an anoxic glove box. The electrolyte, a high-concentration 200 mM HEPES/NaOH buffer (pH 8.0), was used to minimize pH changes during reactions and was purged with pure CO gas for at least 1 hour prior to the CO addition reaction. The prepared electrode was placed in a 3 mL cell containing the same buffer, along with 2 mM DTE and 2 μM resazurin. CV was carried out at a scan rate of

10 mV s$^{-1}$, within a potential range of –0.5 V to 0.5 V versus the reversible hydrogen electrode (RHE). All potentials were referenced to the Ag/AgCl electrode and adjusted to the RHE standard using the Nernst equation[44]:

$E$ (V vs. RHE) = $E$ (V vs. Ag/AgCl) + 0.0592 × $pH$ + 0.197. Each test was performed in triplicate to ensure accuracy and reproducibility.

## Gas chromatography analyses

The amounts of gases were quantified using an Agilent 7890B gas chromatograph outfitted with a thermal conductivity detector (GC-TCD) and a carbon molecular sieve column (Carboxen 1000, 60/80 mesh, Supelco). Throughout the GC-TCD analysis, the initial oven temperature was set at 35 °C for 5 min before being progressively raised to 225 °C at 20 °C min$^{-1}$. Injector and detector temperatures were 250 and 225 °C, respectively, while the carrier gas (Ar) flow rate was consistently maintained at 30 ml·min$^{-1}$. To counteract the pH reduction induced by $CO_2$ in synthetic and waste gases, the reaction buffer was finally adjusted to pH 8 using a suitable quantity of 10 M NaOH in an anaerobic chamber. Reactions were conducted in screw-cap cuvettes containing reaction buffer (20 mM EV$_{ox}$, 200 mM HEPES/NaOH buffer pH 8, purged with the low-CO gas (CO, 5% ($v/v$); $N_2$, 95% ($v/v$)) or waste gases for 1 h, and were initiated by enzyme injection (0.1–0.2 µg) at 30°C. Flue gases BFG, COG, and LDG were sourced from Hyundai Steel[5]: BFG (CO, 25.45% ($v/v$); $CO_2$, 23.56% ($v/v$); $H_2$, 3.75% ($v/v$); $O_2$, 0.01% ($v/v$); $N_2$, 46.71% ($v/v$)), COG (CO, 6.15% ($v/v$); $CO_2$, 2.42% ($v/v$); $CH_4$, 24.16% ($v/v$); $C_mH_n$, 3.07% ($v/v$); $H_2$, 55.4% ($v/v$); $O_2$, 0.1% ($v/v$); $N_2$, 8.63% ($v/v$)) and LDG (CO, 53.17% ($v/v$); $CO_2$, 18.51% ($v/v$); $H_2$, 1.43% ($v/v$); $O_2$, 0.11% ($v/v$); $N_2$, 26.77% ($v/v$); unknown, 0.1% ($v/v$)), while the SRF-derived gas was procured from Korea Institute of Energy Research[25,26]: CO, 7.06 % ($v/v$); $CO_2$, 40.57 % ($v/v$); $CH_4$, 3.5 % ($v/v$); $C_mH_n$, 3.05 % (v/v); $H_2$, 2.14 % ($v/v$); $N_2$, 43.67 % ($v/v$).

## Bioinformatics analysis

Bioinformatic searches for amino acid sequence homologs and multiple sequence alignments were conducted using BLAST and ClustalW, respectively. Conserved domains (CDs) and clusters of orthologous groups (COGs) of proteins were analyzed utilizing the CD-Search tool at the National Center for Biotechnology Information (NCBI; http://www.ncbi.nlm.nih.gov). Phylogenetic trees were constructed with MEGA 10 software[46] employing the neighbor-joining method (1000 replicates). Bootstrap values (>90%) are displayed at branch points. Nonredundant reference sequences (1,710 proteins) of Ni–Fe CODHs belonging to COG1151 from eggNOG 5.0 (orthologous database at http://eggnog5.embl.de)[47] were manually inspected for completeness and length. Sequences shorter than 620 and longer than 700 amino acids were excluded from the alignment. The resulting 247 amino acid sequences, including hybrid-cluster proteins such as $Ch$CooS-V, were employed in phylogenetic analysis. The optimal tree with a sum of branch length of 39.86875165 is presented. All ambiguous positions were removed for each sequence pair (pairwise deletion option). The final dataset contained a total of 875 positions. Three-dimensional molecular structures of viologens (MV, CID_15938; EV, CID_14598831; BV, CID_14196) were retrieved from the PubChem database (https://pubchem.ncbi.nlm.nih.gov). The structure of the uncharacterized $Ds$CODH ($Ds$, *Dethiosulfatarculus sandiegensis*; X474_13265, WP_044349155) was predicted using the AlphaFold code AF-A0A0D2H-F1 available on the AlphaFold database (https://alphafold.ebi.ac.uk)[48], and was displayed with metal clusters by superimposing it on the Fe-S clusters of the $Dv$CODH structure (PDB 6OND)[22]. Icons in Fig. 3e were created by Freepik from Flaticon (www.flaticon.com).

## Crystallization, X-ray data collection and structural determination

To solve the crystal structure of viologen-free $Ch$CODH2 R57G/N59L mutant under anaerobic conditions, crystals were grown using the hanging drop method by mixing 2 µL of protein (10 mg·ml$^{-1}$)

contained in gel filtration buffer (20 mM Tris-HCl pH 7.5 and 3 mM DTE) with 2 µL of reservoir solution consisting of 0.1 M HEPES/NaOH pH 7.5, 200 mM MgCl$_2$, and 25% ($w/v$) polyethylene glycol 3350. Protein crystals for data collection were grown over approximately 14 days and then transferred to to a cryoprotection solution comprising the original reservoir solution plus 7.5% ($w/v$) glycerol. Data collection was conducted at 100 K using 0.1 sec exposures and 1° oscillations at Pohang Light Source (PLS)'s 5 C beamlines, with a wavelength set to 1.0000 Å. The diffraction data were processed and scaled using the *HKL2000* software package[49]. The structures were solved by the molecular replacement method using the $Ch$CODH2 WT model (PDB ID: 1SU7)[14] as a probe. Subsequent manual model building was performed using the *Coot*[50] program, and restrained refinement was carried out using *PHENIX* (ver. 1.19.2_4158)[51] and *CCP4* refmac5[52]. Several rounds of model building, simulated annealing, positional refinement, and individual $B$-factor refinement were performed using *Coot* and *PHENIX*. The atomic coordinates and structural factors of viologen-free R57G/N59L mutant at high (25%, $v/v$) and low (5%, $v/v$) PEG concentration were deposited in the Protein Data Bank (PDB codes: 8X9D and 8X9E).

The $Ch$CODH2 R57G/N59L variant in complex with EV and BV was crystallized by incubating the apo crystals of the variant in a reservoir solution with reduced polyethylene glycol 3,350 concentration (from 25% to 5%) and 10 mM EV or BV. This step aimed to eliminate bound polyethylene glycol fragments near EV and BV. The viologen-incubated crystals were transferred to a reservoir solution with 7.5% ($w/v$) glycerol for cryoprotection. Data collection for the viologen-bound protein mirrored the approach used for the viologen-free protein. The structures were solved using the molecular replacement method with the refined R57G/N59L_apo structure (PDB ID: 8X9D) as a probe. Model building and refinement were conducted using the same methods as applied to 8X9D. The atomic coordinates and structure factors were deposited in the Protein Data Bank (PDB code: 8X9F and 8X9G accessible at http://www.rcsb.org).

In the case of F41A, we obtained crystals but poor resolutions of 7–8 Å. Size exclusion chromatography results showed that the overall dimeric structure of F41A (measured molecular mass ≈133.3 kDa) was maintained compared to that of WT (theoretical molecular mass ≈138 kDa) (Supplementary Fig. 18). Crystallization of the F41C mutant followed a similar method as R57G/N59L. The protein in gel filtration buffer (20 mM Tris-HCl pH 7.5 and 3 mM DTE) was mixed with a reservoir solution, and crystals formed over approximately 14 days. Following cryoprotection in a reservoir solution with 10% ($w/v$) glycerol, data collection occurred at 100 K using 0.1 sec exposures and 0.1° oscillations at the BL26B1 beamline in Spring-8 (SP8), with a wavelength set at 1.7364 Å. The structure was resolved using the same calculation methods as for the R57G/N59L mutant and their subsequent structural analysis. Atomic coordinates and structure factors of F41C were deposited in the Protein Data Bank (PDB code: 8X9H accessible at http://www.rcsb.org). Supplementary Table 9 lists the refinement statistics.

For the Fe anomalous difference Fourier map calculation of the R57G/N59L mutant at high PEG concentration (25%, $v/v$) and the F41C mutant, diffraction data were collected at the BL26B1 beamline in Spring-8 (SP8). The anomalous diffraction data, targeting the K-edge of Fe, were collected at a wavelength of 1.7364 Å. These data sets were processed and scaled using the XDS software package[53]. Metal cluster restraints were initially generated with PHENIX[51] and subsequently refined during the process. Anomalous data-collection statistics are detailed in Supplementary Table 10.

## Isothermal titration calorimetry (ITC)

To investigate the binding affinity of CODH and EV, isothermal titration calorimetry (ITC) experiments were conducted anaerobically at 298 K using Affinity ITC instruments (TA Instruments, New Castle, DE, USA). To prevent oxygen interference, a custom acrylic glove box was

utilized, and $O_2$ and $H_2$ levels were monitored with a Coy Anaerobic Monitor (CAM-12, Coy Lab Products) to ensure an oxygen-free environment (refer to Supplementary Fig. 19). The ITC instrument was operated inside this glove box, which was filled with a 95% ($v/v$) $N_2$/5% ($v/v$) $H_2$ atmosphere.

At 298 K, we analyzed 52 μM of purified *Ch*CODH2 WT and R57G/N59L proteins dissolved in a CO-saturated buffer (20 mM Tris/HCl, pH 7.5) containing 1 mM tris(2-carboxyethyl)phosphine[54]. In the 200 μL ITC cell, 2.4 μL of a 2.5 mM $EV_{ox}$ solution was incrementally injected into the protein solution at intervals of 200 seconds with gentle stirring. Fitted curves were produced using GraphPad Prism 7, showing mean ± SEM from two repeated experiments. The ITC thermograms are typical of these duplicates.

## Reporting summary
Further information on research design is available in the Nature Portfolio Reporting Summary linked to this article.

## Data availability
The datasets generated and analyzed during the current study are available from the corresponding author upon request. The kinetic data generated in this study are provided in the Supplementary Information. The atomic coordinates and structure factors for the *Ch*CODH2 R57G/N59L variants and F41C have been deposited in the Protein Data Bank (http://www.rcsb.org) under the accession codes 8X9D, 8X9E, 8X9F, 8X86G, and 8X9H.

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

## Acknowledgements

This research was supported by the C1 Gas Refinery Program and the Engineering Research Center Program through the National Research Foundation of Korea (NRF), funded by the Ministry of Science, ICT & Future Planning (2015M3D3A1A01064919, 2020R1A5A1019631, and 2021R1A2C1004388, respectively). The synchrotron radiation experiments were performed at BL26B1 in SPring-8 with the approval of RIKEN (Proposal No. 2022A2744).

## Author contributions

Y.H.K. and S.M.K. conceived and planned all the experiments. S.M.K. performed the bioinformatic analysis and gene cloning. S.M.K., S.H.K., J.L., J.K., and E.G.P. performed the biochemical characterization, kinetic analysis, and feasibility evaluation, all under the supervision of Y.H.K., S.M.K., and S.H.K. performed their structural analysis. J.R. and H.K. conducted the electrochemical analysis. Y.Y. performed the binding affinity analysis. Y.H.K. and S.M.K. wrote the manuscript. H.H.L., Y.H., H.-J.Y., and S.Y.K. determined the crystal structure. Y.H.K. and H.H.L. reviewed the manuscript.

## Competing interests

The authors declare no competing interests.
