## [Peer Review File · Nature Communications]

Identifying a key spot for electron mediator-interaction to tailor CO dehydrogenase's affinityEditorial Note: Parts of this Peer Review File have been redacted as indicated to maintain the confidentiality of unpublished data.

REVIEWER COMMENTS

Reviewer #1 (Remarks to the Author):

Remarks to the author

In the current report, Kim et al. describe the identification of the electron mediator (viologens) interaction site by kinetic and structural studies using *codh2* from *Carboxydotherrmus hydrogenoformans* as a model enzyme. The authors developed CODHs by improving the catalytic efficiency using a variant that enhances the interaction of the electron mediator in combination with their recently developed O₂-tolerant CODH, opening up broad applications of Ni,Fe-CODHs for industrial exhaust gas treatment. However, the following comments must be addressed before publication in Nature Communications.

Although the authors classified a gene product of *coos-5* (ChCODH5 in this text) from *C. hydrogenoformans* as a Ni-Fe CODH, the gene product, correctly *CooS-V*, is not a Ni-Fe CODH. *CooS-V* has a morphing active site between the FeOS and FeS clusters rather than a NiFeS cluster at the active site and lacks reversible CO/CO₂ conversion activity, as reported by Jeoung et al. ACIE 2022, and shown in the authors' sequence analysis (Supplementary Figure 5 c). Therefore, *CooS-V* cannot use viologen as a redox mediator for CODH activity and the descriptions of ChCODH5 ad Ni-Fe CODH, especially in Figure 2C and Supplementary Table 2, need to be removed.

The authors claimed that the identification of viologen (EV/BV) interaction sites on the ChCODH2 variant R57G/N59L structures and the increased binding affinity of the variant for viologen is due to an enlarged cavity from the mutation. However, no connection between the expanded cavity and the viologen binding site is clearly discussed or shown. There is only a rough description of where they bind with distances to the Fe/S clusters (p. 9-10), but

no description of how the viologen molecules interact with the protein (π -interactions, hydrophobic or hydrophilic, h-bonds...). Therefore, the reviewer requests that a detailed description of the interaction be included in the main text as well as in Figure 4b in conjunction with the mutation effects.

The authors used a linear transformation of the Michaelis-Menten equation to fit the data of the concentration dependence of the initial velocity (the Hanes-Woolf plot in Supplementary Figure 6). However, the linearization methods are historically used to quickly determine the kinetic parameters without much computing power and are less accurate than the nonlinear regression method. For this reason, it is recommended that the kinetic data be fitted to a hyperbola by nonlinear regression and then reported.

In Figure 3d, the relative activities of wt (squares) and A559W (reverse triangles) look pretty much the same as in Figure 2b of the authors' previous publication (Kim et al. NatCatal 2022). If so, clarify and describe the use of the same published data with the publisher's permission.

General comments: Cite proper references when mentioning the PDB IDs in the main and the SI.

Line 3 in p. 1

The word "mediator" is uncertain in the title. Please add "electron" in front of it to make it "electron mediator" in the title.

Line 70 in p. 3

Replace "[NiFe3S4OHx]" with "[NiFe4S4OHx]" .

Line 111 in p. 5

In the panel of putative sites in Ni-Fe CODHs in Fig. 1b, the sites in different codhs are not clearly visible. Since the structure of ChCODH1 is not known, please cite the software you used to model it with an description in Materials and Methods if applicable.

Line 119 and 122 in p. 5

The authors concluded that F41 is important for viologen reduction because the activity of F41A is abolished. However, biogenesis of D cluster may be affected since F41 in CODH2 is very close to D cluster. Please comment on this.

Line 231 in p. 9

Indicate the clusters of B, C, and D in Figure 4a so that they are comparable with the supplemental Figure 9.

Line 235 in p. 9

The authors calculate the tunnel cavity simply by measuring the distance between the carbon atoms of the symmetry molecule near the cavity (Arg for WT and Pro for variant). However, the authors need to calculate and show the cavity in a more precise way. For example, calculate the volume using pseudo-atom filling or at least measure it in two dimensions in the figure 4a.

Line 240 in p. 9

The sentence "The formation of complex structures appeared to be influenced by high concentrations of polyethylene glycol (PEG, 25% (v/v))" is unclear because there is no explanation of how it was influenced. See Line 456 in p. 17.

Line 243 in p. 9

It would be beneficial to quickly visualize the presence of viologen in the structure by showing both 2Fo-Fc and Fo-Fc maps in Figure 4b and describing the occupancy with B-factors for BV/EV. In addition, Fo-Fc maps can be displayed along with +/- contour levels in a supplementary figure.

Line 255 in p. 10

Enter the references for "within the range of electron transfer".

Line 314 in p. 12

In this report, the authors do not provide information on protein-protein interactions, the reviewer recommends removing "protein-protein" from the text.

Line 316 in p. 13

The authors used MeFDH1 and RcFDH in the study of electron transfer with EV. However, the source of two proteins, including production and activity measurement, is not mentioned at all in the Methods, except for cloned genes in Supplementary Table 3b.

Line 325 in p. 13

Replace "Escherichia coli" with "E. coli".

Line 326 in p. 13

Replace "E. coli BL21" with "E. coli BL21(DE3)".

Line 440 in p. 17

Enter the exposure time per oscillation for the data collection.

Line 452 in p. 17

Enter the measured molecular mass of wt as well for comparison.

Line 457 in p. 18

The authors used a lower concentration of PEG to reduce the competition for binding between the viologen and the PEG molecule during crystallization. However, two molecules of PEG are observed and modeled at the twofold crystallographic axis and are shown in Figure 4b without the electron density maps. Based on the comparable size of BV/EV with two molecules of PEG and the relatively low resolution structures, the reviewer is curious whether the authors model BV or EV instead of the PEG molecules and vice versa.

Line 495 in p. 21

Define "U" in the specific activity.

Line 503 in p. 22

there is no "ND" in the table.

Line 517 in p. 23

Figure 2b and 2d: For the comparison of the catalytic activity between different viologens, it is useful to indicate their standard redox potentials.

Supporting Information

Supplementary Table 1

Include recent crystal structure of ChACS/CODH3 from Ruickoldt et al. ACSCatal 2022, where MV was used for the CO-oxidation activity.

There are bold fonts, but no explanation for them.

Supplementary Table 2

In the table title, ... viologen-based CODHs and FDHs. However, there is no description of FDHs in this table.

Supplementary Table 5

Is there any reason why the authors specified only EVox for ChCODH2, since CO oxidation activity requires an oxidized viologen to accept electrons from the reaction?

Supplementary Table 8

- 1) Please enter the energy used to collect the data in the table.
- 2) The redundancy of the data collections is extremely high for the C2 space group, which can cause X-ray damage.
- 3) Modern crystallographic software allows data with lower $I/\sigma(I)$ to be processed using CC1/2 values. The data reported here are better than those scaled and reported by the authors.
- 4) Please enter Viologen (EV/BV) molecules instead of atoms.
- 5) Add data collection statistics for Fe-anomalous (see line 126, pg. 21) to this or new table if necessary.

Supplementary Figure 1 in p. 14

Displaying the VDW of viologens using a mesh can mislead to the experimentally observed

electron density. Please display the VDW as a surface.

Cite CAVER analyst.

Line 67 in p. 15

Replace “ with Fe-S” with “with Fe-S cluster”.

Line 74 in p. 15

Replace “ C344” with “C344A”.

Supplementary Figure 5 in p. 18

Describe the meaning of the asterisks in Figure 5b

Replace “ChCODH5” with “ChCooS-V” in Figure 5b and 5c.

As shown in Supplementary Figure 8, the authors refined the active site C-cluster of the G57/L59 variant with Ni-4Fe-4S-mut2S, which is known as the inactive state and can be reductively reactivated to the Ni-4Fe-4S-OHx state (Feng & Lindahl JACS 2004; Jeoung & Dobbek Science 2007). However, the authors do not provide any experimental evidence or explanation that the observed bridging atom between Ni and Fe1 is the sulfur atom rather than O or Cl atom . Also, is the bond between Fe1 and S5 in the C-cluster real? Display the bond lengths at the clusters.

Reviewer #2 (Remarks to the Author):

CODH catalyzes the reversible oxidation of CO, making it a potentially useful enzyme for waste gas conversion. To perform this task, CODH requires a source or sink for electrons (a redox protein in vivo or small molecule mediator or electrode in vitro). Here authors engineer a CODH to have a lower KM for electron mediators (viologens), but the value of this effort is not clear. Is it at all practical to consider using viologens for large scale gas waste treatment? If so, this point should be discussed. If not, a different motivation for this study should be provided. As written, this manuscript contributes little to our understanding of CODHs and has many serious flaws. In particular, the conclusions are not supported by

the data, the crystallography is very poorly done and not informative, the methods are poorly described, the referencing is poor, and there some big misunderstandings/gaps concerning enzymology and redox chemistry. I do not recommend publication of this manuscript in Nature Communications. If submitted elsewhere, additional experiments need to be done, and the paper should be rewritten to address all enclosed feedback.

Major issues

1. Authors say throughout the paper that their CODH variants have increased “affinity” for the viologens tested, but authors are not measuring affinity. They are measuring K_M , not association constants or dissociation constants. Mediator dyes like ethylviologen do not need to have a discrete binding site to accept electrons from metallocofactors. They need to be able to get close enough ($\sim 12\text{-}\text{\AA}$ to $15\text{-}\text{\AA}$), but that is it. If authors want to claim that the site determined crystallographically is relevant to viologen reduction, they need to mutate the site in such a way that the redox potential of the D-cluster is unaffected and look for a lack of viologen reduction. I think it is highly unlikely that this site is relevant due to the fact that authors had to lower PEG concentrations to get a viologen-bound structure, indicating that viologens are interacting with CODH more weakly than the crystallization solution (even at 10 mM viologen). Proteins have greasy grooves to which small molecules bind. Observing small molecules bound in a crystal structure does not mean that a binding site is relevant. The structures and all discussion of viologen binding should be deleted or validated (binding constants must be measured and binding sites validated).

2. Authors use site-directed mutagenesis to study the effect of residue substitutions on viologen reduction and use the data for statements such as “residue 41 near the D-cluster constitutes the pivotal site for viologen interactions in ChCODH2.” It has long been known that mutating aromatic residues near cofactors can dramatically alter the cofactor’s redox potential (Biochemistry 1996, 35, 50, 15980 for example). If substituting the F41 didn’t affect viologen reduction that would have been an interesting result or if F41 substitution didn’t change the redox potential but did affect viologen reduction anyway, that would be newsworthy. Authors also substitute an arginine residue. Again, it is well known that having a positivity charged residue near a cofactor influences the redox potential. Also, increasing solvent accessible alters the redox potential. Authors made a number of alterations to the

protein that undoubtedly changed the redox potential and thus affected viologen reduction. Studies like this have been done many times, and in those studies, redox potential were measured. This study duplicates past work but is not done as well as what came before.

3. There are also some portions of the manuscript that do not contain proper citations for literature on the structures of CODH, so some citations should be added.

- Page 3, Line 70: A citation from 2011 is used for the statement regarding the constitution of the C-cluster, whereas the first structures of the C-cluster were published in 2001. The citation on line 70 should be replaced with the following citation: PNAS 2001, Drennan et al. and Science 2001, Dobbek et al

- Page 3, Line 73: Citation 5 refers to one structure from 2022, but does not fully support the statement about all of the structural information available showing conservation. The follow citations should be added:

- o PNAS 2001, Drennan et al.

- o JACS 2004, Dobbek et al.

- o Angew Chem 2017 Domnik et al.

- o Elife 2018, Wittenborn et al.

- Page 6, Lines 145-146: PDB entries are mentioned without citing the source publications. Add citations:

- o PNAS 2001, Drennan et al.

- o JBC 2019, Wittenborn et al.

- Figures 1-4 contain renderings of atomic coordinates deposited in the PDB, but the source publications are not cited in the figure legend and/or in a location in the text relevant to the corresponding figure and in the case of Figure 1, the PDBIDs are also not included. The following information should be added:

- o Figure 1: include PDBIDs used (6ELQ, 1SU7, 6OND, 1JQK) and source publications (Angew Chem 2017 Domnik et al., JACS 2004, Dobbek et al., JBC 2019, Wittenborn et al., PNAS 2001, Drennan et al., respectively). ChCODH1: is there a typo meant to be ChCODH5? If not, what is the source?

- o Figure 2: include source publications: Angew Chem 2017 Domnik et al., JACS 2004, Dobbek et al., JBC 2019, Wittenborn et al., PNAS 2001, Drennan et al., Angew Chem 2022 Jeoung et

al.

o Figures 3 and 4: include citation to: JACS 2004, Dobbek et al.

o

4. There are issues with the crystallography.

- Page 9, Line 243 and Supplementary Fig 8b: an 2Fo-Fc omit map should be shown around the viologen and the nearby sidechains. A Fo-Fc map with positive and negative density should also be shown. This binding site and the interactions should be described, and the buried surface area should be measured. Omit maps should also be shown for structures solved without viologens for comparison. Omit maps with positive and negative difference density should be shown in Supp Fig. 8a too. The C-cluster density looks terrible in the PDB validate file (giant balls of negative density), so this figure misleads the reader toward believing that the C-cluster looks better than it is. Anomalous maps are mentioned but not shown. They should be shown.
- Supplementary Table 8: The gap between Rwork and Rfree is quite large, in particular for 8ISG and 8ISH, and is suggestive of overfitting. The authors should return to model building and refinement to obtain more suitable Rwork/Rfree with separation less than 5%. Were the same Rfree flags used as were used for previous structures? It seems that the cell and space groups are the same, which does raise the question as to why molecular replacement was used.
- Methods: in the “crystallization and structural determination” section, it would be useful for the authors to state the time it took to grow crystals suitable for data collection. Also, authors should provide information about the protein solution – it was 10mg/ml but no information is provided about the buffer or pH or salt or glycerol.

Minor comments:

- Authors should use normal nomenclature for residue substitutions. G57/L59 should be R57G/N59L for example.
- Page 6, Line 137: “cubage” should say “cubane”
- Page 6, Line 158: “DvCODH F43A” should say “DvCODH F44A”
- Page 14, why do enzyme assays have 0-250 micromolar O₂?
- Page 15, are activities being reported per monomer or per dimer?

- Supplementary Fig 7: it would be useful to add a key for the tan sticks in the figure caption, presumably they are residues at the tunnel exits.
- Supplementary Fig 6: resolution is poor – it is impossible to read figure.

Reviewer #3 (Remarks to the Author):

Carbon monoxide dehydrogenase is a redox multimetallic enzyme that plays a central role in carbon metabolism in anaerobic microorganisms, reversibly catalyzing the reduction of CO₂ to CO. This oxygen-sensitive enzyme is a homodimer with several FeS clusters: two buried active sites (one per monomer) composed of heteronuclear FeNiS clusters and additional FeS clusters acting as electron relays between the active site and the protein surface: 2 B-clusters (1 per monomer) and 1 intermolecular D-cluster acting as the first electron acceptor from the solvent.

This redox Ni-enzyme is well known to interact with the well described redox mediators, viologens despite a low affinity in the mM range. In this study, Suk Min Kim and collaborators, describe in detail the interaction of several CODH with a series of viologens, using a combined approach of site-directed mutagenesis, enzymology and X-ray crystallography. Unexpectedly, they identified specific interactions between viologens and protein residues, although these redox mediators have been proposed to interact by random collision with redox enzymes.

CODH and other redox enzymes have a great potential for applications in renewable energy and CO₂ valorization. Consequently, a better understanding of enzyme/redox mediators' interactions will help not only the scientific bio-inorganic community but also the biotechnology sector for the development of tailor-made biocatalysts for industrial applications

The paper is well-written, concise, clear and the methodology is well-adapted. The study is well-conducted and the result of a remarkable piece of work. Undoubtedly, the study will be of great interest to readers of Nature communications. I have only minor comments to address, described below:

1. How do the authors explain that in some cases, the mutation led to increased relative activity (e.g. Y224A or C496A)?

2. In the introduction, the authors propose that viologens interact with protein residues via π - π stacking. Although, the X-ray structure of the G57/L59 double mutant revealed the EV/BV binding site in CODH, it clearly shows that there is no direct interaction between EV/BV and F41, as previously suggested. This point should be discussed.

Despite its essential role highlighted by site-directed mutagenesis, its role in electron transfer should be more discussed. With a distance EV-DCluster of 12.6 Å, a direct electron transfer is still possible. In my opinion, both pathways would coexist (via or not F41) and the authors should be less assertive about the role of F41 in electron transfer. In future studies, DFT calculations should be carried out to investigate the electron pathway in CODH in the close environment of D-cluster. A comparison between a series of CODH, the ChCODH-2 WT enzyme and mutants would be highly informative.

3. The authors should use a method without redox-mediator to measure activities (e.g. electrochemically or with an electron-transfer protein such as ferredoxin). It would be interesting to assess the role of F41 in direct electron transfer to an electrode or “pseudo”-physiological partner. It would highlight (or not) the role of F41 in electron transfer whatever the external electron donor/acceptor.

4. Mediator binding site and putative sites in NiFe-CODHs in Figure 1b are barely visible.

5. In figure 2C and Table S2, the authors should remove the structure of ChCODH5 (7B7Q) which is actually not a CODH enzyme.

6. The authors should add the occupancies of Fe, S and Ni atoms and EV and BV in the description of crystal structures.

SUMMARY OF AUTHOR’S RESPONSE

Brief summary of the revision

1) In our revised manuscript and author’s response, we have proactively addressed the shared comments and issues raised by the three reviewers, resulting in significant enhancements to our research interpretation, results, and the manuscript itself.

2) Bullet-point list of key critiques and corresponding revisions in the manuscript

Main issue	Previous version	Revised version
Enhancement of mediator-bound CODH structural data: Improving structural reliability and accuracy of mediator-bound CODH.	 - Protein Structure: ChCODH2 R57G/N59L (PDB_8ISF), EV-bound R57G/N59L (8ISG), BV-bound R57G/N59L (8ISH) - High redundancy in data collection: Redundancy level of 6 for both 8ISF and 8ISG datasets - Modeled structure using NFS (Ni-4Fe-4S-mut2S) approach 	 - Refined protein structures: ChCODH2 R57G/N59L (8X9D), R57G/N59L in 5% PEG (8X9E), EV-bound R57G/N59L (8X9F), BV-bound R57G/N59L (8X9G) - Optimal redundancy: 3.4 (8X9D), 3.6 (8X9F) - XCC modeled structure: improved electron density map - Improved resolution (8X9F, from 2.6 to 2.5Å) and R_{work}/R_{free} (8X9D, 8X9F, 8X9G)
Determination of viologen molecule binding: Assessing whether viologen binding is specific or a result of random attachment.	 - Not provided details on viologen binding 	 - Detailed information on viologen binding sites: E43, T44, P60, L583, L612 - Mutation and kinetics analysis: E43K, E43R, T44A, P60A, L583A, L612A (K_M changes measured) - Surface charge analysis: WT and R57G/N59L via APBS (measured charge changes) - Isothermal titration calorimetry (ITC) analysis: approximate one binding site per monomer in WT and R57G/N59L
Verification beyond calculated K_M Changes: Investigating actual changes in viologen binding affinity to validate the phenomenon of specific binding.	 - Not provided details on viologen affinity 	 - Direct binding affinity measurement by ITC: R57G/N59L shows four times higher ethyl viologen affinity compared to WT - Kinetics analysis: E43K, E43R, T44A, P60A, L583A, L612A (K_M changes measured) - Surface charge analysis: WT and R57G/N59L via APBS (measured charge changes)
Role of F41 in electron transfer pathway: Analyzing the detailed mechanism of F41’s involvement in electron transfer.	 - Role of F41 residue in electron transfer from the D-cluster was not adequately detailed 	 - Electrochemical study: WT, R57G/N59L, and F41 variants (F41A, F41V, F41L, F41C, F41Y) using protein film cyclic voltammetry - Electron transfer pathways: Dual pathway including viologen mediated through F41 residue and/or D-cluster
Impact of F41 residue changes on D-cluster: Examining effects beyond electron transfer, such as other potential influences on the D-cluster due to alterations in the F41 residue.	 - No information provided regarding the influence of F41 residue changes on the D-cluster 	 - Activity changes in aliphatic F41 variants: F41V, F41L - Redox potential measurement: F41 variants - Structural and electron density analysis: F41C structure (8X9H) and D-cluster changes - Dimeric mass confirmation: F41A crystal image and size exclusion chromatography analysis

REVIEWER COMMENTS

Reviewer #1 (Remarks to the Author):

In the current report, Kim *et al.* describe the identification of the electron mediator (viologens) interaction site by kinetic and structural studies using CODH2 from *Carboxydotherrmus hydrogenoformans* as a model enzyme. The authors developed CODHs by improving the catalytic efficiency using a variant that enhances the interaction of the electron mediator in combination with their recently developed O₂-tolerant CODH, opening up broad applications of Ni,Fe-CODHs for industrial exhaust gas treatment. However, the following comments must be addressed before publication in *Nature Communications*.

→ We sincerely appreciate your insightful comments and meticulous suggestions, and the way they have broadened our research perspective and fostered a more rational viewpoint. Our team has thoroughly reviewed and carefully revised our work, focusing on a clearer understanding and interpretation of the research findings. Through earnest and intensive supplementary experiments, this process has improved the overall manuscript, particularly in areas that were previously unclear. We believe that these enhancements will be beneficial in broad fields, including CODH application research.

Q1. Although the authors classified a gene product of *coos-5* (*ChCODH5* in this text) from *C. hydrogenoformans* as a Ni–Fe CODH, the gene product, correctly *CooS-V*, is not a Ni–Fe CODH. *CooS-V* has a morphing active site between the FeOS and FeS clusters rather than a NiFeS cluster at the active site and lacks reversible CO/CO₂ conversion activity, as reported by Jeung *et al.* *ACIE* 2022, and shown in the authors' sequence analysis (Supplementary Figure 5 c). Therefore, *CooS-V* cannot use viologen as a redox mediator for CODH activity and the descriptions of *ChCODH5* as Ni–Fe CODH, especially in Figure 2C and Supplementary Table 2, need to be removed.

→ In response to the reviewer's comment, we acknowledged that the information on *ChCODH5* as presented in Figure 2c, Supplementary Table 2, Supplementary Fig. 6 and 7 was incorrect. We regret not paying closer attention to this detail. As the reviewer correctly identified, *ChCODH5* is not a Ni–Fe CODH but *CooS-V*, which lacks CO/CO₂ conversion activity. To address this, we have removed *CooS-V* and replaced it with the AlphaFold predicted structure of *DsCODH* (*Ds*, *Dethiosulfatarculus sandiegensis*; X474_13265, WP_044349155) using AlphaFold code: AF-A0A0D2HS-F1. This decision was informed by the limited known structures of the 2Fe–2S cluster. The Methods section has been updated to include relevant information and references about the AlphaFold structure and the superimposition of the FeS cluster. Specifically, “The structure of the uncharacterized *DsCODH* was predicted using AlphaFold, and for visualization purposes, the metal clusters were displayed by superimposing it on the Fe–S clusters of the *DvCODH* structure (PDB 6OND) available from the AlphaFold database (<https://alphafold.ebi.ac.uk>)” (Line 486–490).

Figure 2. c, Conserved coordination of D-cluster in CODHs. The D-cluster environments of both 4Fe4S and 2Fe2S CODHs were analyzed by examining their sequences and structures. While the cysteine residues differ between 4Fe4S and 2Fe2S CODHs, the position of aromatic residues remains unchanged. The PDB IDs are *Ch*CODH2 (1SU7)¹⁴, *Rr*CODH (1JQK)¹³, *Ch*CODH4 (6ELQ)¹⁵, *Dv*CODH (6OND)²¹, and *Ds*CODH (AF-A0A0D2HSF2) is a predicted structure from the AlphaFold database⁴⁶.

Supplementary Table 2 | Numbers of surface aromatic residues in viologen-based CODHs

Group	Metal at D-cluster	Organism	Name (PDB code)	No. of surface exposed F/Y/W	No. of residues within 8 Å of Fe-S clusters	No. of putative site
Ni-Fe CODHs	4Fe4S	Carboxydotherrnus hydrogenoformans	Ch CODH2 (1SU7) ²¹	10	3	2
		Carboxydotherrnus hydrogenoformans	Ch CODH4 (6ELQ) ²²	22	3	1
		Rhodospirillum rubrum	Rr CODH (1JQK) ²³	17	5	2
		Moorella thermoacetica	Mr CODH (1MIG) ²⁴	22	3	1
	2Fe2S	Desulfovibrio vulgaris	Dv CODH (6OND) ²⁵	14	5	2

Supplementary Figure 6 | Multiple alignments of *Ch*CODH-related sequences. In the amino acid alignments of *Ch*CODHs, the amino acid residues are displayed in *black shading with white letters* (identical residue, >90% threshold for shading). The characterized CODHs are marked with asterisks (*). The boxes indicate cysteine residues coordinating each Fe-S cluster (B, C, D): C48, C51, C56, and C70 for B-cluster; C294–295, C333, C446, and C476 for C-cluster; C39, C47, C39, and C47 for D-cluster. The CODHs are as follows: *Ch*CODH1 (WP_011344718), *Ch*CODH2 (WP_011343033), *Ch*CODH4 (WP_011343666), *Ch*CooS-V (WP_011342982) from *Carboxydotherrnus hydrogenoformans*; *Ds*CODH from *Desulfovindulus salinum* (WP_121452276); *Rr*CODH from *Rhodospirillum rubrum* (WP_011389181); *Dv*CODH from *Desulfovibrio vulgaris* (WP_010939375); *Dt*CODH from *D. termitidis* (WP_035067836); *Dve*CODH from *Desulfocurvus vexinensis* (WP_028588641); *Dh*CODH from *Desulfovaba hansenii* (WP_100393885).

Supplementary Figure 7 | Coordinating amino acid sequences in proximity to the D-cluster. **a**, Sequence logo representing the frequency in the region 36–63 of *Ch*CODH2 sequences. The sequence logo was generated using WebLogo³⁷. **b**, Partial alignments of CODHs and HCPs (hybrid cluster proteins). The boxes indicate cysteine (orange) and phenylalanine (magenta) residues that coordinate D-cluster. The asterisks denote the CODHs that have been characterized. **c**, Phylogenetic tree of CODHs. The phylogenetic trees showed orthologous relationships based on the amino

acid sequences of the CODH proteins (see Methods for details). The CODHs are as follows: *Ch*CODH1 (WP_011344718), *Ch*CODH2 (WP_011343033), *Ch*CODH4 (WP_011343666) from *Carboxydotherrnus hydrogenoformans*; *Cfe*CODH from *C. ferrireducens* (WP_028051453); *Dsa*CODH from *Desulfofundulus salinum* (WP_121452276); *Dth*CODH from *D. thermocisterus* (WP_084327079); *Rru*CODH from *Rhodospirillum rubrum* (WP_011389181); *Dv*CODH from *Desulfovibrio vulgaris* (WP_010939375); *Dte*CODH from *D. termitidis* (WP_035067836); *Dve*CODH from *Desulfocurvus vexinensis* (WP_028588641); *Dha*CODH from *Desulfofaba hanseni* (WP_100393885); *Dm*CODH from *Desulfobulbus mediterraneus* (WP_028583186); *Df*CODH from *Desulfoplanes formicivorans* (WP_069856808); *Ds*CODH from *Dethiosulfatarculus sandiegensis* (WP_044349155); *Fp*CODH from *Ferroglobus placidus* (WP_083777753); *Ha*CODH from *Halodesulfobivrio aestuarii* (WP_027361784); *Sf*CODH from *Syntrophobacter fumaroxidans* (WP_011699717); *Ta*CODH from *Thermodesulfobium acidiphilum* (WP_108307686); *Dh*CODH from *Desulfacinum hydrothermale* (WP_084057374); *Di*CODH from *Desulfacinum infernum* (WP_073036188); *Dc*CODH from *Desulfosoma caldarium* (WP_123291211); *Me*CODH from *Methanofollis ethanolicus* (WP_067052819); *Mh*CODH from *Methanomethylovorans hollandica* (WP_015324787); *Sa*CODH from *Syntrophorhabdus aromaticivorans* (WP_028894724); *Tn*CODH from *Thermodesulfurhabdus norvegica* (WP_093393560).

Q2. The authors claimed that the identification of viologen (EV/BV) interaction sites on the *Ch*CODH2 variant R57G/N59L structures and the increased binding affinity of the variant for viologen is due to an enlarged cavity from the mutation. However, no connection between the expanded cavity and the viologen binding site is clearly discussed or shown. There is only a rough description of where they bind with distances to the Fe/S clusters (p. 9-10), but no description of how the viologen molecules interact with the protein (pi-interactions, hydrophobic or hydrophilic, h-bonds...). Therefore, the reviewer requests that a detailed description of the interaction be included in the main text as well as in Figure 4b in conjunction with the mutation effects.

→ Responding to the comments from Reviewers #1 and #3, we conducted additional verifications to clarify the relationship between mutations and the binding site in the viologen-CODH complex, leading to the following interpretations:

1. In the structure of the R57G/N59L-viologen complex, the interaction site for viologens was identified near residue F41, specifically E43, T44, P60, L583, and L612 (indicated in Figure 4b). To investigate the influence of E43's negative charge on the binding with positively charged EV, we measured the K_M changes in variants E43K and E43R, which have positively charged residues (Table 1 and Supplementary Fig. 8). The K_M for these variants showed a 2.5- to 3.2-fold increase compared to WT, indicating an electrostatic interaction between the negatively charged E43 and the positively charged oxidized viologen molecule. As seen in the R57G/N59L-viologen complex structure, the binding of CODH and viologen is supposed mainly mediated by electrostatic interactions between the negatively charged Glu at position 43 and the positively charged oxidized EV planar ring structure (Figure 4b).

Figure 4. b, Crystal structures of the R57G/N59L variant in complex with EV and BV (PDB ID: 8X9F and 8X9G, respectively). Interacting residues and electron density maps for EV and BV are illustrated, with $2Fo-Fc$ maps at 1σ (blue) and $Fo-Fc$ at 2.5σ (green and red for +/- level), respectively. Colored arrows indicate EV and BV positions in the $Fo-Fc$ maps.

Table 1. Kinetic constants of CODH variants

Enzyme	Specific activity ^a (U/mg)	K_M^{EV} (mM)	k_{cat}^{EV} (s ⁻¹)	k_{cat}/K_M^{EV} (s ⁻¹ ·mM ⁻¹)
Wild type				
Ch CODH2	1,800 ± 29	2.4 ± 0.1	2,200 ± 30	910 ± 15
Ch CODH4	85 ± 1.2	1.3 ± 0.1	85 ± 0.1	66 ± 2.0
Rr CODH	1,100 ± 15	2.4 ± 0.1	1,300 ± 12	560 ± 17
To CODH	170 ± 1.5	2.9 ± 0.3	210 ± 16	71 ± 2.7
F41 or F43 variants				
Ch CODH2 F41A	0.3 ± 0.03	7.0 ± 0.5	0.4 ± 0.01	0.06 ± 0.004
Rr CODH F43A	0.5 ± 0.05	11.9 ± 0.3	0.8 ± 0.01	0.06 ± 0.001
Ch CODH2 R57/N59 variants				
R57G/N59F	800 ± 31	0.2 ± 0.1	1,200 ± 40	5,300 ± 347
R57G/N59K	1,500 ± 38	0.5 ± 0.1	1,600 ± 65	3,400 ± 489
R57G/N59L	2,300 ± 21	0.2 ± 0.1	2,100 ± 24	11,800 ± 452
Ch CODH2 A559W variants				
A559W	2,000 ± 13	2.0 ± 0.1	2,100 ± 39	1,000 ± 40
R57G/N59L/A559W	2,700 ± 130	0.4 ± 0.1	2,000 ± 22	4,700 ± 47
Ch CODH2 viologen-related variants				
E43K	1,000 ± 42	5.7 ± 0.1	1,600 ± 47	300 ± 10
E43R	300 ± 30	7.0 ± 0.4	500 ± 47	100 ± 2.8
T44A	2,200 ± 22	1.8 ± 0.1	2,500 ± 50	1,400 ± 23
P60A	1,500 ± 51	1.9 ± 0.2	1,500 ± 60	750 ± 32
L583A	1,700 ± 18	2.0 ± 0.4	1,700 ± 41	590 ± 21
L612A	1,200 ± 72	1.7 ± 0.2	1,000 ± 97	590 ± 33

^a Specific activities were determined at 20 mM ethyl viologen (EV) in HEPES buffer saturated with CO (30°C, pH 8). One unit (U) of CODH activity was defined as the amount of enzyme required to reduce 1 μmol of EV_{ox} per min at 30°C and pH 8. Values are the means ± standard variation, *n* = 3.

[†] Kinetic data were assayed at 30°C, pH 8. The kinetic parameters were calculated by fitting the initial rates obtained at six different EV concentrations (0.0625–32 mM) to the nonlinear hyperbolic regression using SigmaPlot 10.0. All enzymatic activities were determined in triplicate (see details in the Methods section).

[‡] The values of k_{cat} were calculated from V_{max} for EV.

f

Supplementary Figure 8 | f, *Ch*CODH2 viologen-related mutants. The values of $^{app}k_{cat}$ were calculated from V_{max} for EV. The data represent the mean ± S.D. determined from *n* = 3 independent experiments.

2. Meanwhile, the R57 and N59 mutations appear to indirectly affect the viologen interaction (Figure 4a). Observing the changes in catalytic properties of these R57 and N59 variants (Table 1 and Supplementary Table 7), it is evident that the reaction kinetics for viologen reduction differ markedly with the substitution of positively charged Arg57, depending on the charge and size of the substituent residues. This suggests that the mutations alleviate or remove the repulsive influence between the positively charged electron mediator oxidized EV and positively charged residue Arg57, thereby easing spatial constraints for viologen approach (Figure 4a). This led to an improvement in the K_M for EV in the R57G/N59L mutant and a difference in the protein surface binding environment, interpreted as an indirectly positive influence on viologen complex formation in R57G/N59L (8X9D) structure compared to WT (1SU7).

Figure 4. a, Structural comparison of the D-cluster region in the crystal structures of *ChCODH2* WT (grey, PDB ID: 1SU7)¹⁴ and viologen-free R57G/N59L variant (pink, PDB ID: 8X9D). The surface charges and tunnel cavities near the D cluster of the R57G/N59L variant and the WT are shown on the right. Surface charge was calculated using the APBS and PDB2PQR web server⁵⁶ at pH 8, and cavity volumes were quantified through KMfinder's pseudo-atom filling method⁵⁷.

Table 1. Kinetic constants of CODH variants

Enzyme	Specific activity ^a (U/mg)	K_M^{EV} (mM)	k_{cat}^{EV} (s ⁻¹)	k_{cat}/K_M^{EV} (s ⁻¹ ·mM ⁻¹)
Wild type				
ChCODH2	1,800 ± 29	2.4 ± 0.1	2,200 ± 30	910 ± 15
ChCODH4	85 ± 1.2	1.3 ± 0.1	85 ± 0.1	66 ± 2.0
RrCODH	1,100 ± 15	2.4 ± 0.1	1,300 ± 12	560 ± 17
ToCODH	170 ± 1.5	2.9 ± 0.3	210 ± 16	71 ± 2.7
F41 or F43 variants				
ChCODH2 F41A	0.3 ± 0.03	7.0 ± 0.5	0.4 ± 0.01	0.06 ± 0.004
RrCODH F43A	0.5 ± 0.05	11.9 ± 0.3	0.8 ± 0.01	0.06 ± 0.001
ChCODH2 R57/N59 variants				
R57G/N59F	800 ± 31	0.2 ± 0.1	1,200 ± 40	5,300 ± 347
R57G/N59K	1,500 ± 38	0.5 ± 0.1	1,600 ± 65	3,400 ± 489
R57G/N59L	2,300 ± 21	0.2 ± 0.1	2,100 ± 24	11,800 ± 452
ChCODH2 A559W variants				
A559W	2,000 ± 13	2.0 ± 0.1	2,100 ± 39	1,000 ± 40
R57G/N59L/A559W	2,700 ± 130	0.4 ± 0.1	2,000 ± 22	4,700 ± 47
ChCODH2 viologen-related variants				
E43K	1,000 ± 42	5.7 ± 0.1	1,600 ± 47	300 ± 10
E43R	300 ± 30	7.0 ± 0.4	500 ± 47	100 ± 2.8
T44A	2,200 ± 22	1.8 ± 0.1	2,500 ± 50	1,400 ± 23
P60A	1,500 ± 51	1.9 ± 0.2	1,500 ± 60	750 ± 32
L583A	1,700 ± 18	2.0 ± 0.4	1,700 ± 41	590 ± 21
L612A	1,200 ± 72	1.7 ± 0.2	1,000 ± 97	590 ± 33

^a Specific activities were determined at 20 mM ethyl viologen (EV) in HEPES buffer saturated with CO (30°C, pH 8). One unit (U) of CODH activity was defined as the amount of enzyme required to reduce 1 μmol of EV_{ox} per min at 30°C and pH 8. Values are the means ± standard variation, *n* = 3.

^b Kinetic data were assayed at 30°C, pH 8. The kinetic parameters were calculated by fitting the initial rates obtained at six different EV concentrations (0.0625–32 mM) to the nonlinear hyperbolic regression using SigmaPlot 10.0. All enzymatic activities were determined in triplicate (see details in the Methods section).

[†] The values of *k*_{cat} were calculated from *V*_{max} for EV.

Supplementary Table 7 | Kinetic constants of *Ch*CODH2 R57 and N59 variants

Enzyme	Specific activity ^a (U/mg)	K _M ^{EV} (mM)	k _{cat} ^{EV} (s ⁻¹)	k _{cat} / K _M ^{EV} (s ⁻¹ ·mM ⁻¹)
Wild type				
Ch CODH2	1,800 ± 29	2.3 ± 0.1	2,200 ± 28	1,000 ± 11
Ch CODH2 R57 variants				
R57A	3,400 ± 82	0.5 ± 0.1	3,400 ± 78	6,700 ± 330
R57E	3,300 ± 56	0.4 ± 0.1	3,400 ± 13	9,000 ± 153
R57F	1,700 ± 28	0.5 ± 0.1	1,800 ± 23	3,500 ± 231
R57G	3,800 ± 29	0.3 ± 0.1	4,200 ± 51	14,700 ± 576
R57Q	3,600 ± 45	0.4 ± 0.1	3,600 ± 27	8,600 ± 130
R57S	3,500 ± 49	0.4 ± 0.1	3,500 ± 86	9,700 ± 621
Ch CODH2 N59 variants				
N59A	5,800 ± 29	1.0 ± 0.1	5,600 ± 18	5,400 ± 165
N59D	5,200 ± 28	0.5 ± 0.1	4,500 ± 4	9,000 ± 82
N59F	1,600 ± 20	1.8 ± 0.2	1,400 ± 74	800 ± 63
N59G	5,100 ± 30	0.8 ± 0.1	4,800 ± 57	6,100 ± 97
N59K	2,600 ± 25	3.3 ± 0.1	1,900 ± 50	590 ± 23
N59L	3,400 ± 19	1.6 ± 0.1	3,100 ± 43	1,900 ± 56

^a Specific activities were determined at 20 mM ethyl viologen (EV) in HEPES buffer saturated with CO (30°C, pH 8). Values are the means ± standard variation, *n* = 3.

^b Kinetic data were assayed at 30°C, pH 8. The kinetic parameters were calculated by fitting the initial rates obtained at six different EV concentrations (0.0625–32 mM) to the nonlinear hyperbolic regression using SigmaPlot 10.0. All enzymatic activities were determined in triplicate (see details in the Methods section).

[†] The values of *k*_{cat} were calculated from *V*_{max} for EV.

Q3. The authors used a linear transformation of the Michaelis–Menten equation to fit the data of the concentration dependence of the initial velocity (the Hanes–Woolf plot in Supplementary Figure 6). However, the linearization methods are historically used to quickly determine the kinetic parameters without much computing power and are less accurate than the nonlinear regression method. For this reason, it is recommended that the kinetic data be fitted to a hyperbola by nonlinear regression and then reported. → As the reviewer's comment, we revised the kinetic data in Supplementary Fig. 8 using nonlinear regression. Additionally, we updated the numerical values in Table 1 and Supplementary Table 7 to reflect this change.

Table 1. Kinetic constants of CODH variants

Enzyme	Specific activity ^a (U/mg)	K _M ^{EV} (mM)	k _{cat} ^{EV} (s ⁻¹)	k _{cat} / K _M ^{EV} (s ⁻¹ ·mM ⁻¹)
Wild type				
Ch CODH2	1,800 ± 29	2.4 ± 0.1	2,200 ± 30	910 ± 15
Ch CODH4	85 ± 1.2	1.3 ± 0.1	85 ± 0.1	66 ± 2.0
Rr CODH	1,100 ± 15	2.4 ± 0.1	1,300 ± 12	560 ± 17
To CODH	170 ± 1.5	2.9 ± 0.3	210 ± 16	71 ± 2.7
F41 or F43 variants				
Ch CODH2 F41A	0.3 ± 0.03	7.0 ± 0.5	0.4 ± 0.01	0.06 ± 0.004
Rr CODH F43A	0.5 ± 0.05	11.9 ± 0.3	0.8 ± 0.01	0.06 ± 0.001
Ch CODH2 R57/N59 variants				
R57G/N59F	800 ± 31	0.2 ± 0.1	1,200 ± 40	5,300 ± 347

R57G/N59K	1,500 ± 38	0.5 ± 0.1	1,600 ± 65	3,400 ± 489
R57G/N59L	2,300 ± 21	0.2 ± 0.1	2,100 ± 24	11,800 ± 452
ChCODH2 A559W variants				
A559W	2,000 ± 13	2.0 ± 0.1	2,100 ± 39	1,000 ± 40
R57G/N59L/A559W	2,700 ± 130	0.4 ± 0.1	2,000 ± 22	4,700 ± 47
ChCODH2 viologen-related variants				
E43K	1,000 ± 42	5.7 ± 0.1	1,600 ± 47	300 ± 10
E43R	300 ± 30	7.0 ± 0.4	500 ± 47	100 ± 2.8
T44A	2,200 ± 22	1.8 ± 0.1	2,500 ± 50	1,400 ± 23
P60A	1,500 ± 51	1.9 ± 0.2	1,500 ± 60	750 ± 32
L583A	1,700 ± 18	2.0 ± 0.4	1,700 ± 41	590 ± 21
L612A	1,200 ± 72	1.7 ± 0.2	1,000 ± 97	590 ± 33

^a Specific activities were determined at 20 mM ethyl viologen (EV) in HEPES buffer saturated with CO (30°C, pH 8). One unit (U) of CODH activity was defined as the amount of enzyme required to reduce 1 μmol of EV_{ox} per min at 30°C and pH 8. Values are the means ± standard variation, *n* = 3.

^b Kinetic data were assayed at 30°C, pH 8. The kinetic parameters were calculated by fitting the initial rates obtained at six different EV concentrations (0.0625–32 mM) to the nonlinear hyperbolic regression using SigmaPlot 10.0. All enzymatic activities were determined in triplicate (see details in the Methods section).

^c The values of *k*_{cat} were calculated from *V*_{max} for EV.

Supplementary Table 7 | Kinetic constants of *Ch*CODH2 R57 and N59 variants

Enzyme	Specific activity ^a (U/mg)	K _M ^{EV} (mM)	k _{cat} ^{EV} (s ⁻¹)	k _{cat} / K _M ^{EV} (s ⁻¹ ·mM ⁻¹)
Wild type				
Ch CODH2	1,800 ± 29	2.3 ± 0.1	2,200 ± 28	1,000 ± 11
ChCODH2 R57 variants				
R57A	3,400 ± 82	0.5 ± 0.1	3,400 ± 78	6,700 ± 330
R57E	3,300 ± 56	0.4 ± 0.1	3,400 ± 13	9,000 ± 153
R57F	1,700 ± 28	0.5 ± 0.1	1,800 ± 23	3,500 ± 231
R57G	3,800 ± 29	0.3 ± 0.1	4,200 ± 51	14,700 ± 576
R57Q	3,600 ± 45	0.4 ± 0.1	3,600 ± 27	8,600 ± 130
R57S	3,500 ± 49	0.4 ± 0.1	3,500 ± 86	9,700 ± 621
ChCODH2 N59 variants				
N59A	5,800 ± 29	1.0 ± 0.1	5,600 ± 18	5,400 ± 165
N59D	5,200 ± 28	0.5 ± 0.1	4,500 ± 4	9,000 ± 82
N59F	1,600 ± 20	1.8 ± 0.2	1,400 ± 74	800 ± 63
N59G	5,100 ± 30	0.8 ± 0.1	4,800 ± 57	6,100 ± 97
N59K	2,600 ± 25	3.3 ± 0.1	1,900 ± 50	590 ± 23
N59L	3,400 ± 19	1.6 ± 0.1	3,100 ± 43	1,900 ± 56

^a Specific activities were determined at 20 mM ethyl viologen (EV) in HEPES buffer saturated with CO (30°C, pH 8). Values are the means ± standard variation, *n* = 3.

^b Kinetic data were assayed at 30°C, pH 8. The kinetic parameters were calculated by fitting the initial rates obtained at six different EV concentrations (0.0625–32 mM) to the nonlinear hyperbolic regression using SigmaPlot 10.0. All enzymatic activities were determined in triplicate (see details in the Methods section).

^c The values of *k*_{cat} were calculated from *V*_{max} for EV.

Supplementary Figure 8 | Nonlinear hyperbolic kinetic profiles. Catalytic properties of *ChCODH2* mutants for EV were estimated from the non-linear regression method. **a**, CODH WTs. **b**, *ChCODH2* F41A and *RrCODH* F43A. **c**, *ChCODH2* R57 mutants. **d**, *ChCODH2* N59 mutants. **e**, *ChCODH2* double and triple mutants. **f**, *ChCODH2* viologen-related mutants. The values of $^{app}k_{cat}$ were calculated from V_{max} for EV. The data represent the mean \pm S.D. determined from $n = 3$ independent experiments.

Q4. In Figure 3d, the relative activities of WT (squares) and A559W (reverse triangles) look pretty much the same as in Figure 2b of the authors' previous publication (Kim *et al. NatCatal* 2022). If so, clarify and describe the use of the same published data with the publisher's permission.

→ In response to the reviewer's query, we used the same data for the relative activities of WT and A559W published in Kim *et al., Nat. Catal., 2022*. As suggested, we have obtained the necessary permissions from Springer Nature to reuse this data. Accordingly, we have included the following statement in our manuscript: "The relative activities of WT and A559W are from the same published data, reproduced here with permission (reference⁵, Copyright 2022, Springer Nature)" (Line 639–641).

Q5. General comments: Cite proper references when mentioning the PDB IDs in the main and the SI.

→ As the reviewer's comment, we included the appropriate references for all Protein Data Bank (PDB) IDs mentioned in both the main text (Line 107, 150, 206, 245, 259, 503, 594, 599, 614–615, 627, 652) and the

Supplementary Information (SI) (Supplementary Table 2; Supplementary Fig. 5, Fig. 10, Fig. 11, and Fig. 15).

Q6. Line 3 in p. 1

The word "mediator" is uncertain in the title. Please add "electron" in front of it to make it "electron mediator" in the title.

→ As the reviewer's suggestion, we edited the title by changing the word "mediator" to "electron mediator" (Line 3).

Q7. Line 70 in p. 3

Replace "[NiFe₃S₄OH_x]" with "[NiFe₄S₄OH_x]".

→ As the reviewer's comment, we replaced "[NiFe₃S₄OH_x]" with "[NiFe₄S₄OH_x]" in the chemical notation of the mentioned cluster (Line 66).

Q8. Line 111 in p. 5

In the panel of putative sites in Ni–Fe CODHs in Fig. 1b, the sites in different CODHs are not clearly visible. Since the structure of *Ch*CODH1 is not known, please cite the software you used to model it with a description in Materials and Methods if applicable.

→ In response to the comment regarding the depiction of enzyme structures in Figure 1b, we have improved visibility by prominently featuring only *Dv*CODH and *Rr*CODH. We initially modeled the *Ch*CODH1 structure using the predicted structure AF-P59934-F1 from the AlphaFold database. However, we have now removed the *Ch*CODH1 structure from the current version of the figure. Additionally, in the Materials and Methods section of our manuscript, we provided a description and citation for the AlphaFold database regarding the predicted structures used (Line 486–490).

b

Figure 1. b, Procedure to identify key sites for interacting with CODH enzymes and viologen. Aromatic residues on the enzyme surface were replaced with alanine residues, likely alanine braille. The influences of mutations on enzymatic activity show relationships between target residues and viologen. This approach could be expanded to other CODHs including *Rr*CODH (PDB 1JQK)¹³ and *Dv*CODH (PDB 6OND)²¹.

Q9. Line 119 and 122 in p. 5

The authors concluded that F41 is important for viologen reduction because the activity of F41A is abolished. However, biogenesis of D cluster may be affected since F41 in CODH2 is very close to D cluster. Please comment on this.

→ As the shared concerns of Reviewers #1 and #3, we examined the F41 mutations' effects on the D cluster using viologen-free electrochemical reactions with protein films. Electrochemical analysis revealed that the F41 variants exhibit cyclic voltammetry (CV) profiles similar to WT (Supplementary Fig. 4). This indicates minimal electron transfer issues in the D cluster of F41 variants and suggests a low likelihood of F41 mutations affecting the cluster's integrity. As detailed in the Methods and Materials section (Line 519–522), despite the low resolution (7–8 Å), we successfully crystallized the soluble F41A variant and confirmed its dimeric form (Supplementary Fig. 17). Furthermore, the D cluster in the newly analyzed F41C structure

(PDB ID 8X9H) was identical to WT (Supplementary Fig. 5). These results indicate that the impact of the F41 mutation on the biogenesis of the D-cluster is believed negligible.

Supplementary Figure 4 | Viologen-free electrochemical reaction of *Ch*CODH2 variants. Cyclic voltammetry (CV) was conducted on fluorine-tin oxide (FTO) electrodes using CODH enzyme films. The black lines represent experiments performed with 100% (*v/v*) CO, while the grey lines indicate experiments carried out without CO. Experimental conditions included a temperature of 25 °C, a 200 mM HEPES/NaOH buffer (pH 8.0), and a scan rate of 10 mV s⁻¹. The observed potential (V vs RHE) in CO oxidation reactions is indicated by an orange dotted line at 0.28 V vs RHE. All tests were conducted in triplicate to verify reproducible CV profiles. Abbreviations: CODH, carbon monoxide dehydrogenase; FTO, fluorine-tin oxide; RHE, reversible hydrogen electrode; *wo*, without.

Supplementary Figure 17 | *Ch*CODH2 F41A mutant crystallization. **a**, Images of F41A mutant crystals with size scale indicated. **b**, Size Exclusion Chromatography (SEC) analysis of F41A mutant. SEC was used to determine the multimeric state of purified *Ch*CODH2 F41A, with the red dotted line denoting the observed molecular mass (133.3 kDa) compared to the theoretical molecular mass (138 kDa) of the dimeric *Ch*CODH2 WT and F41A.

Supplementary Figure 5 | F41C structure and Fe anomalous maps of clusters. **a**, Structural comparison of the D-cluster region in the crystal structures of *Ch*CODH2 WT (grey, PDB ID: 1SU7)²¹ and F41C variant (light blue, PDB ID: 8X9H). **b**, Anomalous difference Fourier maps

illustrating the positions of Fe atoms in B, C, and D clusters contoured at 10, 5, and 10 σ , respectively, are shown in orange mesh. Fe, S, and Ni atoms are coloured orange, yellow and green, respectively. In C cluster, 1SU7²¹ adopts Ni-4Fe-4S-mut2S (NFS), and 1SU7²¹ and 8X9H have Ni-4Fe-4S (XCC) conformations, respectively. The B and C clusters adopt the same conformation in both structures. In F41C, the S2 atom in the C cluster was not observed, and the XCC model, which lacks the μ_2 -S ligand (S2)^{35,36}, was found to be a suitable and accurate model for this conformation.

Q10. Line 231 in p. 9

Indicate the clusters of B, C, and D in Figure 4a so that they are comparable with the supplemental Figure 9.

→ We have revised Figure 4a to include distinct markers for the B, C, and D clusters. This enhancement aligns Figure 4a with Supplementary Fig. 15, allowing for direct comparison.

Figure 4. a, Structural comparison of the D-cluster region in the crystal structures of *ChCODH2* WT (grey, PDB ID: 1SU7)¹⁴ and viologen-free R57G/N59L variant (pink, PDB ID: 8X9D). The surface charges and tunnel cavities near the D cluster of the R57G/N59L variant and the WT are shown on the right. Surface charge was calculated using the APBS and PDB2PQR web server⁵⁴ at pH 8, and cavity volumes were quantified through KMfinder's pseudo-atom filling method⁵⁵.

Q11. Line 235 in p. 9

The authors calculate the tunnel cavity simply by measuring the distance between the carbon atoms of the symmetry molecule near the cavity (Arg for WT and Pro for variant). However, the authors need to calculate and show the cavity in a more precise way. For example, calculate the volume using pseudo-atom filling or at least measure it in two dimensions in the figure 4a.

→ To address the reviewer's concern, we employed KVfinder (*BMC Bioinformatics* **15**: 197, 2014) to calculate the volume of the cavity in the protein structure using pseudo-atom filling (Figure 4a). Our analysis also considered the importance of charge effects in EV interaction along with the cavity size. Consequently, Figure 4a provides illustrations demonstrating the differences in charge and cavity volume.

Figure 4. a, Structural comparison of the D-cluster region in the crystal structures of *Ch*CODH2 WT (grey, PDB ID: 1SU7)¹⁴ and viologen-free R57G/N59L variant (pink, PDB ID: 8X9D). The surface charges and tunnel cavities near the D cluster of the R57G/N59L variant and the WT are shown on the right. Surface charge was calculated using the APBS and PDB2PQR web server⁵⁶ at pH 8, and cavity volumes were quantified through KMfinder's pseudo-atom filling method⁵⁷.

Q12. Line 240 in p. 9

The sentence "The formation of complex structures appeared to be influenced by high concentrations of polyethylene glycol (PEG, 25% (v/v))" is unclear because there is no explanation of how it was influenced. See Line 456 in p. 17.

→ In response to the reviewer's comment, we clarified the influence of polyethylene glycol (PEG) on the formation of the CODH-viologen complex. At a high concentration of 25% (v/v) PEG, PEG is broadly distributed in the cavity of the D cluster and around residue F41 in both WT and mutant structures, as illustrated in Supplementary Fig. 13. At a lower concentration of 5% (v/v), the wide distribution observed at higher PEG concentration becomes relatively narrower, allowing for more effective interaction of E43, a residue near F41, with viologen. The explanation and Supplementary Fig. 13 in the manuscript has been updated: "The formation of complex structures appeared to be influenced by high concentrations of polyethylene glycol (PEG, 25% (v/v)) in the crystallization solution, potentially interfering viologen access to D clusters. However, when reduced to 5% (v/v), PEG's distribution narrowed near D-cluster cavity (Supplementary Fig. 13)" (Line 249–252). This observation led us to conclude that the concentration of PEG significantly affects the formation of the CODH-viologen complex.

Supplementary Figure 13 | PEG Electron Density in R57G/N59L Variants. a, *2Fo-Fc* maps of deca-ethylene glycol in R57G/N59L high PEG structure (PDB: 8X9D), contoured at 1σ (grey mesh). **b**, *2Fo-Fc* maps of tetra-ethylene glycol in R57G/N59L low PEG structure with EV complex (PDB: 8X9F), contoured at 1σ (grey mesh). the access of viologens to CODH.

Q13. Line 243 in p. 9

It would be beneficial to quickly visualize the presence of viologen in the structure by showing both $2Fo-Fc$ and $Fo-Fc$ maps in Figure 4b and describing the occupancy with B-factors for BV/EV. In addition, $Fo-Fc$ maps can be displayed along with \pm contour levels in a supplementary figure.

→ Following the reviewer's suggestion, Figure 4b was revised with viologen structures by showing both $2Fo-Fc$ ($1\ \sigma$) and $Fo-Fc$ ($2.5\ \sigma$) maps with \pm contour levels, as following. In Supplementary Fig. 12, *omit* map ($2.5\ \sigma$) with $Fo-Fc$ (\pm contour) map ($2.5\ \sigma$) is presented. The occupancy and B-factors are 1, 50.71 for EV and 1, 74.74 for BV, respectively, which is described in manuscript (Line 252–256).

Figure 4. b, Crystal structures of the R57G/N59L variant in complex with EV and BV (PDB ID: 8X9F and 8X9G, respectively). Interacting residues and electron density maps for EV and BV are illustrated, with $2Fo-Fc$ maps at $1\ \sigma$ (blue) and $Fo-Fc$ at $2.5\ \sigma$ (green and red for \pm -level), respectively. Colored arrows indicate EV and BV positions in the $Fo-Fc$ maps.

Supplementary Figure 12 | Omit maps of viologens in the R57G/N59L variants. **a,** The *omit* map of EV in the R57G/N59L variant is displayed in grey mesh contoured at $2\ \sigma$. Accompanying this is the $Fo-Fc$ difference electron-density map at $2.5\ \sigma$, with positive and negative levels shown in green and red, respectively, indicated by corresponding colored arrows. Interacting residues with viologens are represented in stick form. **b,** For BV in the R57G/N59L variant, the *omit* map is similarly shown in grey mesh contoured at $2\ \sigma$, along with the $Fo-Fc$ difference electron-density maps at $2.5\ \sigma$ in green and red for positive and negative levels, respectively.

Q14. Line 255 in p. 10

Enter the references for “within the range of electron transfer”.

→ Following the reviewer's suggestion, we included two references in the revised manuscript (Line 280) to support the statement “within the range of electron transfer”: *Nature* **402**, 47–52 (1999). doi.org/10.1038/46972; *Science* **293**, 1281–1285 (2001). doi.org/10.1126/science.1061500.

Q15. Line 314 in p. 12

In this report, the authors do not provide information on protein-protein interactions, the reviewer recommends removing "protein-protein" from the text.

→ As the reviewer's comment, we removed the phrase "protein-protein" from the text (Line 300).

Q16. Line 316 in p. 13

The authors used *MeFDH1* and *RcFDH* in the study of electron transfer with EV. However, the source of two proteins, including production and activity measurement, is not mentioned at all in the Methods, except for cloned genes in Supplementary Table 3b.

→ Following the reviewer's suggestion, we have expanded the Methods section of our manuscript to include detailed descriptions of the cloning, expression, and activity measurement of FDHs (*MeFDH1* and *RcFDH*) (Line 347–351; Line 360–365; Line 380–387; Line 409–414; Line 424–426). These additional details in the Methods section complement the information about cloned genes provided in Supplementary Table 3b.

Q17. Line 325 in p. 13

Replace “*Escherichia coli*” with “*E. coli*”.

→ We replaced “*Escherichia coli*” with “*E. coli*” (Line 344).

Q18. Line 326 in p. 13

Replace “*E. coli* BL21” with “*E. coli* BL21(DE3)”.

→ We replaced “*E. coli* BL21” with “*E. coli* BL21(DE3)” (Line 345).

Q19. Line 440 in p. 17

Enter the exposure time per oscillation for the data collection.

→ As reviewer's suggested, we added the exposure time per oscillation (0.1 second per 1°) for the data collection in method (Line 499–500; 525–527).

Q20. Line 452 in p. 17

Enter the measured molecular mass of WT as well for comparison.

→ As reviewer's suggested, we added the molecular mass of WT (~138 kDa) in method (Line 521).

Q21. Line 457 in p. 18

The authors used a lower concentration of PEG to reduce the competition for binding between the viologen and the PEG molecule during crystallization. However, two molecules of PEG are observed and modeled at the twofold crystallographic axis and are shown in Figure 4b without the electron density maps. Based on the comparable size of BV/EV with two molecules of PEG and the relatively low resolution structures, the reviewer is curious whether the authors model BV or EV instead of the PEG molecules and vice versa.

→ Addressing the reviewer's concern about distinguishing between EV and PEG molecules in our structures, we took specific measures for verification. We first analyzed the crystal structure of the GL Apo form with lowered PEG (5%, v/v) concentration (PDB: 8X9E). This analysis showed no additional *omit* map at the presumed EV site, indicating that the original map indeed represented EV binding. Additionally,

we tested the possibility of mistaking PEG for EV/BV by modeling Tri-ethylene glycol (a PEG derivative) at the EV site and performing refinement. The resulting negative (-) *Fo-Fc* map at 2.5σ suggests that the observed *omit* map in Supplementary Fig. 12 represents EV and BV, not PEG. For clear differentiation, we have included electron density maps (*2Fo-Fc* and *Fo-Fc*) for both PEG and EV/BV in Supplementary Figs. 12 and 13. This approach provides concrete evidence that the identified binding sites are for EV and BV molecules.

Supplementary Figure 13 | PEG Electron Density in R57G/N59L Variants. **a**, *2Fo-Fc* maps of deca-ethylene glycol in R57G/N59L high PEG structure (PDB: 8X9D), contoured at 1σ (grey mesh). **b**, *2Fo-Fc* maps of tetra-ethylene glycol in R57G/N59L low PEG structure with EV complex (PDB: 8X9F), contoured at 1σ (grey mesh). the access of viologens to CODH.

Supplementary Figure 12 | *Omit* maps of viologens in the R57G/N59L variants. **a**, The *omit* map of EV in the R57G/N59L variant is displayed in grey mesh contoured at 2σ . Accompanying this is the *Fo-Fc* difference electron-density map at 2.5σ , with positive and negative levels shown in green and red, respectively, indicated by corresponding colored arrows. Interacting residues with viologens are represented in stick form. **b**, For BV in the R57G/N59L variant, the *omit* map is similarly shown in grey mesh contoured at 2σ , along with the *Fo-Fc* difference electron-density maps at 2.5σ in green and red for positive and negative levels, respectively.

Q22. Line 495 in p. 21

Define “U” in the specific activity.

→ We defined "U" in the context of specific activity within the text. Now, it reads: “One unit (U) of CODH activity was defined as the amount of enzyme required to reduce 1 μmol of EV_{ox} per min at 30°C and pH 8” (Line 585–586).

Q23. Line 503 in p. 22

there is no “ND” in the table.

→ “ND” is deleted in the table (Line 590).

Q24. Line 517 in p. 23

Figure 2b and 2d: For the comparison of the catalytic activity between different viologens, it is useful to indicate their standard redox potentials.

→ In accordance with the reviewer's comment, we have included the standard redox potentials in Figure 2b and 2d, as well as in the figure legends. The standard redox potentials (E_0) are determined under standard conditions (pH 7 and 25°C), sourced from respective references (*Chem. Soc. Rev.* **10**: 49, 1981. doi.org/10.1039/cs9811000049; *Chem. Phys. Lett.* **74**:314, 1980. doi.org/10.1016/0009-2614(80)85166-9). Specifically, the redox potentials are as follows: MV reduction ($MV^{2+} + e^- \rightarrow MV^{+}$) at -446 mV, EV reduction ($EV^{2+} + e^- \rightarrow EV^{+}$) at -449 mV, BV reduction ($BV^{2+} + e^- \rightarrow BV^{+}$) at -359 mV, and DQ reduction ($DQ^{2+} + e^- \rightarrow DQ^{+}$) at -370 mV.

Figure 2. EV response of surface aromatic residues in CODHs. **a**, Comparison of the relative activities of the *ChCODH2* variants. Enzyme activities were assayed using 20 mM EV (ethyl viologen) at 30°C, pH 8. The dotted line indicates the relative activity of WT (wild type). The data represent the mean ± standard deviation (S.D.), as determined from $n = 3$ independent experiments. **b**, Relative F41 activity for various viologens. The activity of the WT and the F41A mutant towards viologen homologs was observed. The dotted line indicates WT activity toward EV. The F41A mutant showed no activity towards any of the viologen homologs, whereas the WT enzyme showed activity. The data represent the mean ± S.D., as determined from $n = 3$ independent experiments. Abbreviation: MV, methyl viologen ($R_1=R_2=CH_3$); EV, ethyl viologen ($R_1=R_2=CH_2CH_3$); BV, benzyl viologen ($R_1=R_2=CH_2C_6H_5$); DQ, diquat. **c**, Conserved coordination of D-cluster in CODHs. The D-cluster environments of both 4Fe4S and 2Fe2S CODHs were analyzed by examining their sequences and structures. While the cysteine residues differ between 4Fe4S and 2Fe2S CODHs, the position of aromatic residues remains unchanged. The PDB IDs are *ChCODH2* (1SU7)¹⁴, *RrCODH* (1JQK)¹³, *ChCODH4* (6ELQ)¹⁵, *DvCODH* (6OND)²¹, and *DsCODH* (AF-A0A0D2HSF2) is a predicted structure from the AlphaFold database⁴⁶. **d**, Relative activities of *RrCODH* F43A and *DvCODH* F44A for various viologens. The standard redox potential (E_0), under standard conditions (pH 7 and 25°C), is derived from the

references^{52,53}. MV reduction ($MV^{2+} + e^- \rightarrow MV^{+}$) of -446 mV, EV reduction ($EV^{2+} + e^- \rightarrow EV^{+}$) of -449 mV, BV reduction ($BV^{2+} + e^- \rightarrow BV^{+}$) of -359 mV, DQ reduction ($DQ^{2+} + e^- \rightarrow DQ^{+}$) of -370 mV. The data represent the mean \pm S.D., as determined from $n = 3$ independent experiments.

Supporting Information

Q25. Supplementary Table 1

Include recent crystal structure of *ChACS/CODH3* from Ruickoldt *et al.* *ACSCatal* 2022, where MV was used for the CO-oxidation activity. There are bold fonts, but no explanation for them.

→ Following the reviewer's suggestion, we have included the recent crystal structure of *ChACS/CODH3* from Ruickoldt *et al.*, *ACS Catal.* 2022, in Supplementary Table 1. Additionally, we have revised the formatting in our manuscript by changing all bold fonts to normal fonts for consistency and clarity in presentation.

Supplementary Table 1 | CO and CO₂ utilizing enzymes

Group	Name (EC number)	Classification	Origin	PDB code	Mediator use	Reference
Carbon monoxide dehydrogenase (CODH)	Ni-Fe CODH (1.2.7.4)	Homodimeric CODH	Carboxydotherrmus hydrogenoformans	Oxygen-sensitive type: 1SU6, 1SU7, 1SU8, 1SUF, 3B51, 3B52, 3B53, 3I39, 5FLE, 2YIV, 4UDX, 4UDY Less oxygen-sensitive type: 6ELQ, 7ERR, 7XDM, 7XDN, 7XDP	BV, EV	This study
			Desulfovibrio vulgaris	6B6V, 6B6W, 6B6X, 6B6Y, 6DC2, 6ONC, 6OND, 6ONS, 6VWY, 6VWZ, 6VX0, 6VX1	MV	Source ¹
			Rhodospirillum rubrum	1IQK	BV, MV	Source ²
			Thermococcus onnurineus NA1	–	BV, EV	This study
		Heteromeric ACS/CODH complex	Thermococcus sp. AM4	6T7J	MV	Source ³
			Carboxydotherrmus hydrogenoformans	7ZKJ, 7ZKK, 7ZKV	MV	Source ⁴
			Clostridium autoethanogenum	6YTT, 6YU9, 6YUA	MV	Source ⁵
			Methanosarcina barkeri	3CF4	MV	Source ⁶
	Moorella thermoacetica	3I01, 3I04, 1MJG, 1OAO, 6X5K, 2Z8Y	MV	Source ⁷		
Formate dehydrogenase (FDH)	Non-metal FDH (1.17.1.9)	Non-metal containing	Arabidopsis thaliana	3JTM, 3N7U, 3NAQ	NAD ⁺	Source ⁸
			Candida boidinii	2FSS, 2J6I, 5DN9, 5DNA, 6D4B, 6D4C	NAD ⁺	Source ⁹
			Chaetomium thermophilum	6T8Y, 6T8Z, 6T92, 6T94	NAD ⁺	Source ¹⁰
			Granulicella mallensis	4XYB, 4XYG, 6T8C, 6T8J, 6T9W, 6T9X, 6TB6	NADP ⁺	Source ¹¹
			Moraxella sp. C-1	2GSD, 3FN4	NAD ⁺	Source ¹²
			Physcomitrium patens	7ARZ		
			Pseudomonas sp. 101	2GO1, 2GUG, 2NAC, 2NAD, 6JUJ, 6JUK, 6JWG, 6JX1	NAD ⁺ , NAD	Source ¹³
		Thiobacillus sp. KNK65MA	3WR5	NAD ⁺	This study	
	Mo FDH (1.17.1.9)	Mo (molybdenum) containing	Cupriavidus necator	6VW7, 6VW8	BV, EV, NAD ⁺	Source ¹⁴ Source ¹⁵
			Escherichia coli	1AA6, 1FDI, 1FDO, 1KQF, 1KQG, 2IV2	BV	Source ¹⁶
			Methanospirillum hungatei	7BKB, 7BKC, 7BKC, 7BKD, 7BKE	BV	Source ¹⁷
			Rhodobacter capsulatus	6TG9, 6TGA	EV	This study
	W FDH (1.17.1.9)	W (tungsten) containing	Desulfovibrio gigas	1H0H	BV, MV	Source ¹⁸
Desulfovibrio vulgaris			6SDR, 6SDV	BV, MV	Source ^{19,20}	
Methanothermobacter wolfeii			5T5I, 5T5M, 5T6I			
Methylobacterium extorquens			7ESZ	BV, EV, NAD ⁺	This study	

Q26. Supplementary Table 2

In the table title, ... viologen-based CODHs and FDHs. However, there is no description of FDHs in this table.

→ In the Supplementary Table 2, we corrected the table title. The term "FDHs" was now removed. The revised title of the table is: "Numbers of surface aromatic residues in viologen-based CODHs".

Q27. Supplementary Table 5

Is there any reason why the authors specified only EV_{ox} for *Ch*CODH2, since CO oxidation activity requires an oxidized viologen to accept electrons from the reaction?

→ We have updated as Supplementary Table 6 to include both oxidized MV (MV_{ox}) and oxidized EV (EV_{ox}). As the reviewer correctly points out, "CO oxidation activity requires an oxidized viologen to accept electrons from the reaction", which is why we used EV_{ox} for measuring CODH activity. To clarify this in our manuscript, we have added the following explanation to the table description: "The measurement of CODH activity employed EV_{ox} because CO oxidation activity necessitates an oxidized viologen to receive electrons from the reaction".

Supplementary Table 6 | K_M for viologens in CODHs and FDHs

Group (EC no.)	Name	Viologens ^a	K _M (mM)	Temp. (°C)/pH	References
CODH (1.2.7.4)	Ch CODH2	EV _{ox}	2.3 ± 0.1	30°C/pH 8	This study
		MV _{ox}	4	20°C/pH 9.5	Source ³⁰
	Ch CODH4	EV _{ox}	1.3 ± 0.1	30°C/pH 8	This study
	Mb CODH	MV _{ox}	7.1 ± 0.01	25°C/pH 7	Source ⁶
	Mt CODH	MV _{ox}	3.0 ± 0.01	50°C/pH 8.4	Source ³¹
FDH (1.17.1.9)	To CODH	EV _{ox}	2.4 ± 0.2	30°C/pH 8	This study
		EV _{red}	0.03 ± 0.002	30°C/pH 7	This study
FDH (1.17.1.9)	Rc FDH	EV _{red}	0.01 ± 0.001	30°C/pH 7	This study
		EV _{red}	0.01 ± 0.001	30°C/pH 7	This study

^a The measurement of CODH activity employed EV_{ox} because CO oxidation activity necessitates an oxidized viologen to receive electrons from the reaction. Values are the means ± standard variation, n = 3.

Abbreviations: CODH, carbon monoxide dehydrogenase; FDH, formate dehydrogenase; ox, oxidized; red, reduced; EV, ethyl viologen; MV, methyl viologen, *Ch*, *Carboxydotherrnus hydrogenoformans*; *Mb*, *Methanosarcina barkeri*, *Me*, *Methylobacterium extorquens*; *Rc*, *Mt*, *Moorella thermoacetica*, *Rh*, *Rhodobacter capsulatus*; *To*, *Thermococcus onnurineus*.

Q28. Supplementary Table 8

1) Please enter the energy used to collect the data in the table.

→ As reviewer's suggested, the energy used for data collection is 12.398 keV and has been added to Supplementary Table 9.

Supplementary Table 9 | Statistics for data collection and refinement

Data set	R57G/N59L apo (PDB ID: 8X9D)	R57G/N59L apo in 5% PEG (PDB ID: 8X9E)	R57G/N59L with EV (PDB ID: 8X9F)	R57G/N59L with BV (PDB ID: 8X9G)	F41C (PDB ID: 8X9H)
A. Data collection					
Energy (keV)	12.398	12.398	12.398	12.398	7.140
Space group	C2	C2	C2	C2	C2
Cell dimensions					
a , b , c (Å)	112.0, 74.6, 71.3	112.1, 75.1, 71.1	112.3, 74.8, 71.2	112.6, 75.4, 72.2	112.1, 75.3, 70.7
α, β, γ (°)	90.0, 111.5, 90.0	90.0, 111.4, 90.0	90.0, 111.2, 90.0	90.0, 111.8, 90.0	90.0, 111.1, 90.0
Resolution range (Å)	50.00 – 2.10 (2.14 – 2.10)*	50.00 – 2.50 (2.54 – 2.50)*	30.00 – 2.50 (2.54 – 2.50)*	50.00 – 3.10 (3.15 – 3.10)*	40.78 – 2.20 (2.21 – 2.20)*
Total/ unique reflections	106,342 / 31,368	71,132 / 18,938	69,338 / 19,005	51,029 / 10,065	112,785 / 27,608
Completeness (%)	99.5 (99.4)*	99.9 (92.8)*	98.0 (98.9)*	98.8 (98.8)*	98.4 (96.8)*
Average I / σ(I)	14.4 (2.4)*	8.7 (1.4)*	8.4 (1.9)*	9.6 (2.6)*	19.07 (13.0)*
CC _{1/2}	98.9 (82.1)*	99.5 (81.3)*	98.6 (80.9)*	97.4 (87.3)*	99.6 (99.1)*
R _{merge} [‡] (%)	8.0 (47.8)*	9.9 (47.4)*	11.4 (43.9)*	14.8 (44.3)*	5.6 (7.8)*
Redundancy	3.4 (3.4)*	3.8 (3.4)*	3.6 (3.7)*	5.1 (4.7)*	4.1 (3.9)*
B. Model refinement statics					
Resolution range (Å)	33.02 – 2.11	35.22 – 2.50	28.05 – 2.48	35.32 – 3.11	37.70 – 2.20
R _{work} / R _{free} [‡] (%)	16.53 / 21.91	17.66 / 22.69	24.46 / 26.73	22.92 / 27.23	14.11 / 19.31
No. atoms					
Protein	4,682	4,661	4,646	4,646	4,648
Cluster atoms	22	22	21	21	22

EV/BV molecules			16	26	
Water	327	119	117	3	355
B -factors (Å ²)					
Protein	28.75	40.82	37.19	56.27	21.82
Cluster atoms	23.1	38.26	36.50	52.53	24.80
EV/BV molecules					
Water	32.64	39.38	35.61	48.27	27.00
R.m.s. deviations					
Bond lengths (Å)	0.009	0.010	0.003	0.003	0.008
Bond angles (°)	1.063	1.145	0.628	0.712	0.979
Ramachandran					
Favored (%)	96.67	96.20	96.99	96.51	97.31
Allowed (%)	3.17	3.65	3.01	3.33	2.38
Outliers (%)	0.16	0.16	0.00	0.16	0.32

* Values in parentheses refer to the highest-resolution shell. Data were collected from one crystal.

[†] $R_{\text{merge}} = \frac{\sum_{hkl} \sum_i |I_i(hkl) - \langle I(hkl) \rangle|}{\sum_{hkl} \sum_i I_i(hkl)}$, where $I(hkl)$ is the intensity of reflection hkl , Σ_{hkl} is the sum over all reflections, and Σ_i is the sum over i measurements of reflection hkl .

[‡] $R = \frac{\sum_{hkl} (|F_{\text{obs}}| - |F_{\text{calc}}|)}{\sum_{hkl} |F_{\text{obs}}|}$, where R_{free} was calculated for a randomly chosen 10% of reflections, which were not used for structural refinement, and R_{work} was calculated for the remaining ones.

2) The redundancy of the data collections is extremely high for the C2 space group, which can cause X-ray damage.

→ Addressing concerns about high data redundancy in the C2 space group and potential X-ray damage, we've rescaled the GL apo (PDB: 8ISF) and GL with EV (PDB: 8ISG) data to lower redundancy from 6.6 to 3.4 and 6.0 to 3.6, respectively. This careful reduction aimed to maintain data completeness and map integrity. However, for the GL structure with BV (redundancy of 5.1), reducing redundancy compromised the data quality, leading us to retain the original data. This decision, albeit keeping redundancy slightly above the ideal 3 to 4 range, was a considered trade-off between minimizing X-ray damage and preserving data quality, crucial for precise structural analysis. Consequently, these results ensure the reliability of our crystal structures and scientific conclusions, despite the challenges of high redundancy in data collection.

3) Modern crystallographic software allows data with lower $I/\sigma I$ to be processed using $CC_{1/2}$ values. The data reported here are better than those scaled and reported by the authors.

→ Thanks for the reviewer's suggestion, we rescaled the data sets by lowering $I/\sigma I$ using $CC_{1/2}$ values and the resolution GL with EV data was improved from 2.6 Å to 2.48 Å. In case of GL apo structure, R_{merge} value was reduced from 10.4 to 8.0.

4) Please enter Viologen (EV/BV) molecules instead of atoms.

→ As suggested, atoms were corrected to viologen molecules.

5) Add data collection statistics for Fe-anomalous (see line 126, pg. 21) to this or new table if necessary.

→ We have collected Fe anomalous data for both GL apo and F41C variants. In accordance with the reviewer's suggestion, we have compiled and added a detailed table of data collection statistics for these Fe anomalous datasets. This information can be found in the newly included Supplementary Table 10.

Supplementary Table 10 | Data collection statistics for Fe anomalous data

Data set	R57G/N59L apo (PDB: 8X9D)	F41C (PDB: 8X9H)
Data collection		
Energy (keV)	7.140	7.140
Space group	C2	C2
Unit-cell length (a , b , c , Å)	112.6, 75.2, 71.6	112.1, 75.3, 70.7
Unit-cell angle (α , β , γ , °)	90.0 111.6, 90.0	90.0, 111.1, 90.0
Resolution range (Å)	50.00 – 2.80 (2.82 – 2.80)*	50.0 – 1.80 (1.81 – 1.80)*
Total / unique reflections	55,136 / 25,928	202,769 / 94,544
Completeness (%)	96.3 (96.6)*	94.3 (90.2)*
Average $I/\sigma(I)$	18.7 (9.6)*	15.4 (8.6)*
$CC_{1/2}$	99.8 (98.9)*	99.5 (98.7)*
$R_{\text{merge}}^{\dagger}$ (%)	3.4 (6.8)*	4.2 (7.2)*
Redundancy	2.1 (2.1)*	2.1 (2.1)*

*Values in parentheses refer to the highest-resolution shell. Data were collected from one crystal.

[†] $R_{\text{merge}} = \frac{\sum_{hkl} \sum_i |I_i(hkl) - \langle I(hkl) \rangle|}{\sum_{hkl} \sum_i I_i(hkl)}$, where $I(hkl)$ is the intensity of reflection hkl , Σ_{hkl} is the sum over all reflections, and Σ_i is the sum over i measurements of reflection hkl .

Q29. Supplementary Figure 1 in p. 14

1) Displaying the VDW of viologens using a mesh can mislead to the experimentally observed electron density. Please display the VDW as a surface.

→ Following the reviewer's suggestion, we have illustrated the van der Waals (VDW) surfaces of viologens in **Supplementary Figure 1**.

Supplementary Figure 1 | Viologens' inability to traverse substrate tunnels due to size constraints. **a**, The van der Waals diameter of viologens (MV, methyl viologen; EV, ethyl viologen; BV, benzyl viologen).

2) Cite CAVER analyst.

→ We have added the reference to the CAVER analysis in the legend of **Supplementary Figure 1**. “The bottleneck diameter of substrate tunnels from catalytic C-cluster in *ChCODH2*. The substrate were tunnels predicted as light blue through CAVER analysis³⁴.”

Q30. Line 67 in p. 15

Replace “with Fe-S” with “with Fe-S cluster”.

→ In response to the reviewer's comment, we replaced “with Fe-S” with “with Fe-S cluster” (**Supplementary Fig. 2 legend**).

Q31. Line 74 in p. 15

Replace “C344” with “C344A”.

→ In response to the reviewer's comment, we replaced “C344” with “C344A” (**Supplementary Fig. 3 legend**).

Q32. Supplementary Figure 5 in p. 18

1) Describe the meaning of the asterisks in Figure 5b

→ In response to the reviewer's request for clarification, we have added a description to **Supplementary Figure 7b**. It now states, “The asterisks denote the CODHs that have been characterized”.

2) Replace “*ChCODH5*” with “*ChCooS-V*” in Figure 5b and 5c.

→ We have updated **Supplementary Figure 7b and 7c** by replacing “*ChCODH5*” with “*ChCooS-V*”.

Supplementary Figure 7 | Coordinating amino acid sequences in proximity to the D-cluster. **b**, Partial alignments of CODHs and HCPs (hybrid cluster proteins). The boxes indicate cysteine (orange) and phenylalanine (magenta) residues that coordinate D-cluster. The asterisks denote the

CODHs that have been characterized. **c**, Phylogenetic tree of CODHs. The phylogenetic trees showed orthologous relationships based on the amino acid sequences of the CODH proteins (see Methods for details).

Q33. As shown in Supplementary Figure 8, the authors refined the active site C-cluster of the G57/L59 variant with Ni-4Fe-4S-mut2S, which is known as the inactive state and can be reductively reactivated to the Ni-4Fe-4S-OHx state (Feng & Lindahl *JACS* 2004; Jeoung & Dobbek *Science* 2007). However, the authors do not provide any experimental evidence or explanation that the observed bridging atom between Ni and Fe1 is the sulfur atom rather than O or Cl atom. Also, is the bond between Fe1 and S5 in the C-cluster real? Display the bond lengths at the clusters.

→ To address the reviewer's concerns, we updated the C cluster model in our structure from the Ni-4Fe-4S-mut2S (NFS) to a more accurate Ni-4Fe-4S (XCC) model. In the R57G/N59L structure, the absence of the S2 atom in the C cluster was noted. The XCC model, lacking the μ_2 -S ligand (S2), provided a more suitable representation for this conformation. This change resulted in a better match with the electron density map and reduced negative *Fo-Fc* discrepancies. The revised model is clearly demonstrated in the *Fo-Fc omit* map at 3.0 σ , providing a more accurate representation of the C cluster. Details of these improvements are presented in Supplementary Fig. 11.

Supplementary Figure 11 | *Fo-Fc* difference electron density maps and Fe anomalous maps of clusters in R57G/N59L variant. The *Fo-Fc* difference electron-density maps of B, C, and D clusters in R57G/N59L variant are shown in grey mesh contoured at 3 σ . Fe, S, and Ni atoms are coloured orange, yellow and green, respectively. In the C cluster, 1SU7²¹ adopts the Ni-4Fe-4S-mut2S (NFS) conformation, while both 1SUF²¹ and 8X9H feature the Ni-4Fe-4S (XCC) conformation. The B and C clusters adopt the same conformations in both structures. In R57G/N59L, the S2 atom in the C cluster was not observed, and the XCC model, which lacks the μ_2 -S ligand (S2)^{35, 36}, was found to be a more suitable and accurate model for this conformation. Bond lengths of clusters in R57G/N59L variant are presented on each bond. Anomalous difference Fourier maps illustrating the positions of Fe atoms in clusters are shown as orange mesh, contoured at 10, 5, and 10 σ in each cluster.

Reviewer #2 (Remarks to the Author):

CODH catalyzes the reversible oxidation of CO, making it a potentially useful enzyme for waste gas conversion. To perform this task, CODH requires a source or sink for electrons (a redox protein *in vivo* or small molecule mediator or electrode *in vitro*). Here authors engineer a CODH to have a lower K_M for electron mediators (viologens), but the value of this effort is not clear. Is it at all practical to consider using viologens for large scale gas waste treatment? If so, this point should be discussed. If not, a different motivation for this study should be provided.

→ In response to the reviewer's remarks, we try to elaborate on the practical application of our study in the context of large-scale waste gas treatment. Recently, we successfully demonstrated molar-scale formate production using a common electron mediator, viologen, in an artificial enzymatic combination of CODH and FDH. This was achieved by directly feeding the real industrial off-gas (from Hyundai steel mill, South Korea) to a 10 L reactor containing immobilized CODH and FDH, as detailed in our recent work (Molar-scale formate production via enzymatic hydration of industrial off-gases, 2023 *Res. Sq.* doi.org/10.21203/rs.3.rs-3137085/v1, currently under revision at *Nature Chemical Engineering*).

Biological reactions derived from natural systems, such as natural microbial/enzyme reactions, face limitations due to their too-specific electron transfer carriers, O₂ sensitivity, and challenges in mass enzyme production (varying metal contents, incorporation of selenocysteine, low activity, and expression problems in industrial strains). The use of too specific electron carrier proteins complicates the combination of heterologous enzymes with varied performances. Hence, for efficient utilization of industrial off-gases, a system allowing free electron exchange through artificial enzymatic combinations from different microbial sources is needed. Research to improve the K_M differences for viologens among these enzymes is a crucial aspect of this (Supplementary Table 6). Currently, we have also confirmed molar scale formate production in a 100-L scale field reaction. Therefore, we believe that enzyme reactions based on viologens have practical potential in waste gas treatment if appropriate separation process is adapted for produced formate and positively charged EV mediator. We plan to publish these research findings soon in another paper and have supplemented the manuscript with the related references as per the reviewer's suggestion (Line 54, 73, 320).

[FIGURE AND FIGURE LEGEND REDACTED]

[FIGURE AND FIGURE LEGEND REDACTED]

Supplementary Table 6 | K_M for viologens in CODHs and FDHs

Group (EC no.)	Name	Viologens*	K_m (mM)	Temp. (°C)/pH	References
CODH (1.2.7.4)	Ch CODH2	EV _{ox}	2.3 ± 0.1	30°C/pH 8	This study
		MV _{ox}	4	20°C/pH 9.5	Source ³⁰
	Ch CODH4	EV _{ox}	1.3 ± 0.1	30°C/pH 8	This study
		Mb CODH	MV _{ox}	7.1 ± 0.01	25°C/pH 7
	Mt CODH	MV _{ox}	3.0 ± 0.01	50°C/pH 8.4	Source ³¹
FDH (1.17.1.9)	To CODH	EV _{ox}	2.4 ± 0.2	30°C/pH 8	This study
		EV _{red}	0.03 ± 0.002	30°C/pH 7	This study
	Me FDH1	EV _{red}	0.01 ± 0.001	30°C/pH 7	This study
		Rc FDH	EV _{red}	0.01 ± 0.001	30°C/pH 7
	Me FDH1	EV _{red}	0.03 ± 0.002	30°C/pH 7	This study
		Rc FDH	EV _{red}	0.01 ± 0.001	30°C/pH 7

* The measurement of CODH activity employed EV_{ox} because CO oxidation activity necessitates an oxidized viologen to receive electrons from the reaction. Values are the means ± standard variation, $n = 3$.

Abbreviations: CODH, carbon monoxide dehydrogenase; FDH, formate dehydrogenase; ox, oxidized; red, reduced; EV, ethyl viologen; MV, methyl viologen, *Ch*, *Carboxydotherrnus hydrogenoformans*; *Mb*, *Methanosarcina barkeri*, *Me*, *Methylobacterium extorquens*; *Rc*, *Mt*, *Moorella thermoacetica*, *Rhodobacter capsulatus*; *To*, *Thermococcus onnurineus*.

As written, this manuscript contributes little to our understanding of CODHs and has many serious flaws. In particular, the conclusions are not supported by the data, the crystallography is very poorly done and not informative, the methods are poorly described, the referencing is poor, and there are some big misunderstandings/gaps concerning enzymology and redox chemistry. I do not recommend publication of this manuscript in *Nature Communications*. If submitted elsewhere, additional experiments need to be done, and the paper should be rewritten to address all enclosed feedback.

→ In response to the reviewer's critical feedback, we have taken these comments as constructive challenges. Careful additional analyses have been conducted to address the issues raised by the reviewer, and these efforts are now incorporated into the revised manuscript. Our aim is to significantly improve its quality, ensuring clearer interpretation and more in-depth analysis. We believe these revisions have adequately addressed the concerns related to the support of our conclusions by the experimental data, the clarity of our methods, the adequacy of referencing, and our understanding of enzymology and redox chemistry.

Major issues

Q1. Authors say throughout the paper that their CODH variants have increased “affinity” for the viologens tested, but authors are not measuring affinity. They are measuring K_M , not association constants or dissociation constants. Mediator dyes like ethyl viologen do not need to have a discrete binding site to accept electrons from metallocofactors. They need to be able to get close enough (~12-Å to 15-Å), but that is it. If authors want to claim that the site determined crystallographically is relevant to viologen reduction, they need to mutate the site in such a way that the redox potential of the D-cluster is unaffected and look for a lack of viologen reduction. I think it is highly unlikely that this site is relevant due to the fact that authors had to lower PEG concentrations to get a viologen-bound structure, indicating that viologens are interacting with CODH more weakly than the crystallization solution (even at 10 mM viologen). Proteins have greasy grooves to which small molecules bind. Observing small molecules bound in a crystal structure does not mean that a binding site is relevant. The structures and all discussion of viologen binding should be deleted or validated (binding constants must be measured and binding sites validated).

→ In response to the points and concerns raised by the reviewer, we have conducted verification and supplementary experiments, and the findings are as follows:

1. In the crystallographically determined structure of the R57G/N59L variant, E43 appears to show specific binding through electrostatic interaction (Fig. 4b). When E43 was replaced with positively charged residues (E43K and E43R), the K_M for EV showed a significant increase in E43 variants, suggesting a close relation to charge effects (Table 1 and Supplementary Fig. 8). This specific response of certain residues to viologen (with K_M) suggests that it's not just a random attachment in “greasy grooves”. If it were merely random binding, K_M would likely not be measurable. These mutant results indicate that the current R57G/N59L-viologen complex involves specific interaction between the negative charge of E43 and the positive charge of oxidized ethyl viologen (Fig. 4b and Table 1).

Figure 4. b, Crystal structures of the R57G/N59L variant in complex with EV and BV (PDB ID: 8X9F and 8X9G, respectively). Interacting residues and electron density maps for EV and BV are illustrated, with $2Fo-Fc$ maps at 1σ (blue) and $Fo-Fc$ at 2.5σ (green and red for +/- level), respectively. Colored arrows indicate EV and BV positions in the $Fo-Fc$ maps.

Table 1. Kinetic constants of CODH variants

Enzyme	Specific activity ^a (U/mg)	K_M^{EV} (mM)	k_{cat}^{EV} (s ⁻¹)	k_{cat}/K_M^{EV} (s ⁻¹ ·mM ⁻¹)
Wild type				
Ch CODH2	1,800 ± 29	2.4 ± 0.1	2,200 ± 30	910 ± 15
Ch CODH4	85 ± 1.2	1.3 ± 0.1	85 ± 0.1	66 ± 2.0
Rr CODH	1,100 ± 15	2.4 ± 0.1	1,300 ± 12	560 ± 17
To CODH	170 ± 1.5	2.9 ± 0.3	210 ± 16	71 ± 2.7
F41 or F43 variants				
Ch CODH2 F41A	0.3 ± 0.03	7.0 ± 0.5	0.4 ± 0.01	0.06 ± 0.004
Rr CODH F43A	0.5 ± 0.05	11.9 ± 0.3	0.8 ± 0.01	0.06 ± 0.001
Ch CODH2 R57/N59 variants				
R57G/N59F	800 ± 31	0.2 ± 0.1	1,200 ± 40	5,300 ± 347
R57G/N59K	1,500 ± 38	0.5 ± 0.1	1,600 ± 65	3,400 ± 489
R57G/N59L	2,300 ± 21	0.2 ± 0.1	2,100 ± 24	11,800 ± 452
Ch CODH2 A559W variants				
A559W	2,000 ± 13	2.0 ± 0.1	2,100 ± 39	1,000 ± 40
R57G/N59L/A559W	2,700 ± 130	0.4 ± 0.1	2,000 ± 22	4,700 ± 47
Ch CODH2 viologen-related variants				
E43K	1,000 ± 42	5.7 ± 0.1	1,600 ± 47	300 ± 10
E43R	300 ± 30	7.0 ± 0.4	500 ± 47	100 ± 2.8
T44A	2,200 ± 22	1.8 ± 0.1	2,500 ± 50	1,400 ± 23
P60A	1,500 ± 51	1.9 ± 0.2	1,500 ± 60	750 ± 32
L583A	1,700 ± 18	2.0 ± 0.4	1,700 ± 41	590 ± 21
L612A	1,200 ± 72	1.7 ± 0.2	1,000 ± 97	590 ± 33

^a Specific activities were determined at 20 mM ethyl viologen (EV) in HEPES buffer saturated with CO (30°C, pH 8). One unit (U) of CODH activity was defined as the amount of enzyme required to reduce 1 μ mol of EV_{ox} per min at 30°C and pH 8. Values are the means ± standard variation, $n = 3$.

^b Kinetic data were assayed at 30°C, pH 8. The kinetic parameters were calculated by fitting the initial rates obtained at six different EV concentrations (0.0625–32 mM) to the nonlinear hyperbolic regression using SigmaPlot 10.0. All enzymatic activities were determined in triplicate (see details in the Methods section).

^c The values of k_{cat} were calculated from V_{max} for EV.

f

Supplementary Figure 8 | f, ChCODH2 viologen-related mutants. The values of $^{app}k_{cat}$ were calculated from V_{max} for EV. The data represent the mean \pm S.D. determined from $n = 3$ independent experiments.

2. Moreover, to determine the affinity of WT and R57G/N59L mutants for viologen, we conducted Isothermal Titration Calorimetry (ITC) analysis under anaerobic conditions. The reaction of CO oxidation based on mediator oxidized viologen (Md_{ox}) follows a ping-pong mechanism.

Therefore, we measured the affinity of WT and R57G/N59L under conditions with substrate CO (Supplementary Fig. 9). The K_D values were $449 \mu\text{M}$ for WT and $144 \mu\text{M}$ for R57G/N59L, indicating that the affinity of WT is about four times lower than R57G/N59L. This aligns with the observed lower K_M values for R57G/N59L compared to WT. Although there are differences in the values between ITC affinity and K_M due to different measuring conditions and enzyme states, the trends were consistent, indicating a significant increase in the affinity of R57G/N59L for ethyl viologen compared to WT.

a

b

ChCODH2	Substrate	Mediator	Binding affinity, K_D (μM)	Number of binding sites
WT	CO	EV_{ox}	449 ± 4.3	1.8 ± 0.4
R57G/N59L	CO	EV_{ox}	144 ± 1.3	2.8 ± 0.1

Supplementary Figure 9 | ITC analysis for EV_{ox} binding in *Ch*CODH2 WT and R57G/N59L. **a**, Thermograms and binding isotherms depict the interaction of *Ch*CODH2 WT and R57G/N59L for oxidized ethyl viologen (EV_{ox}) in the presence of CO. **b**, Comparative analysis of binding affinity and the number of binding sites in *Ch*CODH2 WT and R57G/N59L variant. ITC experiments were conducted under anaerobic conditions in a customized glove box to eliminate oxygen interference. The experiments involved a concentration of 52 μ M purified CODH enzyme and 2.4 μ L of 2.5 mM EV_{ox}, reacted in CO-saturated buffer (20 mM Tris/HCl, pH 7.5) containing 1 mM tris(2-carboxyethyl)phosphine at 25°C (see Methods section). Values are the means \pm standard variation, $n = 2$.

3. As described in the response to question Q2, we observed the redox potential changes in the D-cluster by conducting EV-free electrochemical reactions through protein film for WT, F41 variants, and R57G/N59L mutant. Despite the various specific activity changes in EV-based mutants, the EV-free electrochemical responses showed similar CO oxidation peaks' redox potentials for all F41 variants compared to WT (Supplementary Fig. 4). This indicates that mutations at the F41 position do not significantly alter the overall redox potential of the protein and that the internal D-cluster is functioning normally in electron transfer, with no significant changes in redox potential.

Supplementary Figure 4 | Viologen-free electrochemical reaction of *Ch*CODH2 variants. Cyclic voltammetry (CV) was conducted on fluorine-oxide (FTO) electrodes using CODH enzyme films. The black lines represent experiments performed with 100% (*v/v*) CO, while the grey lines indicate experiments carried out without CO. Experimental conditions included a temperature of 25 °C, a 200 mM HEPES/NaOH buffer (pH 8.0), and a scan rate of 10 mV s⁻¹. The observed potential (V vs RHE) in CO oxidation reactions is indicated by an orange dotted line at 0.28V vs RHE. All tests were conducted in triplicate to verify reproducible CV profiles. Abbreviations: CODH, carbon monoxide dehydrogenase; FTO, fluorine-oxide; RHE, reversible hydrogen electrode; *wo*, without.

4. We view the issues raised by the reviewer as invaluable from a research standpoint, as they provide an opportunity to re-examine and reinterpret our research from a unique perspective. However, based on the electrochemical response results, it appears that electron transfer occurs not only via the viologen-based F41 but also directly to the D-cluster. Therefore, we conclude that electron transfer can occur either through F41 or directly to the D-cluster.

Q2. Authors use site-directed mutagenesis to study the effect of residue substitutions on viologen reduction and use the data for statements such as “residue 41 near the D-cluster constitutes the pivotal site for viologen interactions in *Ch*CODH2.” It has long been known that mutating aromatic residues near cofactors can dramatically alter the cofactor’s redox potential (*Biochemistry* 1996, 35, 50, 15980 for example). If substituting the F41 didn’t affect viologen reduction that would have been an interesting result or if F41 substitution didn’t change the redox potential but did affect viologen reduction anyway, that would be newsworthy. Authors also substitute an arginine residue. Again, it is well known that having a positivity charged residue near a cofactor influences the redox potential. Also, increasing solvent accessible alters the

redox potential. Authors made a number of alterations to the protein that undoubtedly changed the redox potential and thus affected viologen reduction. Studies like this have been done many times, and in those studies, redox potential were measured. This study duplicates past work but is not done as well as what came before.

→ In response to the reviewer's comment, we agree that it would be interesting and newsworthy if the substitution of F41 in *Ch*CODH2 affected viologen reduction without altering the redox potential. To verify this, we conducted viologen-free electrochemical reactions through protein film for *Ch*CODH2 WT, R57G/N59L, and F41 variants. The redox potentials observed indicated that the WT displayed a CO oxidation potential around 300 mV, which is consistent with the method reported in previous studies (*J. Phys. Chem. B* **119**:13690, 2015. doi.org/10.1021/acs.jpcc.5b03098). Notably, the R57G/N59L and all F41 variants tested also showed similar cyclic voltammetry (CV) profiles to WT. This suggests that mutations at the F41 site do not alter the D-cluster's redox potential, but specifically impact viologen reduction, as reflected in our catalytic property analyses.

Supplementary Figure 4 | Viologen-free electrochemical reaction of *Ch*CODH2 variants. Cyclic voltammetry (CV) was conducted on fluorine-tin oxide (FTO) electrodes using CODH enzyme films. The black lines represent experiments performed with 100% (v/v) CO, while the grey lines indicate experiments carried out without CO. Experimental conditions included a temperature of 25 °C, a 200 mM HEPES/NaOH buffer (pH 8.0), and a scan rate of 10 mV s⁻¹. The observed potential (V vs RHE) in CO oxidation reactions is indicated by an orange dotted line at 0.28V vs RHE. All tests were conducted in triplicate to verify reproducible CV profiles. Abbreviations: CODH, carbon monoxide dehydrogenase; FTO, fluorine-tin oxide; RHE, reversible hydrogen electrode; wo, without.

Furthermore, the reference provided by the reviewer (*Biochemistry* **35**: 15980, 1996. doi.org/10.1021/bi962124n) pertains to studies on FMN-based flavodoxin, where changes in surrounding residues altered the redox potential of flavodoxin. However, as our electrochemical reactions have shown, the F41 variants we introduced did not exhibit changes in redox potential. Therefore, the activity changes in F41 variants appear to be due to a mechanism different from the change in key molecule redox potential that led to activity loss in the referenced study. Consequently, considering all these aspects, we believe that our current study does not duplicate past work and provides new insights.

Q3. There are also some portions of the manuscript that do not contain proper citations for literature on the structures of CODH, so some citations should be added.

1) Page 3, Line 70: A citation from 2011 is used for the statement regarding the constitution of the C-cluster, whereas the first structures of the C-cluster were published in 2001. The citation on line 70 should be replaced with the following citation: *PNAS* 2001, Drennan *et al.* and *Science* 2001, Dobbek *et al.*

→ In accordance with the reviewer's comment, we have updated the citation to reflect more foundational and pertinent references. The previous citation from 2011 has been replaced with the suggested references: “*PNAS* 2001, Drennan *et al.*” and “*Science* 2001, Dobbek *et al.*” (Line 66).

2) Page 3, Line 73: Citation 5 refers to one structure from 2022, but does not fully support the statement about all of the structural information available showing conservation. The follow citations should be added:

- o *PNAS* 2001, Drennan *et al.*
- o *JACS* 2004, Dobbek *et al.*
- o *Angew Chem* 2017 Domnik *et al.*
- o *Elife* 2018, Wittenborn *et al.*

→ In response to the reviewer's comment, we have added four additional references to support the statement about the conservation of structural information (Line 68).

3) Page 6, Lines 145-146: PDB entries are mentioned without citing the source publications. Add citations:

- o *PNAS* 2001, Drennan *et al.*
- o *JBC* 2019, Wittenborn *et al.*

→ In response to the reviewer's comment, we have added the necessary citations for the mentioned PDB entries in our manuscript (Line 150).

4) Figures 1-4 contain renderings of atomic coordinates deposited in the PDB, but the source publications are not cited in the figure legend and/or in a location in the text relevant to the corresponding figure and in the case of Figure 1, the PDBIDs are also not included. The following information should be added:
o Figure 1: include PDBIDs used (6ELQ, 1SU7, 6OND, 1JQK) and source publications (*Angew Chem* 2017 Domnik *et al.*, *JACS* 2004, Dobbek *et al.*, *JBC* 2019, Wittenborn *et al.*, *PNAS* 2001, Drennan *et al.*, respectively). *ChCODH1*: is there a typo meant to be *ChCODH5*? If not, what is the source?

→ In accordance with the reviewer's comments, we have added the relevant references for PDB IDs mentioned in the text and specifically in Figure 1b. In Figure 1b, along with the figure legends, we have included *RrCODH* (PDB 1JQK) and *DvCODH* (PDB 6OND), indicating the respective references. To enhance visibility in the figure, only *RrCODH* and *DvCODH* are prominently displayed. Furthermore, regarding the previously depicted *ChCODH1*, which was initially inserted using a structure predicted from the AlphaFold database but is now removed, we have included a description of the methodology for using predicted structures (such as *DsCODH* from *Dethiosulfatarculus sandiegensis*; AF-A0A0D2HSF2) in the Methods section (Line 486–490).

Additionally, for Figure 1, we have included the PDB IDs (1SU7, 6OND, 1JQK) and the source publications: *JACS* 2004 Dobbek *et al.* for 1SU7, *JBC* 2019 Wittenborn *et al.* for 6OND, and *PNAS* 2001 Drennan *et al.* for 1JQK.

b

o Figure 2: include source publications: *Angew Chem* 2017 Domnik *et al.*, *JACS* 2004, Dobbek *et al.*, *JBC* 2019, Wittenborn *et al.*, *PNAS* 2001, Drennan *et al.*, *Angew Chem* 2022 Jeoung *et al.*

→ In response to the reviewer's comment, we have included the specific references for *ChCODH2* (1SU7), *RrCODH* (1JQK), *ChCODH4* (6ELQ), and *DvCODH* (6OND) in Figure 2c. The corresponding references are as follows: *Angew Chem* 2017, Domnik *et al.*; *JACS* 2004, Dobbek *et al.*; *JBC* 2019, Wittenborn *et al.*; *PNAS* 2001, Drennan *et al.* Additionally, the previously included structure of *ChCooS-V* has been replaced with the AlphaFold database predicted structure of *DsCODH* (AF-A0A0D2HSF2).

Figure 2. c, Conserved coordination of D-cluster in CODHs. The D-cluster environments of both 4Fe4S and 2Fe2S CODHs were analyzed by examining their sequences and structures. While the cysteine residues differ between 4Fe4S and 2Fe2S CODHs, the position of aromatic residues remains unchanged. The PDB IDs are *ChCODH2* (1SU7)¹⁴, *RrCODH* (1JQK)¹³, *ChCODH4* (6ELQ)¹⁵, *DvCODH* (6OND)²¹, and *DsCODH* (AF-A0A0D2HSF2) is a predicted structure from the AlphaFold database⁴⁷.

o Figures 3 and 4: include citation to: *JACS* 2004, Dobbek *et al.*

→ In response to the reviewer's comment, we have incorporated the citation to Dobbek *et al.*, *JACS* 2004, into the legends of Figures 3 and 4 (Line 626–627, 652).

Q4. There are issues with the crystallography.

1) Page 9, Line 243 and Supplementary Fig 8b: an $2F_o - F_c$ omit map should be shown around the viologen and the nearby sidechains. A $F_o - F_c$ map with positive and negative density should also be shown. This binding site and the interactions should be described, and the buried surface area should be measured. Omit maps should also be shown for structures solved without viologens for comparison. Omit maps with positive and negative difference density should be shown in Supp Fig. 8a too. The C-cluster density looks terrible in the PDB validate file (giant balls of negative density), so this figure misleads the reader toward believing that the C-cluster looks better than it is. Anomalous maps are mentioned but not shown. They should be shown.

→ In response to the reviewer's suggestions:

1. We have revised Figure 4b to include the viologen structures and their nearby sidechains, showcasing both $2F_o - F_c$ (at 1σ) and $F_o - F_c$ (at 2.5σ) maps with positive and negative contour levels. Supplementary Fig. 12 now features omit maps at 2.5σ for viologens with $F_o - F_c$ maps (+/- contour) for comparison. We have added detailed descriptions of the protein-ligand interactions, noting hydrophobic interactions between EV and residues 44T, 583L, 612L, and charged interactions with E43. The buried surface areas of EV and BV were calculated to be 160.5 \AA^2 and 230.3 \AA^2 , respectively, as per PISA analysis (*J. Mol. Biol.* **372**: 774, 2007), respectively. These descriptions are added in the revised manuscript (Line 252–259).

Figure 4. b, Crystal structures of the R57G/N59L variant in complex with EV and BV (PDB ID: 8X9F and 8X9G, respectively). Interacting residues and electron density maps for EV and BV are illustrated, with $2Fo-Fc$ maps at 1σ (blue) and $Fo-Fc$ at 2.5σ (green and red for $+/-$ level), respectively. Colored arrows indicate EV and BV positions in the $Fo-Fc$ maps.

Supplementary Figure 12 | Omit maps of viologens in the R57G/N59L variants. **a**, The omit map of EV in the R57G/N59L variant is displayed in grey mesh contoured at 2σ . Accompanying this is the $Fo-Fc$ difference electron-density map at 2.5σ , with positive and negative levels shown in green and red, respectively, indicated by corresponding colored arrows. Interacting residues with viologens are represented in stick form. **b**, For BV in the R57G/N59L variant, the omit map is similarly shown in grey mesh contoured at 2σ , along with the $Fo-Fc$ difference electron-density maps at 2.5σ in green and red for positive and negative levels, respectively.

2. To address the reviewer's concerns, we updated the C cluster model in our structure from the Ni-4Fe-4S-mut2S (NFS) to a more accurate Ni-4Fe-4S (XCC) model. In the R57G/N59L structure, the absence of the S2 atom in the C cluster was noted. The XCC model, lacking the μ_2 -S ligand (S2), provided a more suitable representation for this conformation. This change resulted in a better match with the electron density map

and reduced negative *Fo-Fc* discrepancies. The revised model is clearly demonstrated in the *Fo-Fc omit* map at 3.0σ , providing a more accurate representation of the C cluster. Details of these improvements are presented in Supplementary Fig. 11.

3. To validate the integrity of atoms within the clusters, we computed the Fe anomalous map for GL apo in high PEG concentration (PDB: 8X9D) and F41C (PDB: 8X9H). This analysis confirmed that the Fe atoms in the clusters remained unchanged, even with mutations. For comprehensive documentation, a table with data collection statistics has been added as Supplementary Table 10.

Supplementary Figure 11 | *Fo-Fc* difference electron density maps and Fe anomalous maps of clusters in R57G/N59L variant. The *Fo-Fc* difference electron-density maps of B, C, and D clusters in R57G/N59L variant are shown in grey mesh contoured at 3σ . Fe, S, and Ni atoms are coloured orange, yellow and green, respectively. In the C cluster, 1SU7²¹ adopts the Ni-4Fe-4S-mut2S (NFS) conformation, while both 1SUF²¹ and 8X9H feature the Ni-4Fe-4S (XCC) conformation. The B and C clusters adopt the same conformations in both structures. In R57G/N59L, the S2 atom in the C cluster was not observed, and the XCC model, which lacks the μ_2 -S ligand (S2)^{35, 36}, was found to be a more suitable and accurate model for this conformation. Bond lengths of clusters in R57G/N59L variant are presented on each bond. Anomalous difference Fourier maps illustrating the positions of Fe atoms in clusters are shown as orange mesh, contoured at 10, 5, and 10 σ in each cluster.

Supplementary Table 10 | Data collection statistics for Fe anomalous data

Data set	R57G/N59L apo (PDB: 8X9D)	F41C (PDB: 8X9H)
Data collection		
Energy (keV)	7.140	7.140
Space group	C2	C2
Unit-cell length (a , b , c , Å)	112.6, 75.2, 71.6	112.1, 75.3, 70.7
Unit-cell angle (α , β , γ , °)	90.0 111.6, 90.0	90.0, 111.1, 90.0
Resolution range (Å)	50.00 – 2.80 (2.82 – 2.80)*	50.0 – 1.80 (1.81 – 1.80)*

Total / unique reflections	55,136 / 25,928	202,769 / 94,544
Completeness (%)	96.3 (96.6)*	94.3 (90.2)*
Average $I/\sigma(I)$	18.7 (9.6)*	15.4 (8.6)*
CC _{1/2}	99.8 (98.9)*	99.5 (98.7)*
$R_{\text{merge}}^{\dagger}$ (%)	3.4 (6.8)*	4.2 (7.2)*
Redundancy	2.1 (2.1)*	2.1 (2.1)*

*Values in parentheses refer to the highest-resolution shell. Data were collected from one crystal.

$\dagger R_{\text{merge}} = \sum_{hkl} \sum_i |I_i(hkl) - \langle I(hkl) \rangle| / \sum_{hkl} \sum_i I_i(hkl)_i$, where $I(hkl)$ is the intensity of reflection hkl , \sum_{hkl} is the sum over all reflections, and \sum_i is the sum over i measurements of reflection hkl .

2) Supplementary Table 8: The gap between R_{work} and R_{free} is quite large, in particular for 8ISG and 8ISH, and is suggestive of overfitting. The authors should return to model building and refinement to obtain more suitable $R_{\text{work}}/R_{\text{free}}$ with separation less than 5%. Were the same R_{free} flags used as were used for previous structures? It seems that the cell and space groups are the same, which does raise the question as to why molecular replacement was used.

→ In response to the reviewer’s concern about the significant gap between $R_{\text{work}}/R_{\text{free}}$, particularly for the 8ISG and 8ISH structures, we have undertaken additional rounds of model building and refinement. This effort was aimed at achieving a more appropriate $R_{\text{work}}/R_{\text{free}}$ separation of less than 5%. Consequently, the final $R_{\text{work}}/R_{\text{free}}$ values for the R57G/N59L_apo in high PEG (25%, v/v) (PDB id: 8X9D), R57G/N59L_apo in low PEG (5%, v/v) (PDB id: 8X9E), R57G/N59L with EV (PDB id: 8X9F), R57G/N59L with BV (PDB id: 8X9G), and F41C structures are now approximately 0.16/0.21, 0.17/0.22, 0.24/0.26, 0.22/0.27, and 0.14/0.19, respectively. These revised values have been updated in Supplementary Table 9.

Regarding the R_{free} flags, we confirm that the same flags were consistently used across all refinements for structural calculations. Although the cell and space groups for all structures are identical, raising the question of the necessity for molecular replacement, we opted for this method for specific reasons. We used molecular replacement for determining the R57G/N59L_apo in high PEG structure (PDB id: 8X9D), utilizing the 1SU7 model as a reference. This refined structure of the R57G/N59L_apo was then employed as a basis for calculating the other structures. This approach ensured consistency and accuracy in our structural determinations.

3) Methods: in the “crystallization and structural determination” section, it would be useful for the authors to state the time it took to grow crystals suitable for data collection. Also, authors should provide information about the protein solution – it was 10mg/ml but no information is provided about the buffer or pH or salt or glycerol.

→ Protein crystals for data collection grew in about 14 days, and the composition of protein solution was 20 mM Tris-HCl pH 7.5 and 3 mM dithioerythritol (DTE), which is described in method section (Line 493–500).

Minor comments:

Q5. Authors should use normal nomenclature for residue substitutions. G57/L59 should be R57G/N59L for example.

→ We have revised the manuscript to adhere to the standard nomenclature for residue substitutions. Accordingly, G57 has been changed to R57G, L59 to N59L, and W559 to A559W in the text (Line 32, 200, 204–205, 208, 213, 217, 223, 225, 403, 557).

Q6. Page 6, Line 137: “cubage” should say “cubane”

→ We changed the typographical error “cubage” to “cubane” (Line 142).

Q7. Page 6, Line 158: “DvCODH F43A” should say “DvCODH F44A”

→ We have corrected the text where "*Dv*CODH F43A" was mistakenly written. It has been updated to "*Dv*CODH F44A" (Line 162). Additionally, we have made the same correction in the legend of Figure 2d to accurately reflect "*Dv*CODH F44A" (Line 616).

Q8. Page 14, why do enzyme assays have 0-250 micromolar O₂?

→ We have measured up to 250 μM O₂ to approximate the conditions of atmospheric air, which contains about 20% (v/v) O₂ (equivalent to ~260 μM). This detail has been specified in the text as follows: "Measurements were taken up to 250 μM O₂, which is close to the atmospheric air concentration of 20% (v/v) O₂ (~260 μM)." (Line 401–402)

Q9. Page 15, are activities being reported per monomer or per dimer?

→ In response to the reviewer's question, the specific activity of the CODH enzyme is quantified relative to the amount of enzyme protein. As outlined in the text, specific activity is defined and expressed in terms of enzyme activity (number of μmol of reduced EV_{ox}) per mass of protein (mg). Considering that the functional form of the CODH enzyme is known to be a homodimer, and that two monomeric subunits must combine to form an active unit, it is appropriate to express the specific activity in terms of the dimer unit. Therefore, the calculation of specific activity in our activity measurements is based on the functional unit, which is the dimer.

Q10. Supplementary Fig 7: it would be useful to add a key for the tan sticks in the figure caption, presumably they are residues at the tunnel exits.

→ In response to your suggestion, we have added a key to the caption of Supplementary Fig. 10.

Supplementary Figure 10 | F41 neighboring and tunnel-forming residues in *Ch*CODH2 W559. The locations of residues, including R57, N59, and W559, were displayed with tunnels of the *Ch*CODH2 variant (PDB ID: 7XDM)²⁶. F41, R57, and N59 (cyan stick) were found to be distantly located from W559 (red stick) and were also observed to be separated from the protein substrate tunnel exits. The residues at the tunnel exits are as follows: for tunnel #1, E43/K450/P585; for tunnel #2, T593/T597/V610; for tunnel #3, Q206/S599/I603; for tunnel #4, L168/A537/P577; for tunnel #5, V433/L543/L632/L634; for tunnel #6, L419/N431/V503/I507.

Q11. Supplementary Fig 6: resolution is poor – it is impossible to read figure.

→ We have updated Supplementary Fig. 8 with a high-resolution version.

Supplementary Figure 8 | Nonlinear hyperbolic kinetic profiles. Catalytic properties of *Ch*CODH2 mutants for EV were estimated from the non-linear regression method. **a**, CODH WTs. **b**, *Ch*CODH2 F41A and *Rr*CODH F43A. **c**, *Ch*CODH2 R57 mutants. **d**, *Ch*CODH2 N59 mutants. **e**, *Ch*CODH2 double and triple mutants. **f**, *Ch*CODH2 viologen-related mutants. The values of $^{app}k_{cat}$ were calculated from V_{max} for EV. The data represent the mean \pm S.D. determined from $n = 3$ independent experiments.

Reviewer #3 (Remarks to the Author):

Carbon monoxide dehydrogenase is a redox multimetallic enzyme that plays a central role in carbon metabolism in anaerobic microorganisms, reversibly catalyzing the reduction of CO₂ to CO. This oxygen-sensitive enzyme is a homodimer with several FeS clusters: two buried active sites (one per monomer) composed of heteronuclear FeNiS clusters and additional FeS clusters acting as electron relays between the active site and the protein surface: 2 B-clusters (1 per monomer) and 1 intermolecular D-cluster acting as the first electron acceptor from the solvent. This redox Ni-enzyme is well known to interact with the well described redox mediators, viologens despite a low affinity in the mM range. In this study, Suk Min Kim and collaborators, describe in detail the interaction of several CODH with a series of viologens, using a combined approach of site-directed mutagenesis, enzymology and X-ray crystallography. Unexpectedly, they identified specific interactions between viologens and protein residues, although these redox mediators have been proposed to interact by random collision with redox enzymes. CODH and other redox enzymes have a great potential for applications in renewable energy and CO₂ valorization. Consequently, a better understanding of enzyme/redox mediators' interactions will help not only the scientific bio-inorganic community but also the biotechnology sector for the development of tailor-made biocatalysts for industrial applications

The paper is well-written, concise, clear and the methodology is well-adapted. The study is well-conducted and the result of a remarkable piece of work. Undoubtedly, the study will be of great interest to readers of *Nature communications*. I have only minor comments to address, described below:

→ Thank you for your positive and insightful review of our CODH study. Your encouraging words significantly boost our motivation. We're diligently working on extending our research to further applications and societal benefits, and look forward to sharing our progress for collaborative growth. We particularly value your observations on the specific CODH-viologen interactions, which enhance our understanding of enzyme-mediator dynamics—critical for bio-inorganic research and biocatalyst development. We're glad our paper's clarity and methodology resonated with you and will address your comments with thorough attention. Your support propels our research efforts, and we're excited about our ongoing contribution to science and society.

Q1. How do the authors explain that in some cases, the mutation led to increased relative activity (e.g. Y224A or C496A)?

→ For the measurement of CODH activity, we used Ni-NTA purified proteins. It's important to note that there are inherent variations in protein purity among different samples. As a result, for Y224A, we observed a difference of about 17% in activity compared to the wild type (WT), but this variation does not appear to be severe (Line 113–114). On the other hand, for C496A, as the reviewer correctly noted, there was a significant difference in the previous values presented in Supplementary Fig. 3. To address this, we conducted verification experiments. After repeating these experiments, it was confirmed that the relative activity of C496A was 60.6%, compared to that of WT (Line 119). Consequently, we have updated Supplementary Fig. 3 to reflect these findings like “Except C344A, which was not expressed in *E. coli*, surface-exposed C and M variants displayed no significant alterations in reactivity toward EV” (Line 120–121).

Supplementary Figure 3 | Relative activities of C496A, M116A and M355A variants. Enzymatic activity is anaerobically measured through CO oxidation at 30°C in CO-saturated HEPES buffer pH 8 containing 20 mM EV_{ox}. The relative activities of variants were calculated by comparison of the WT's specific activity. The data represent the mean ± S.D., determined from $n = 3$ independent experiments. C344A was not expressed in *E. coli*.

Q2. In the introduction, the authors propose that viologens interact with protein residues via π - π stacking. Although, the X-ray structure of the G57/L59 double mutant revealed the EV/BV binding site in CODH, it clearly shows that there is no direct interaction between EV/BV and F41, as previously suggested. This point should be discussed. Despite its essential role highlighted by site-directed mutagenesis, its role in electron transfer should be more discussed. With a distance EV-D Cluster of 12.6 Å, a direct electron transfer is still possible. In my opinion, both pathways would coexist (via or not F41) and the authors should be less assertive about the role of F41 in electron transfer. In future studies, DFT calculations should be carried out to investigate the electron pathway in CODH in the close environment of D-cluster. A comparison between a series of CODH, the *Ch*CODH-2 WT enzyme and mutants would be highly informative.

→ Responding to the comments from Reviewers #1 and #3, we conducted additional verifications to clarify the relationship between mutations and the binding site in the viologen-CODH complex, leading to the following interpretations:

1. In the structure of the R57G/N59L-viologen complex, the interaction site for viologens was identified near residue F41, specifically E43, T44, P60, L583, L612 (indicated in Figure 4b). To investigate the influence of E43's negative charge on the binding with positively charged EV, we measured the K_M changes in variants E43K and E43R, which have positively charged residues (Table 1 and Supplementary Fig. 8). The K_M for these variants showed a 2.5- to 3.2-fold increase compared to WT, indicating an electrostatic interaction between the negatively charged E43 and the positively charged oxidized viologen molecule. As seen in the R57G/N59L-viologen complex structure, the binding of CODH and viologen is supposed mainly mediated by electrostatic interactions between the negatively charged Glu at position 43 and the positively charged oxidized EV planar ring structure (Figure 4b). Therefore, we conclude that the binding of CODH to viologen molecules is critically influenced by the electrostatic interaction at position 43.

Figure 4. b. Crystal structures of the R57G/N59L variant in complex with EV and BV (PDB ID: 8X9F and 8X9G, respectively). Interacting residues and electron density maps for EV and BV are illustrated, with $2Fo-Fc$ maps at 1σ (blue) and $Fo-Fc$ at 2.5σ (green and red for +/- level), respectively. Colored arrows indicate EV and BV positions in the $Fo-Fc$ maps.

Table 1. Kinetic constants of CODH variants

Enzyme	Specific activity ^a (U/mg)	K_M^{EV} (mM)	k_{cat}^{EV} (s ⁻¹)	k_{cat}/K_M^{EV} (s ⁻¹ ·mM ⁻¹)
Wild type				
Ch CODH2	1,800 ± 29	2.4 ± 0.1	2,200 ± 30	910 ± 15
Ch CODH4	85 ± 1.2	1.3 ± 0.1	85 ± 0.1	66 ± 2.0
Rt CODH	1,100 ± 15	2.4 ± 0.1	1,300 ± 12	560 ± 17
To CODH	170 ± 1.5	2.9 ± 0.3	210 ± 16	71 ± 2.7

F41 or F43 variants				
Ch CODH2 F41A	0.3 ± 0.03	7.0 ± 0.5	0.4 ± 0.01	0.06 ± 0.004
Rr CODH F43A	0.5 ± 0.05	11.9 ± 0.3	0.8 ± 0.01	0.06 ± 0.001
Ch CODH2 R57/N59 variants				
R57G/N59F	800 ± 31	0.2 ± 0.1	1,200 ± 40	5,300 ± 347
R57G/N59K	1,500 ± 38	0.5 ± 0.1	1,600 ± 65	3,400 ± 489
R57G/N59L	2,300 ± 21	0.2 ± 0.1	2,100 ± 24	11,800 ± 452
Ch CODH2 A559W variants				
A559W	2,000 ± 13	2.0 ± 0.1	2,100 ± 39	1,000 ± 40
R57G/N59L/A559W	2,700 ± 130	0.4 ± 0.1	2,000 ± 22	4,700 ± 47
Ch CODH2 viologen-related variants				
E43K	1,000 ± 42	5.7 ± 0.1	1,600 ± 47	300 ± 10
E43R	300 ± 30	7.0 ± 0.4	500 ± 47	100 ± 2.8
T44A	2,200 ± 22	1.8 ± 0.1	2,500 ± 50	1,400 ± 23
P60A	1,500 ± 51	1.9 ± 0.2	1,500 ± 60	750 ± 32
L583A	1,700 ± 18	2.0 ± 0.4	1,700 ± 41	590 ± 21
L612A	1,200 ± 72	1.7 ± 0.2	1,000 ± 97	590 ± 33

^a Specific activities were determined at 20 mM ethyl viologen (EV) in HEPES buffer saturated with CO (30°C, pH 8). One unit (U) of CODH activity was defined as the amount of enzyme required to reduce 1 μmol of EV_{ox} per min at 30°C and pH 8. Values are the means ± standard variation, *n* = 3.

^b Kinetic data were assayed at 30°C, pH 8. The kinetic parameters were calculated by fitting the initial rates obtained at six different EV concentrations (0.0625–32 mM) to the nonlinear hyperbolic regression using SigmaPlot 10.0. All enzymatic activities were determined in triplicate (see details in the Methods section).

[†] The values of *k*_{cat} were calculated from *V*_{max} for EV.

f

Supplementary Figure 8 | f, *Ch*CODH2 viologen-related mutants. The values of ^{app}*k*_{cat} were calculated from *V*_{max} for EV. The data represent the mean ± S.D. determined from *n* = 3 independent experiments.

2. Meanwhile, the R57 and N59 mutations appear to indirectly affect the viologen interaction (Fig. 4a). Observing the changes in catalytic properties of these R57 and N59 variants (Table 1 and Supplementary Table 7), it is evident that the reaction kinetics for viologen reduction differ markedly with the substitution of positively charged Arg57, depending on the charge and size of the substituent residues. This suggests that the mutations alleviate or remove the repulsive influence between the positively charged electron mediator oxidized EV and positively charged residue Arg57, thereby easing spatial constraints for viologen approach (Figure 4a). This led to an improvement in the *K*_M for EV in the R57G/N59L mutant and a difference in the protein surface binding environment, interpreted as an indirectly positive influence on viologen complex formation in R57G/N59L (8X9D) structure compared to WT (1SU7).

Figure 4. a, Structural comparison of the D-cluster region in the crystal structures of *ChCODH2* WT (grey, PDB ID: 1SU7)¹⁴ and viologen-free R57G/N59L variant (pink, PDB ID: 8X9D). The surface charges and tunnel cavities near the D cluster of the R57G/N59L variant and the WT are shown on the right. Surface charge was calculated using the APBS and PDB2PQR web server⁵⁶ at pH 8, and cavity volumes were quantified through KMfinder's pseudo-atom filling method⁵⁷.

Table 1. Kinetic constants of CODH variants

Enzyme	Specific activity ^a (U/mg)	K_M^{EV} (mM)	k_{cat}^{EV} (s ⁻¹)	k_{cat}/K_M^{EV} (s ⁻¹ ·mM ⁻¹)
Wild type				
ChCODH2	1,800 ± 29	2.4 ± 0.1	2,200 ± 30	910 ± 15
ChCODH4	85 ± 1.2	1.3 ± 0.1	85 ± 0.1	66 ± 2.0
RrCODH	1,100 ± 15	2.4 ± 0.1	1,300 ± 12	560 ± 17
ToCODH	170 ± 1.5	2.9 ± 0.3	210 ± 16	71 ± 2.7
F41 or F43 variants				
ChCODH2 F41A	0.3 ± 0.03	7.0 ± 0.5	0.4 ± 0.01	0.06 ± 0.004
RrCODH F43A	0.5 ± 0.05	11.9 ± 0.3	0.8 ± 0.01	0.06 ± 0.001
ChCODH2 R57/N59 variants				
R57G/N59F	800 ± 31	0.2 ± 0.1	1,200 ± 40	5,300 ± 347
R57G/N59K	1,500 ± 38	0.5 ± 0.1	1,600 ± 65	3,400 ± 489
R57G/N59L	2,300 ± 21	0.2 ± 0.1	2,100 ± 24	11,800 ± 452
ChCODH2 A559W variants				
A559W	2,000 ± 13	2.0 ± 0.1	2,100 ± 39	1,000 ± 40
R57G/N59L/A559W	2,700 ± 130	0.4 ± 0.1	2,000 ± 22	4,700 ± 47
ChCODH2 viologen-related variants				
E43K	1,000 ± 42	5.7 ± 0.1	1,600 ± 47	300 ± 10
E43R	300 ± 30	7.0 ± 0.4	500 ± 47	100 ± 2.8
T44A	2,200 ± 22	1.8 ± 0.1	2,500 ± 50	1,400 ± 23
P60A	1,500 ± 51	1.9 ± 0.2	1,500 ± 60	750 ± 32
L583A	1,700 ± 18	2.0 ± 0.4	1,700 ± 41	590 ± 21
L612A	1,200 ± 72	1.7 ± 0.2	1,000 ± 97	590 ± 33

^a Specific activities were determined at 20 mM ethyl viologen (EV) in HEPES buffer saturated with CO (30°C, pH 8). One unit (U) of CODH activity was defined as the amount of enzyme required to reduce 1 μmol of EV_{ox} per min at 30°C and pH 8. Values are the means ± standard variation, *n* = 3.

^b Kinetic data were assayed at 30°C, pH 8. The kinetic parameters were calculated by fitting the initial rates obtained at six different EV concentrations (0.0625–32 mM) to the nonlinear hyperbolic regression using SigmaPlot 10.0. All enzymatic activities were determined in triplicate (see details in the Methods section).

[†] The values of k_{cat} were calculated from V_{max} for EV.

Supplementary Table 7 | Kinetic constants of *ChCODH2* R57 and N59 variants

Enzyme	Specific activity ^a (U/mg)	K_M^{EV} (mM)	k_{cat}^{EV} (s ⁻¹)	k_{cat}/K_M^{EV} (s ⁻¹ ·mM ⁻¹)
Wild type				
ChCODH2	1,800 ± 29	2.3 ± 0.1	2,200 ± 28	1,000 ± 11
ChCODH2 R57 variants				

R57A	3,400 ± 82	0.5 ± 0.1	3,400 ± 78	6,700 ± 330
R57E	3,300 ± 56	0.4 ± 0.1	3,400 ± 13	9,000 ± 153
R57F	1,700 ± 28	0.5 ± 0.1	1,800 ± 23	3,500 ± 231
R57G	3,800 ± 29	0.3 ± 0.1	4,200 ± 51	14,700 ± 576
R57Q	3,600 ± 45	0.4 ± 0.1	3,600 ± 27	8,600 ± 130
R57S	3,500 ± 49	0.4 ± 0.1	3,500 ± 86	9,700 ± 621
ChCODH2 N59 variants				
N59A	5,800 ± 29	1.0 ± 0.1	5,600 ± 18	5,400 ± 165
N59D	5,200 ± 28	0.5 ± 0.1	4,500 ± 4	9,000 ± 82
N59F	1,600 ± 20	1.8 ± 0.2	1,400 ± 74	800 ± 63
N59G	5,100 ± 30	0.8 ± 0.1	4,800 ± 57	6,100 ± 97
N59K	2,600 ± 25	3.3 ± 0.1	1,900 ± 50	590 ± 23
N59L	3,400 ± 19	1.6 ± 0.1	3,100 ± 43	1,900 ± 56

^a Specific activities were determined at 20 mM ethyl viologen (EV) in HEPES buffer saturated with CO (30°C, pH 8). Values are the means ± standard variation, $n = 3$.

^b Kinetic data were assayed at 30°C, pH 8. The kinetic parameters were calculated by fitting the initial rates obtained at six different EV concentrations (0.0625–32 mM) to the nonlinear hyperbolic regression using SigmaPlot 10.0. All enzymatic activities were determined in triplicate (see details in the Methods section).

^c The values of k_{cat} were calculated from V_{max} for EV.

3. New electrochemical verification experiments revealed that F41 plays a key role in the electron relay based on viologen and can also directly transfer electrons through the D-cluster (Supplementary Fig. 4). Variants substituting F41 with hydrophobic residues Ala, Val, and Leu (F41A, F41V, F41L) lost their ability to reduce viologen (Supplementary Table 4). However, in viologen-free electrochemical responses using protein films, the cyclic voltammetry (CV) cycles for CO oxidation in F41 variants were similar to WT (Supplementary Fig. 4). These electrochemical results indicate that the electron transfer ability of F41 variants and the redox potential of FeS clusters (B, C, D) including the D-cluster are well maintained. Considering the observed kinetic data and electrochemical responses, as Reviewer#3 commented, 1) electron transfer to mediator viologen occurs at F41, or 2) electrons are directly transferred to the D-cluster, facilitating the CO oxidation reaction of CODH. Therefore, we infer that the electron transfer based on viologen through E43 is mediated via the key residue F41.

Supplementary Figure 4 | Viologen-free electrochemical reaction of *Ch*CODH2 variants. Cyclic voltammetry (CV) was conducted on fluorine-tin oxide (FTO) electrodes using CODH enzyme films. The black lines represent experiments performed with 100% (v/v) CO, while the grey lines indicate experiments carried out without CO. Experimental conditions included a temperature of 25 °C, a 200 mM HEPES/NaOH buffer (pH 8.0), and a scan rate of 10 mV s⁻¹. The observed potential (V vs RHE) in CO oxidation reactions is indicated by an orange dotted line at 0.28V vs RHE. All tests were conducted in triplicate to verify reproducible CV profiles. Abbreviations: CODH, carbon monoxide dehydrogenase; FTO, fluorine-tin oxide; RHE, reversible hydrogen electrode; *wo*, without.

Supplementary Table 4 | Specific activities of *Ch*CODH2 F41 variants

Ch CODH2 F41 variants	Specific activity (U·mg ⁻¹)	Relative activity (%)
WT (F41)	1,800 ± 29	100 ± 2
F41Y	1,900 ± 68	105 ± 4
F41A	0.3 ± 0.03	0.02 ± 0.002
F41V	8.1 ± 0.9	0.5 ± 0.1
F41L	2.4 ± 0.7	0.1 ± 0.04

The enzyme reaction was performed at 30°C using CO-saturated HEPES buffer pH 8 with 20 mM EV_{ox}. The absorbance change at 578 nm was spectrophotometrically monitored in triplicate to determine the specific activity. The data represent the mean ± standard deviation (S.D.), as determined from $n = 3$ independent experiments. Abbreviations: *Ch*, *Carboxydothemus hydrogenoformans*; CODH, carbon monoxide dehydrogenase; EV_{ox}, oxidized ethyl viologen.

4. As the reviewer suggested, we agree that DFT calculations would be immensely helpful in investigating the electron pathway in CODH near the D-cluster. Currently, our research team is focused on large-scale application studies based on CODH (Molar-scale formate production via enzymatic hydration of industrial off-gases, 2023 *Res. Sq.* doi.org/10.21203/rs.3.rs-3137085/v1), with a 100-L scale pilot study soon to be completed. In future research, based on the reviewer's suggestion, we plan to incorporate this through collaboration with expert research groups.

Q3. The authors should use a method without redox-mediator to measure activities (e.g. electrochemically or with an electron-transfer protein such as ferredoxin). It would be interesting to assess the role of F41 in direct electron transfer to an electrode or “pseudo”- physiological partner. It would highlight (or not) the role of F41 in electron transfer whatever the external electron donor/acceptor.

→ In response to the reviewer's suggestion, we performed viologen-free electrochemical reactions using protein film for *Ch*CODH2 WT, R57G/N59L, and F41 variants (F41A/V/C/Y) to assess CO oxidation. The cyclic voltammetry (CV) cycles showed that both R57G/N59L and F41 variants exhibited CO oxidation peaks similar to WT (Supplementary Fig. 4). This indicates that the overall redox potentials involving the D-cluster in CODH mutants are similar to WT and the mutations have minimal impact on the D-cluster. Additionally, this suggests that direct electron transfer to the D-cluster can occur without a mediator, as predicted by the reviewer. Thus, as previously mentioned, electron transfer in CODH may occur either via viologen-based F41 or directly to the D-cluster.

Supplementary Figure 4 | Viologen-free electrochemical reaction of *Ch*CODH2 variants. Cyclic voltammetry (CV) was conducted on fluorine-tin oxide (FTO) electrodes using CODH enzyme films. The black lines represent experiments performed with 100% (v/v) CO, while the grey lines indicate experiments carried out without CO. Experimental conditions included a temperature of 25 °C, a 200 mM HEPES/NaOH buffer (pH 8.0),

and a scan rate of 10 mV s^{-1} . The observed potential (V vs RHE) in CO oxidation reactions is indicated by an orange dotted line at 0.28V vs RHE. All tests were conducted in triplicate to verify reproducible CV profiles. Abbreviations: CODH, carbon monoxide dehydrogenase; FTO, fluorine-tin oxide; RHE, reversible hydrogen electrode; *wo*, without.

Furthermore, to assess the structural impact of F41 mutation on the D-cluster, we determined the structure of F41C (PDB ID 8X9H) (Supplementary Table 9 and Supplementary Fig. 5). When comparing the C, B, and D clusters of F41C with WT (PDB 1SU7 and 1SUF), we observed that all FeS clusters are similar to WT, indicating no structural changes in the D-cluster due to the F41 mutation. Therefore, based on these electrochemical responses and crystal structure results, our observations suggest that the F41 mutation does not merely lead to an adverse change in the D-cluster but plays a key role in the electron transfer relay.

Supplementary Figure 5 | F41C structure and Fe anomalous maps of clusters. **a**, Structural comparison of the D-cluster region in the crystal structures of *Ch*CODH2 WT (grey, PDB ID: 1SU7)²¹ and F41C variant (light blue, PDB ID: 8X9H). **b**, Anomalous difference Fourier maps illustrating the positions of Fe atoms in B, C, and D clusters contoured at 10, 5, and 10 σ , respectively, are shown in orange mesh. Fe, S, and Ni atoms are coloured orange, yellow and green, respectively. In C cluster, 1SU7²¹ adopts Ni-4Fe-4S-mut2S (NFS), and 1SUF²¹ and 8X9H have Ni-4Fe-4S (XCC) conformations, respectively. The B and C clusters adopt the same conformation in both structures. In F41C, the S2 atom in the C cluster was not observed, and the XCC model, which lacks the μ_2 -S ligand (S2)^{35,36}, was found to be a suitable and accurate model for this conformation.

Q4. Mediator binding site and putative sites in NiFe-CODHs in Figure 1b are barely visible.

→ In response to the reviewer's comment regarding the visibility of the mediator key site and putative sites in NiFe-CODHs in Figure 1b, we have made adjustments for better visibility. We now present only two types of CODHs (*Rr*CODH, 1JQK; *Dv*CODH, 6OND) with the putative mediator binding sites highlighted, and have increased their size for clearer visualization.

b

Figure 1. b, Procedure to identify key sites for interacting with CODH enzymes and viologen. Aromatic residues on the enzyme surface were replaced with alanine residues, likely alanine braille. The influences of mutations on enzymatic activity show relationships between target residues and viologen. This approach could be expanded to other CODHs including *RrCODH* (PDB 1JQK)¹³ and *DvCODH* (PDB 6OND)²¹.

Q5. In figure 2c and Table S2, the authors should remove the structure of *ChCODH5* (7B7Q) which is actually not a CODH enzyme.

→ In response to the reviewer's comment, we acknowledge that the structure originally displayed as “*ChCODH5* (7B7Q)” in Figure 2c, Supplementary Table 2, and Supplementary Fig. 6 and 7 was incorrectly included and is in fact *ChCooS-V*, not a CODH enzyme. Due to the scarcity of known 2Fe–2S CODH structures, we have replaced it with the AlphaFold predicted structure of *DsCODH* (AF-A0A0D2HSF2). As described in the Methods section, the metal cluster portion of FeS was predicted using the known structure of *DvCODH* (6OND) as a model (Line 486–490).

Figure 2. c, Conserved coordination of D-cluster in CODHs. The D-cluster environments of both 4Fe4S and 2Fe2S CODHs were analyzed by examining their sequences and structures. While the cysteine residues differ between 4Fe4S and 2Fe2S CODHs, the position of aromatic residues remains unchanged. The PDB IDs are *ChCODH2* (1SU7)¹⁴, *RrCODH* (1JQK)¹³, *ChCODH4* (6ELQ)¹⁵, *DvCODH* (6OND)²¹, and *DsCODH* (AF-A0A0D2HSF2) is a predicted structure from the AlphaFold database¹⁶.

Supplementary Table 2 | Numbers of surface aromatic residues in viologen-based CODHs

Group	Metal at D-cluster	Organism	Name (PDB code)	No. of surface exposed F/Y/W	No. of residues within 8 Å of Fe-S clusters	No. of putative site
Ni-Fe CODHs	4Fe4S	Carboxydotherrnus hydrogenoformans	ChCODH2 (1SU7) ²¹	10	3	2
		Carboxydotherrnus hydrogenoformans	ChCODH4 (6ELQ) ²²	22	3	1
		Rhodospirillum rubrum	RrCODH (1JQK) ²³	17	5	2
		Moorella thermoacetica	MrCODH (1MJG) ²⁴	22	3	1
	2Fe2S	Desulfovibrio vulgaris	DvCODH (6OND) ²⁵	14	5	2

Supplementary Figure 6 | Multiple alignments of *Ch*CODH-related sequences. In the amino acid alignments of *Ch*CODHs, the amino acid residues are displayed in **black shading with white letters** (identical residue, >90% threshold for shading). The characterized CODHs are marked with asterisks (*). The boxes indicate cysteine residues coordinating each Fe-S cluster (B, C, D): C48, C51, C56, and C70 for B-cluster; C294–295, C333, C446, and C476 for C-cluster; C39, C47, C'39, and C'47 for D-cluster. The CODHs are as follows: *Ch*CODH1 (WP_011344718), *Ch*CODH2 (WP_011343033), *Ch*CODH4 (WP_011343666), *Ch*CooS-V (WP_011342982) from *Carboxydothemus hydrogenoformans*; *Ds*CODH from *Desulfofundulus salinum* (WP_121452276); *Rr*CODH from *Rhodospirillum rubrum* (WP_011389181); *Dv*CODH from *Desulfovibrio vulgaris* (WP_010939375); *Dt*CODH from *D. termitidis* (WP_035067836); *Dve*CODH from *Desulfococcus vexinensis* (WP_028588641); *Dh*CODH from *Desulfobaba hansenii* (WP_100393885).

Supplementary Figure 7 | Coordinating amino acid sequences in proximity to the D-cluster. **a**, Sequence logo representing the frequency in the region 36–63 of *Ch*CODH2 sequences. The sequence logo was generated using WebLogo³⁷. **b**, Partial alignments of CODHs and HCPs (hybrid cluster proteins). The boxes indicate cysteine (orange) and phenylalanine (magenta) residues that coordinate D-cluster. The asterisks denote the CODHs that have been characterized. **c**, Phylogenetic tree of CODHs. The phylogenetic trees showed orthologous relationships based on the amino acid sequences of the CODH proteins (see Methods for details). The CODHs are as follows: *Ch*CODH1 (WP_011344718), *Ch*CODH2 (WP_011343033), *Ch*CODH4 (WP_011343666) from *Carboxydothemus hydrogenoformans*; *Cje*CODH from *C. ferrireducens* (WP_028051453); *Dsa*CODH from *Desulfofundulus salinum* (WP_121452276); *Dth*CODH from *D. thermocisternus* (WP_084327079); *Rri*CODH from *Rhodospirillum rubrum* (WP_011389181); *Dv*CODH from *Desulfovibrio vulgaris* (WP_010939375); *Dte*CODH from *D. termitidis* (WP_035067836); *Dve*CODH from *Desulfococcus vexinensis* (WP_028588641); *Dha*CODH from *Desulfobaba hansenii* (WP_100393885); *Dm*CODH from *Desulfobulbus mediterraneus* (WP_028583186); *Df*CODH from *Desulfoplanes formicivorans* (WP_069856808); *Ds*CODH from *Dethiosulfatarculus sandiegensis* (WP_044349155); *Fp*CODH from *Ferroglobus placidus* (WP_083777753); *Ha*CODH from *Halodesulfobivrio aestuarii* (WP_027361784); *Sf*CODH from *Syntrophobacter fumaroxidans* (WP_011699717); *Ta*CODH from *Thermodesulfobium acidiphilum* (WP_108307686); *Dh*CODH from *Desulfacinum hydrothermale* (WP_084057374); *Di*CODH from *Desulfacinum infernum* (WP_073036188); *De*CODH from *Desulfosoma caldarium* (WP_123291211); *Me*CODH from *Methanofollis ethanolicus* (WP_067052819); *Mh*CODH from *Methanomethylivorans hollandica* (WP_015324787); *Sa*CODH from *Syntrophorhabdus aromaticivorans* (WP_028894724); *Tn*CODH from *Thermodesulfurhabdus norvegica* (WP_093393560).

Q6. The authors should add the occupancies of Fe, S and Ni atoms ad EV and BV in the description of crystal structures.

→ In response to the reviewer’s request, we have included the occupancies of Fe, S, and Ni atoms, as well as EV and BV, in the descriptions of our crystal structures. For the B and D clusters, the occupancies of all atoms (Fe and S) are set at 1 in all our structures: R57G/N59L_ apo in high PEG (25%, v/v) (PDB id: 8X9D), low PEG (5%, v/v) (PDB id: 8X9E), R57G/N59L with EV (PDB id: 8X9F), R57G/N59L with BV (PDB

id: 8X9G), and F41C (PDB id: 8X9H). Similarly, the occupancies for all atoms in the viologens (EV and BV) are also set at 1. However, the C cluster exhibits varying occupancies for its atoms across different structures. Specifically, the occupancies for Ni, Fe, and S atoms in the C cluster are as follows: 0.7 in R57G/N59L_apo in high PEG (8X9D), 0.7 in R57G/N59L_apo in low PEG (8X9E), 0.62 in R57G/N59L with EV (8X9F), 0.8 in R57G/N59L with BV (8X9G), and 0.7 in the F41C structure (8X9H)

Supplementary Table 9 | Statistics for data collection and refinement

Data set	R57G/N59L apo (PDB ID: 8X9D)	R57G/N59L apo in 5% PEG (PDB ID: 8X9E)	R57G/N59L with EV (PDB ID: 8X9F)	R57G/N59L with BV (PDB ID: 8X9G)	F41C (PDB ID: 8X9H)
A. Data collection					
Energy (keV)	12.398	12.398	12.398	12.398	7.140
Space group	C2	C2	C2	C2	C2
Cell dimensions					
a, b, c (Å)	112.0, 74.6, 71.3	112.1, 75.1, 71.1	112.3, 74.8, 71.2	112.6, 75.4, 72.2	112.1, 75.3, 70.7
α, β, γ (°)	90.0, 111.5, 90.0	90.0, 111.4, 90.0	90.0, 111.2, 90.0	90.0, 111.8, 90.0	90.0, 111.1, 90.0
Resolution range (Å)	50.00 – 2.10 (2.14 – 2.10)*	50.00 – 2.50 (2.54 – 2.50)*	30.00 – 2.50 (2.54 – 2.50)*	50.00 – 3.10 (3.15 – 3.10)*	40.78 – 2.20 (2.21 – 2.20)*
Total/ unique reflections	106,342 / 31,368	71,132 / 18,938	69,338 / 19,005	51,029 / 10,065	112,785 / 27,608
Completeness (%)	99.5 (99.4)*	99.9 (92.8)*	98.0 (98.9)*	98.8 (98.8)*	98.4 (96.8)*
Average $I / \sigma(I)$	14.4 (2.4)*	8.7 (1.4)*	8.4 (1.9)*	9.6 (2.6)*	19.07 (13.0)*
CC _{1/2}	98.9 (82.1)*	99.5 (81.3)*	98.6 (80.9)*	97.4 (87.3)*	99.6 (99.1)*
$R_{\text{merge}}^{\ddagger}$ (%)	8.0 (47.8)*	9.9 (47.4)*	11.4 (43.9)*	14.8 (44.3)*	5.6 (7.8)*
Redundancy	3.4 (3.4)*	3.8 (3.4)*	3.6 (3.7)*	5.1 (4.7)*	4.1 (3.9)*
B. Model refinement statistics					
Resolution range (Å)	33.02 – 2.11	35.22 – 2.50	28.05 – 2.48	35.32 – 3.11	37.70 – 2.20
$R_{\text{work}} / R_{\text{free}}^{\S}$ (%)	16.53 / 21.91	17.66 / 22.69	24.46 / 26.73	22.92 / 27.23	14.11 / 19.31
No. atoms					
Protein	4,682	4,661	4,646	4,646	4,648
Cluster atoms	22	22	21	21	22
EV/BV molecules			16	26	
Water	327	119	117	3	355
B -factors (Å ²)					
Protein	28.75	40.82	37.19	56.27	21.82
Cluster atoms	23.1	38.26	36.50	52.53	24.80
EV/BV molecules			50.71	74.74	
Water	32.64	39.38	35.61	48.27	27.00
R.m.s. deviations					
Bond lengths (Å)	0.009	0.010	0.003	0.003	0.008
Bond angles (°)	1.063	1.145	0.628	0.712	0.979
Ramachandran					
Favored (%)	96.67	96.20	96.99	96.51	97.31
Allowed (%)	3.17	3.65	3.01	3.33	2.38
Outliers (%)	0.16	0.16	0.00	0.16	0.32

* Values in parentheses refer to the highest-resolution shell. Data were collected from one crystal.

$\ddagger R_{\text{merge}} = \frac{\sum_{hkl} \sum_i |I_i(hkl) - \langle I(hkl) \rangle|}{\sum_{hkl} \sum_i I_i(hkl)}$, where $I(hkl)$ is the intensity of reflection hkl , \sum_{hkl} is the sum over all reflections, and \sum_i is the sum over i measurements of reflection hkl .

$\S R = \frac{\sum_{hkl} (|F_{\text{obs}}| - |F_{\text{calc}}|)}{\sum_{hkl} |F_{\text{obs}}|}$, where R_{free} was calculated for a randomly chosen 10% of reflections, which were not used for structural refinement, and R_{work} was calculated for the remaining ones.

REVIEWERS' COMMENTS

Reviewer #1 (Remarks to the Author):

I am very pleased to have received a very carefully revised manuscript based on my comments and have no further comments. Therefore, I strongly recommend the manuscript for publication in Nature Communication.

The authors have sufficiently addressed the comments of reviewer #2. There are a few minor comments that need to be addressed.

Minor comments:

In the legend of Figure 4b, replace "EV and EV" with "EV and BV" and describe the meaning of the dashed black lines in the figure and in Supplementary Figure 12 as well.

In Supplementary Figure 9, define the meaning of the number of binding sites. Are they calculated per monomer or per dimer?

In the legend of Supplementary Figure 11, replace "and 8X9H feature" with "and 8X9H (Supplementary Figure 5) feature".

Since F41C has good resolution ($d_{min} = 2.2 \text{ \AA}$), the active site C-cluster can be better resolved than in Supplementary Figure 5, especially regarding the presence of the OHx ligand on Fe1 as an active form of the C-cluster (Jeoung and Dobbek Science 2007).

However, the functional OHx ligand is not described in the figure. Please comment on this in the figure, legend, or text, as appropriate, especially why the authors did not model it as part of the C cluster.

Reviewer #3 (Remarks to the Author):

I think that the authors have adequately addressed the comments made by the reviewers in the revised version of the manuscript. Therefore, I have no further comments.

REVIEWERS' COMMENTS

Reviewer #1 (Remarks to the Author):

I am very pleased to have received a very carefully revised manuscript based on my comments and have no further comments. Therefore, I strongly recommend the manuscript for publication in *Nature Communication*.

→ We sincerely appreciate the time and effort you dedicated to providing in-depth and meticulous comments, which significantly contributed to enhancing the scientific completeness of our manuscript. The experience gained from this revision process is profoundly valued, serving as a foundation for deep contemplation on various research aspects and a stepping stone towards our research maturity.

Reviewed by Reviewer #1 for Reviewer #2 (Remarks to the Author):

(Reviewer 1 assessed our responses to Reviewer 2, who was unable to assist at the second round)

The authors have sufficiently addressed the comments of reviewer #2. There are a few minor comments that need to be addressed.

→ We extend our gratitude to Reviewer #1 for stepping in to review our manuscript on behalf of Reviewer #2, who was unable to participate in this round. We have addressed the additional minor comments provided by the reviewer.

Minor comments:

Q1. In the legend of Figure 4b, replace "EV and EV" with "EV and BV" and describe the meaning of the dashed black lines in the figure and in Supplementary Figure 12 as well.

→ Following the reviewer's comment, we have corrected the legends for Figure 4b and Supplementary Figure 12 accordingly. “**Figure 4b**, Crystal structures of the R57G/N59L variant in complex with EV and BV (PDB ID: 8X9F and 8X9G, respectively). Interacting residues and electron density maps for EV and BV are illustrated, with *2Fo-Fc* maps at 1σ (blue) and *Fo-Fc* at 2.5σ (green and red mesh indicated by arrows for +/- level), respectively. Colored arrows indicate EV and BV positions in the *Fo-Fc* maps. The black dashed lines indicate electrostatic and hydrophobic interactions between EV, BV, and surrounding residues.”; “**Supplementary Figure 13**, The *omit* map of EV in the R57G/N59L variant is displayed in grey mesh contoured at 2σ . Accompanying this is the *Fo-Fc* difference electron-density map at 2.5σ , with positive and negative levels shown in green and red, respectively, indicated by corresponding colored arrows. The black dashed lines indicate electrostatic and hydrophobic interactions between EV, BV, and surrounding residues. Interacting residues with viologens are represented in stick form.”

Q2. In Supplementary Figure 9, define the meaning of the number of binding sites. Are they calculated per monomer or per dimer?

→ Following the reviewer's comment, we have defined the meaning of the number of binding sites, which are calculated per dimer. “**Supplementary Figure 10** Comparative analysis of binding affinity and the number of binding sites in *ChCODH2* WT and R57G/N59L variant. The number of binding sites indicates the average EV_{ox} molecules bound per each dimeric CODH enzyme. ITC experiments were conducted under anaerobic conditions in a customized glove box to eliminate oxygen interference. The experiments involved a concentration of $52\mu M$ purified CODH enzyme and $2.4\mu L$ of $2.5mM$ EV_{ox} , reacted in CO-saturated buffer ($20mM$ Tris/HCl, pH 7.5) containing $1mM$ tris(2-carboxyethyl)phosphine at $25^{\circ}C$ (see Methods section). Values are the means \pm standard variation, $n = 2$.”

Q3. In the legend of Supplementary Figure 11, replace “and 8X9H feature” with “and 8X9H (Supplementary Figure 5) feature”.

→ Following the reviewer's comment, we have updated the legend of Supplementary Figure 11, replacing “and 8X9H feature” with “and 8X9H (Supplementary Figure 5) feature.” The revised legend reads: “**Supplementary Figure 12 | *Fo-Fc* difference electron density maps and Fe anomalous maps of clusters in R57G/N59L variant.** The *Fo-Fc* difference electron-density maps of B, C, and D clusters in R57G/N59L variant are shown in grey mesh contoured at 3σ . Fe, S, and Ni atoms are coloured orange, yellow and green, respectively. In the C cluster, $1SU7^{21}$ adopts the Ni-4Fe-4S-mut2S (NFS) conformation, while both $1SUF^{21}$ and 8X9H (Supplementary Figure 5) feature the Ni-4Fe-4S (XCC) conformation. The B and C clusters adopt the same conformations in both structures. In R57G/N59L, the S2 atom in the C cluster was not observed, and the XCC model, which lacks the μ_2 -S ligand (S2)^{35,36}, was found to be a more suitable and accurate model for this conformation. Bond lengths of clusters in R57G/N59L variant are presented on each bond. Anomalous difference Fourier maps illustrating the positions of Fe atoms in clusters are shown as orange mesh, contoured at $10, 5,$ and 10σ in each cluster.”

Q4. Since F41C has good resolution ($d_{\min} = 2.2 \text{ \AA}$), the active site C-cluster can be better resolved than in Supplementary Figure 5, especially regarding the presence of the OH_x ligand on Fe1 as an active form of the C-cluster (Jeoung and Dobbek Science 2007). However, the functional OH_x ligand is not described in the figure. Please comment on this in the figure, legend, or text, as appropriate, especially why the authors did not model it as part of the C cluster.

→ In response to the reviewer comment, we reevaluated the C-cluster structure in the F41C variant, focusing on the inclusion of the OH_x ligand on Fe1 as suggested. Despite the high resolution ($d_{\min} = 2.2 \text{ \AA}$) of our structure, we encountered challenges in definitively modeling the OH_x ligand. Specifically, while the F_o-F_c map (green, contoured at 3σ) hinted at a potential site for OH placement next to Fe1, subsequent refinement failed to produce a corresponding $2F_o-F_c$ density map (blue, contoured at 1σ), indicating uncertainty in the presence of the OH ligand at this location. This observation suggests that without clear electron density for the OH, its inclusion in the model remains speculative.

Our analysis indicates that for the F41C variant, like some other variants where an OH map is not observable, the absence of definitive electron density for the OH ligand prevents confident modeling. This issue appears rooted in the resolution of our dataset, which, at 2.2 \AA , may not capture the fine details necessary for unequivocal atomic modeling of small ligands like OH.

Given these constraints, our current model favors the XCC conformation for the F41C variant, as it aligns more closely with our data. In other variants examined, an OH map was visible in the GL apo structure, but not in others, further complicating consistent modeling across different structures.

The fundamental reason for the absence of clear modeling for the OH ligand in F41C, and whether specific factors inhibit its visualization, remains an area for future investigation. Our current analysis, constrained by the available data and resolution, suggests that the XCC conformation is more representative of our findings. This highlights the need for enhanced resolution or additional experimental approaches to clarify these aspects definitively.

Figure. Electron density map around C cluster of F41C before (left) and after (right) OH_x modeling and refinement

Figure. Electron density map around C cluster of GL apo in PEG low before (left) and after (right) OH_x modeling and refinement

Meanwhile, we have included a new figure illustrating the C cluster for 3B51 (Ni-4Fe-4S-OH_x) and explained the absence of OH_x ligand modeling in our structure within both the figure legend and the text of Supplementary Figure 5.

Supplementary Figure 5 | F41C structure and Fe anomalous maps of clusters. **a**, Structural comparison of the D-cluster region in the crystal structures of *ChCODH2* WT (grey, PDB ID: 1SU7)²¹ and F41C variant (light blue, PDB ID: 8X9H). **b**, Anomalous difference Fourier maps illustrating the positions of Fe atoms in B, C, and D clusters contoured at 10, 5, and 10 σ , respectively, are shown in orange mesh. Fe, S, and Ni atoms are coloured orange, yellow and green, respectively. For the C cluster, 1SU7²¹ and 3B51³⁵ exhibit the Ni-4Fe-4S-mut2S (NFS) and Ni-4Fe-4S-OH_x (XCC with OH_x) conformations, respectively, 1SUF²¹ and 8X9H display the Ni-4Fe-4S (XCC) conformation. Both the B and D clusters maintain identical conformations across all structure. In F41C, despite a *Fo-Fc* difference map (3 σ) indicating a potential site next to Fe1, the S2 atom and the OH_x ligand were not modelled within the C cluster due to indeterminate electron density. Consequently, the XCC model, which lacks the μ_2 -S ligand (S2) and OH_x ligand^{35,36}, was found to be a suitable and accurate model for this variant's conformation.

Reviewer #3 (Remarks to the Author):

I think that the authors have adequately addressed the comments made by the reviewers in the revised version of the manuscript. Therefore, I have no further comments.

→ We deeply appreciate your essential comments on maintaining scientific neutrality and focusing on the research balance. Thanks to your positive attention and efforts, we believe that we have been able to communicate our scientific observations more effectively. The authors are very pleased with this outcome.